# Evolution Strategies at the Hyperscale

Bidipta Sarkar [* 1 2]   Mattie Fellows [* 1]   Juan Agustin Duque [* 2 3]
Alistair Letcher [† 1]   Antonio León Villares [† 1]   Anya Sims [† 1]   Clarisse Wibault [† 1]   Dmitry Samsonov [† 4]
Dylan Cope [† 1]   Jarek Liesen [† 1]   Kang Li [† 1]   Lukas Seier [† 1]   Theo Wolf [† 1]   Uljad Berdica [† 1]   Valentin Mohl [† 1]
Alexander David Goldie [1 2]   Aaron Courville [3 5]   Karin Sevegnani [6]   Shimon Whiteson [‡ 2]   Jakob N. Foerster [‡ 1]

## Abstract

Evolution Strategies (ES) is a class of powerful black-box optimisation methods that are highly parallelisable and can handle non-differentiable and noisy objectives. However, naïve ES becomes prohibitively expensive at scale on GPUs due to the low arithmetic intensity of batched matrix multiplications with unstructured random perturbations. We introduce Evolution Guided GeneRal Optimisation via Low-rank Learning (EGGROLL), which improves arithmetic intensity by structuring individual perturbations as rank-$r$ matrices, resulting in a hundredfold increase in training speed for billion-parameter models at large population sizes, achieving up to 91% of the throughput of pure batch inference. We provide a rigorous theoretical analysis of ES for high-dimensional parameter objectives, investigating conditions needed for ES updates to converge in high dimensions, revealing a linearising effect, and proving consistency between EGGROLL and ES as parameter dimension increases. Our experiments show that EGGROLL: (1) enables the stable pretraining of nonlinear recurrent language models that operate purely in integer datatypes, (2) is competitive with GRPO for post-training LLMs on reasoning tasks, and (3) does not compromise performance compared to ES in tabula rasa RL settings, despite being faster.

## 1. Introduction

Evolution Strategies (ES) (Rechenberg, 1978; Beyer, 1995; Beyer & Schwefel, 2002) is an attractive alternative to first-order methods based on gradient backpropagation for several reasons. First, ES does not require differentiability; it can optimise a broader class of models, like those with discrete parametrisations (cellular automata) or objectives for which gradients are unavailable or noisy, such as outcome-only rewards in LLM fine-tuning (Qiu et al., 2025). Second, ES can be more robust to noisy and ill-conditioned optimisation landscapes (Wierstra et al., 2011; Xue et al., 2021). Population-based exploration smooths irregularities (Salimans et al., 2017), tolerates discontinuities, and mitigates issues like ill-conditioned curvature or vanishing and exploding gradients in long-range or recurrent settings (Hansen, 2023). Third, ES is highly amenable to parallel scaling, since fitness evaluations are independent across population members and require only the communication of scalar fitnesses, which maps cleanly onto modern inference infrastructure and yields near-linear speedups on large clusters (Salimans et al., 2017). By contrast, backpropagation requires communicating and aggregating gradients across devices, yielding updates with high memory and computational costs. Furthermore, backpropagation requires special care when training models with low-precision datatypes (Fishman et al., 2025), whereas ES can directly optimise any model with the same datatypes used at inference time. Together, these properties position ES as a potentially powerful tool for training large, discrete, or hybrid architectures, and end-to-end systems with non-differentiable components, including LLMs (Brown et al., 2020; Chowdhery et al., 2023; Du et al., 2022; Fedus et al., 2022).

However, there are currently practical obstacles to employing ES at scale. In deep learning architectures (Goodfellow et al., 2016), the majority of trainable parameters form linear mappings represented by matrices (Rosenblatt, 1962; Hochreiter & Schmidhuber, 1996; Bengio et al., 2000; Krizhevsky et al., 2012; Goodfellow et al., 2014; Kingma & Welling, 2014; Vaswani et al., 2017). Naïvely adapting ES therefore requires generating full-rank matrix perturbations that replicate the entire parameter set for every population member. This inflates memory costs and forces frequent movement of large weight tensors. Evaluating these perturbations then requires a separate sequence of matrix multiplications per member, so the total compute

[1]FLAIR, University of Oxford, Oxford, United Kingdom [2]WhiRL, University of Oxford, Oxford, United Kingdom [3]MILA – Québec AI Institute, Montréal, Québec, Canada [4]NormaCore.dev [5]CIFAR AI Chair [6]NVIDIA AI Technology Center. Correspondence to: Bidipta Sarkar <bidipta.sarkar@eng.ox.ac.uk>, Jakob N. Foerster <jakob.foerster@eng.ox.ac.uk>.

*Proceedings of the 43^{rd} International Conference on Machine Learning*, Seoul, South Korea. PMLR 306, 2026. Copyright 2026 by the author(s).

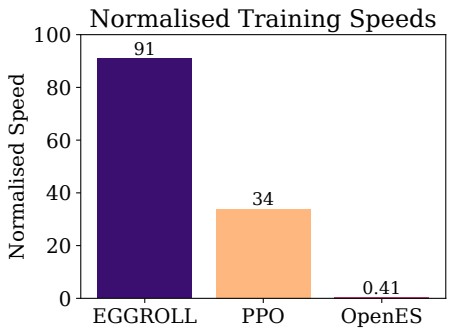
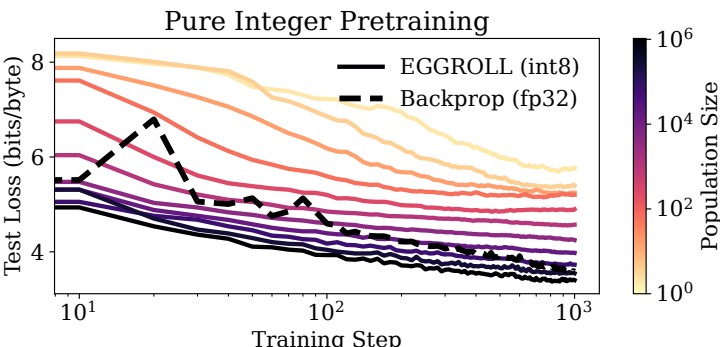

*Figure 1.* (a) Relative speed of our method, EGGROLL, in terms of experience throughput versus prior methods, where 100 is the maximum batch inference throughput. See Appendix E for more details. (b) We use EGGROLL to train an int8 RNN language model from scratch, scaling population size from 2 to 1,048,576 with a fixed data batch size of 16. The dotted line is a fp32 Transformer trained with backprop SGD. EGGROLL's test next-token cross-entropy of 3.40 bits/byte while backprop only gets 3.58 bits/byte.

and wall-clock time scale roughly with the population size and sequence length since batched matrix multiplication has a low arithmetic intensity, i.e., the ratio of arithmetic operations to memory traffic (Williams, 2008). In billion-parameter regimes, these two costs dominate, limiting ES to small models and small populations (Qiu et al., 2025; Korotyshova et al., 2025).

To mitigate both memory and computational bottlenecks, we introduce Evolution Guided GeneRal Optimisation via Low-rank Learning (EGGROLL), an ES algorithm that allows for the efficient training of neural network architectures with billions of parameters. Analogous to LoRA's low-rank adapters in gradient-based training (Hu et al., 2022), EGGROLL generates *low-rank* parameter-space perturbations for ES; instead of sampling a full-rank matrix $E \in \mathbb{R}^{m \times n}$, we sample $A \in \mathbb{R}^{m \times r}$ and $B \in \mathbb{R}^{n \times r}$ with $r \ll \min(m, n)$ and form $E = \frac{1}{\sqrt{r}} AB^\top$. This reduces auxiliary perturbation matrix storage from $mn$ to $(m + n)r$ per layer, and proportionally reduces tensor movement.

Moreover, we use a counter-based deterministic random number generator (RNG) (Salmon et al., 2011; Bradbury et al., 2018) to reconstruct noise on demand, so matrix perturbations need not persist in memory. When evaluating the fitness of members of multiple perturbations in parallel, EGGROLL batches a population of low-rank adapters and shares the base activations, enabling a single forward pass that applies all $AB^\top$ updates via specialised batched matrix multiplications with significantly higher arithmetic intensity, resulting in over a hundredfold increase in training throughput for large neural networks at large population sizes, as shown in Figure 1a. Crucially, EGGROLL does not restrict updates to be low-rank, as the overall update is a weighted average of rank $r$ matrices across the population, making the matrix parameter update rank $\min(Nr, m, n)$.

We analyse the convergence properties of general ES in high dimensions, showing that ES exhibits a critical noise scaling $\sigma_d = o(d^{-1/2})$ under which the update provably linearises and converges to the first-order derivative for a broad class of (possibly discontinuous) objectives. We identify three distinct regimes—linearisation, critical, and divergence—and establish provably tight conditions for stable ES optimisation in large models. Building on this, we extend the analysis to EGGROLL and prove that even fixed low-rank updates (including rank-1) converge to the true ES gradient as dimension grows, despite heavier-tailed perturbations. Our results explain the empirical success of EGGROLL in high-dimensional neural networks and connect its behaviour to neural tangent kernel-style linearisation (Jacot et al., 2018), yielding explicit convergence rates under standard overparameterised regimes. We also provide a rigorous theoretical analysis of the low-rank approximation accuracy, proving that EGGROLL updates converge to the full-rank Gaussian ES updates at a fast $\mathcal{O}(1/r)$ rate.

Furthermore, in our extensive empirical evaluation, we test this hypothesis across a wide range of domains. In tabula rasa and multi-agent RL (MARL) settings, we show that EGGROLL does not compromise performance compared to naïve ES despite being faster. We demonstrate the scalability of EGGROLL for LLM fine-tuning with experiments on pretrained RWKV7 (Peng et al., 2025) models, modern recurrent language models that enable large batch inference due to their constant state size. Finally, we develop a nonlinear RNN language model that operates purely in integer datatypes, and demonstrate that EGGROLL can stably pretrain this language model, a feat which is only feasible due to the large population sizes enabled by EGGROLL. A key finding is that EGGROLL *is effective with rank as small as* $r = 1$, which enables substantial computational and memory savings for negligible decrease in performance.

## 2. Preliminaries

### 2.1. Low Rank Matrix Approximations

When adapting high-dimensional foundation models for specific tasks, updating the parameters using gradient-based methods has high memory requirements. LoRA (Hu et al., 2022) applies low-rank approximations to the matrix multiplications to reduce these costs. For each matrix $M_i \in \mathbb{R}^{m \times n}$ in the model, a low-rank approximation can be made by decomposing each matrix: $M_i \approx M_i^0 + A_i B_i^\top$, where $M_i^0 := \text{StopGrad}(M_i)$ is the imported matrix from the foundation model with frozen parameters and $A_i \in \mathbb{R}^{m \times r}$ and $B_i \in \mathbb{R}^{n \times r}$ are low-width column matrices (i.e., $r \ll \min(m, n)$) whose parameters are updated through gradient-based optimisation during task-specific adaptation. This reduces the number of optimisation parameters for each matrix from $mn$ to $r(m + n)$. EGGROLL uses a similar low-rank approximation for evolutionary strategies.

### 2.2. Matrix Gaussian Distribution & Norms

In this paper, we focus on evolution strategies that target *matrix parameters*. Many variables we study are Gaussian distributed. When working in matrix space, it is convenient to use the matrix Gaussian distribution (Dawid, 1981), which is defined directly over matrices $X \in \mathbb{R}^{m \times n}$: $\mathcal{N}(M, U, V) = \frac{\exp\left(-\frac{1}{2}\text{tr}\left(V^{-1}(X-M)^\top U^{-1}(X-M)\right)\right)}{(2\pi)^{\frac{mn}{2}} \det(U)^{\frac{n}{2}} \det(V)^{\frac{m}{2}}}$, where $M \in \mathbb{R}^{m \times n}$ is the mean matrix, $U \in \mathbb{R}^{m \times m}$ is the row covariance matrix, and $V \in \mathbb{R}^{n \times n}$ is the column covariance matrix. The matrix Gaussian distribution is a generalisation of the multivariate Gaussian distribution $\mathcal{N}(\mu, \Sigma)$ defined over vector space. We denote the $\ell_2$ norm as $\|\cdot\|$. To measure distance between matrices, we use the Frobenius norm: $\|M\|_F := \sqrt{\sum_{i,j} m_{i,j}^2} = \|\text{vec}(M)\|$, which upper-bounds the matrix spectral norm (Petersen & Pedersen, 2012).

### 2.3. Evolution Strategies

Evolution strategies (ES) (Rechenberg, 1978; Beyer, 1995; Beyer & Schwefel, 2002) is a set of black-box optimisation methods that has emerged as a useful alternative to gradient-based methods, particularly for noisy or non-differentiable systems. Our problem setting focuses on fitness functions whose parameters are matrices: our goal is to find a matrix $M^\star \in \mathbb{R}^{m \times n}$ that maximises the fitness function, i.e. $M^\star \in \arg\max_{M \in \mathbb{R}^{m \times n}} f(M)$. Unlike first-order gradient-based methods, which query derivatives $\nabla_M f(M)$ to update $M$, evolutionary methods update a population distribution over the parameter space $\pi(M|\theta)$, which is smoothly parametrised by $\theta \in \Theta$. The problem of optimising the fitness $f(M)$ for $M$ reduces to optimising the parameters of the population $\theta$. This is achieved by solving a *secondary* optimisation problem to maximise the expected fitness under a population distribution $\pi(M|\theta)$ for $\theta$:

$$J(\theta) = \mathbb{E}_{M \sim \pi(M|\theta)}\left[f(M)\right]. \tag{1}$$

Introducing a population distribution *smooths* the fitness landscape; since $\pi(M|\theta)$ is smooth in $\theta$, the resulting objective $J(\theta)$ is also smooth in $\theta$, provided $f(M)$ is measurable and integrable but not necessarily differentiable. Evolution strategies can therefore optimise black-box problems that may be non-differentiable as the derivatives of $J(\theta)$ exist for fitness functions that are discontinuous, yielding a gradient with respect to $\theta$:

$$\nabla_\theta J(\theta) = \mathbb{E}_{M \sim \pi(M|\theta)}\left[\nabla_\theta \log \pi(M|\theta) f(M)\right],$$

where $\nabla_\theta \log \pi(M|\theta)$ is known as the score function. A Monte Carlo estimate is formed by sampling $N$ search matrices $M_i \sim \pi(M_i|\theta)$ and computing an average of the score-weighted fitnesses:

$$\tilde{\nabla}_\theta J(\theta) = \frac{1}{N} \sum_{i=1}^{N} \nabla_\theta \log \pi(M_i|\theta) f(M_i), \tag{2}$$

with which we update $\theta$ via stochastic gradient ascent with a suitable stepsize $\alpha_t$:

$$\theta_{t+1} \leftarrow \theta_t + \alpha_t \tilde{\nabla}_\theta J(\theta_t).$$

ES does not require taking derivatives directly through the fitness function; instead the Monte Carlo update in Equation (2) only requires evaluation of $f(M_i)$ for each $M_i$ to estimate $\nabla_\theta J(\theta)$. As ES only queries $f(M)$ and not $\nabla_M f(M)$, it is a *zeroth-order* optimisation method.

### 2.4. Gaussian Matrix Evolution Strategies

In this paper, we study ES using Gaussian policies: $\pi(M|\theta) = \mathcal{N}(\mu, I_m\sigma, I_n\sigma)$. In addition to its mathematical convenience, the central limit theorem means that the Gaussian distribution emerges naturally from a low-rank approximation as rank increases, even if the matrices $A$ and $B$ are themselves non-Gaussian. Moreover, most widely-used ES algorithms assume (multivariate) Gaussian search distributions (Rechenberg, 1978; Schwefel, 1995; Hansen & Ostermeier, 2001a; Beyer & Schwefel, 2002; Auger & Hansen, 2011; Wierstra et al., 2011; Salimans et al., 2017). In our setting, ES optimises over the mean matrix $\theta = \mu$ which acts as a proxy for the true maximum of the fitness function and the variance parameter $\sigma$ is treated as a hyperparameter to be tuned. By adapting well-known derivations (Wierstra et al., 2011) to the Gaussian matrix case, we derive the ES gradient for our setting in Appendix B: $J(\mu) = \mathbb{E}_{E \sim P(E)}\left[f(M = \mu + \sigma E)\right]$, and the matrix gradient of the ES objective:

$$\nabla_\mu J(\mu) = \frac{1}{\sigma}\mathbb{E}_{E \sim P(E)}\left[E \cdot f(M = \mu + \sigma E)\right], \tag{3}$$

where $P(E)$ is a zero-mean standard normal $p(E) = \mathcal{N}(0, I_m, I_n)$. From Equation (3), Gaussian matrix ES methods optimise the objective $J(\mu)$ by generating search matrices $E \sim P(E)$ from a standard matrix normal distribution $\mathcal{N}(0, I_m, I_n)$ around the parameter matrix $\theta = \mu$.

## 3. Related Work

### 3.1. Evolutionary Algorithms

Evolutionary algorithms have long been a compelling alternative to backpropagation-based training methods (e.g., genetic algorithms (Such et al., 2018) or symbolic evolution (Koza, 1994)). Much research in evolution has focused on developing algorithms for deep learning that scale well to large-scale parallel computation (Jaderberg et al., 2017; Hansen & Ostermeier, 2001b; Salimans et al., 2017). These approaches have increased in popularity following the application of ES to policy learning in deep RL environments (Salimans et al., 2017). Since then, evolution has been widely applied in other domains, such as meta-learning (e.g., (Lu et al., 2022; Metz et al., 2022; Lange et al., 2023; Goldie et al., 2024; 2025)), hyperparameter tuning (e.g., (Parker-Holder et al., 2021; Tani et al., 2021; Vincent & Jidesh, 2023)) and drug discovery (Towers et al., 2025). Here, we consider how to apply ES at a scale beyond the small networks and population sizes of prior work. For example, Salimans et al. (2017) use a maximum population size of 1440, whereas we use over a million.

While low-rank structures have been used in prior evolutionary algorithms, they have been applied to different ends, with different trade-offs, relative to EGGROLL. Choromanski et al. (2019) use a low-rank search space found via principal component analysis, which provides a better search direction to more efficiently use small populations. Garbus & Pollack (2025) optimise a low-rank factorisation instead of the full dense matrix with neuroevolution, achieving similar computational gains to EGGROLL but is limited to the low-rank structure regardless of population size.

### 3.2. Evolution Strategies for LLMs

Although gradient backpropagation is typically used for LLM training and fine-tuning, prior work explores ES variants for fine-tuning. In particular, Zhang et al. (2024)'s two-point zeroth-order gradient estimator, which can be viewed as an ES-inspired method using a single perturbation direction and two function queries per update, is used by Malladi et al. (2023) for memory-efficient LLM fine-tuning. Yu et al. (2025) extend this approach by projecting perturbations to a low-rank subspace, improving convergence. Jin et al. (2024) perform ES directly on LoRA matrices. These works focus on supervised fine-tuning and report performance comparable to full fine-tuning, but do not address whether pretraining is possible with two-point zeroth-order methods; we find that large population sizes are necessary for pretraining,

---

**Algorithm 1** EGGROLL($r, \alpha, \sigma, T_{\max}, N_{\text{workers}}$)

---
**initialise** $\mu$ and workers with known random seeds $\varsigma$
**for** $T_{\max}$ timesteps **do**
  **for** each worker $i \in \{1, \ldots N_{\text{workers}}\}$ in parallel **do**
    $A_i \sim p(A_i), B_i \sim p(B_i)$
    $E_i \leftarrow \frac{1}{\sqrt{r}} A_i B_i^\top$
    $f_i \leftarrow f(\mu + \sigma E_i)$
  **end for**
  workers share scalar fitness $f_i$ with other workers
  **for** each worker $i \in \{1, \ldots N_{\text{workers}}\}$ in parallel **do**
    reconstruct $E_j$ for $j \in \{1, \ldots N_{\text{workers}}\}$ from $\varsigma$
    $\mu \leftarrow \mu + \alpha \frac{1}{N_{\text{Workers}}} \sum_{j=1}^{N_{\text{Workers}}} E_j f_j$
  **end for**
**end for**

---

indicating such methods are unsuitable here.

Recent work also explores ES in the context of LLM reasoning. Korotyshova et al. (2025) first train LoRA adapters using supervised fine-tuning (SFT) before decomposing them into fixed SVD bases alongside singular values that are trained using CMA-ES. They achieve comparable performance to GRPO (Shao et al., 2024) in significantly less wall-clock time on maths reasoning benchmarks. Qiu et al. (2025) directly use ES to optimise all LLM parameters for reasoning, with stronger performance than GRPO on the countdown reasoning task. However, both of these approaches use relatively small population sizes, on the order of a hundred unique perturbations per update, and instead collect hundreds of rollouts per perturbation to efficiently use GPUs. By contrast, our approach allows all generations to use different perturbations, such that our maximum population size per update is orders of magnitude larger (equal to the maximum inference batch size), without compromising token generation throughput.

## 4. EGGROLL

We now introduce EGGROLL (Algorithm 1). A practical issue with using a low-rank matrix approximation is that its distribution and score function have no analytic solution except for degenerate cases, so in Section 4.1 we derive the EGGROLL approximate score function from the limiting high-rank Gaussian. Section 4.2 describes how to efficiently implement EGGROLL on modern hardware.

### 4.1. Low Rank Evolution Strategies

Recall the Gaussian matrix ES update from Equation (3). Our goal is to introduce a tractable approximation to generating full-rank matrices by using low-rank matrices $AB^\top$ as our search matrices instead. Let $p(A)$ and $p(B)$ denote the distribution of $A \in \mathbb{R}^{m \times r}$ and $B \in \mathbb{R}^{n \times r}$.

**Assumption 4.1.** Assume all elements $a_{i,j} \in A$ and $b_{i,j} \in B$ are continuous, identically and independently distributed random variables according to some zero-mean, symmetric,

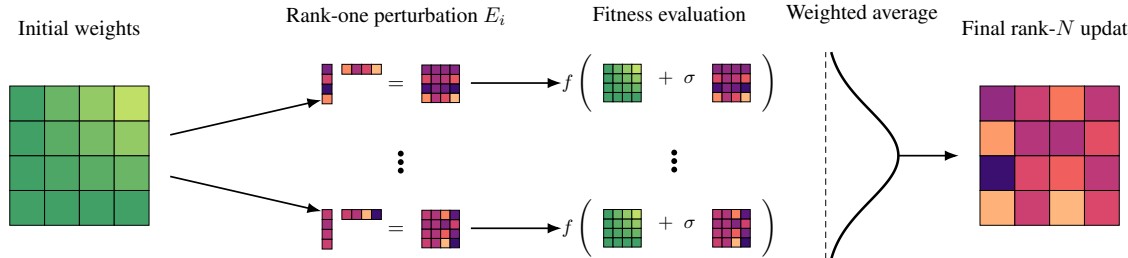

*Figure 2.* Schematic visualization of EGGROLL using $N$ workers.

absolutely continuous distribution $p_0(\cdot)$ with finite fourth-order moments and unit variance.

This assumption is easily satisfied for most perturbation distributions used by ES, including members from the set of generalised Gaussian distributions like Laplace, normal, and uniform distributions. We then form a low rank search matrix:

$$E = \frac{1}{\sqrt{r}} AB^\top.$$

The $\frac{1}{\sqrt{r}}$ scaling ensures the variance of $E$ remains bounded for all $r$. We denote the induced distribution of $E$ as $P(E)$.

$E = \frac{1}{\sqrt{r}} AB^\top$ maps to the manifold $\mathbb{M}^r \subset \mathbb{R}^{m \times n}$ of rank-$r$ matrices. Hence, the density $p(E)$ is defined with respect to a unit volume on the manifold and cannot be defined with respect to the standard unit volume in Euclidean space. For the corresponding score function, gradients with respect to $\log p(E)$ are not defined over the usual Euclidean space. Instead, we use an approximation for the score function $\hat{S}(E)$, yielding our low-rank update:

$$\hat{g}_{\text{LR}} = -\frac{1}{\sigma} \mathbb{E}_{E \sim p(E)} \left[ \hat{S}(E) f(M = \mu + \sigma E) \right]. \quad (4)$$

In our experiments, analysis and Algorithm 1, we use a Gaussian approximate score function:

$$\hat{S}(E) = -E, \quad (5)$$

which is the score function for the Gaussian distribution $\mathcal{N}(0, I_m, I_n)$. This choice is motivated by two theoretical insights from Section 5. The matrix $AB^\top$ can be decomposed as a sum of independent, zero-mean vector outer products. Under Assumption 4.1, the central limit theorem applies to this sum of variables, proving that $P(E)$ converges in distribution to a Gaussian $\mathcal{N}(0, I_m, I_n)$ as rank $r$ increases, recovering the approximate Gaussian score in the limit. Secondly, we investigate the convergence of ES and EGGROLL as the number of parameters grows, proving both updates converge to a linearised form that is consistent with the EGGROLL update using the Gaussian approximate score function.

EGGROLL is not wedded to any particular score function approximator and we derive and explore a set of mean-field approximators in Appendix C.1 as alternatives. However, our experiments show that the Gaussian approximator has the best overall performance on the tasks we consider. To optimise the ES objective using the EGGROLL update, we adapt the parallelised evolutionary strategies algorithm from Salimans et al. (2017). We make a Monte Carlo estimate of the expectation in Equation (4) with $N_{\text{workers}}$ samples to optimise the parameters $\mu$ using (approximate) stochastic gradient ascent. This yields:

---

**EGGROLL UPDATE:** For each worker $i$ (in parallel), sample $A_{i,t} \sim p(A_{i,t})$, $B_{i,t} \sim p(B_{i,t})$ and form a low-rank perturbation $E_{i,t} = \frac{1}{\sqrt{r}} A_{i,t} B_{i,t}^\top$. Update matrix parameters using:

$$\mu_{t+1} = \mu_t + \frac{\alpha_t}{N_{\text{workers}}} \sum_{i=1}^{N_{\text{workers}}} E_{i,t} f(M = \mu_t + \sigma E_{i,t}).$$

$$(6)$$

---

Here we absorb the constant $\frac{1}{\sigma}$ into the tunable learning rate $\alpha_t$. As each random matrix $E_{i,t}$ in Equation (6) has rank $r$ almost surely and the matrix is updated using a sum of $N_{\text{worker}}$ such matrices, the overall EGGROLL matrix parameter update has rank $\min(Nr, m, n)$ almost surely, i.e., the overall parameter update is not restricted to be low rank. For all experiments in Section 6.1, $Nr > \min(m, n)$, i.e., EGGROLL parameter updates are full rank.

### 4.2. Hardware-Efficient Implementation

A key reason to use EGGROLL over standard ES is that large populations can be simulated in parallel on a GPU thanks to the low-rank perturbations. For the sake of exposition, we write equations from the perspective of a single worker, $i$, and explain in text how this corresponds to batched GPU operations.

Consider the task of computing a batched forward pass over inputs $x_i \in \mathbb{R}^{d_{in}}$ for a linear layer with mean parameter $\mu \in \mathbb{R}^{d_{out} \times d_{in}}$. The standard forward pass is just a regular matrix multiplication, $x\mu^T$, since $\mu$ is constant across

all threads. In contrast, naïvely applying ES by trying to compute $x_i(\mu + \sigma E_i)^T$ becomes a batched matrix multiplication, which is inefficient on GPUs since every element of $\mu + \sigma E_i$ is only used in a single multiplication, yielding poor arithmetic intensity.

However, with EGGROLL we know that:

$$x_i(\mu + \sigma E_i)^T = x_i\mu^T + \frac{\sigma}{\sqrt{r}}(x_i B_i)A_i^T,$$

which improves arithmetic intensity since it preserves the efficient general matrix multiplication used in batched inference while adding some additional cheap work per perturbation. In this context, the bulk of compute is spent on the efficient calculation of $x_i\mu^T$ using regular matrix multiplication. Meanwhile, when $r = 1$, $x_i B_i$ simply becomes an inexpensive batch of $N$ vector-vector dot products of length $d_{in}$ to get a batch of $N$ scalars, which is then processed by a batched scalar-vector multiplication when multiplying by $A_i^T$. This decomposition is key to efficient batched LoRA inference, such as those used by vLLM (Kwon et al., 2023), which is why EGGROLL achieves the same speeds as batched LoRA inference systems. The batched LoRA inference enables high arithmetic intensity, enabling us to saturate compute with unique perturbations per input. Note that this is impossible with naïve ES because each perturbation requires a separate matrix-vector multiplication, setting an upper bound of 1 for arithmetic intensity regardless of population size; see Appendix F for a full derivation.

We additionally optimise the update by not explicitly materialising the individual $E_i$ in the computation of $\sum_{i=1}^{N} E_i f_i$, the key term in the Gaussian approximate score function. In particular, when the rank is 1, we reconstruct $A \in \mathbb{R}^{N \times d_{out}}$ and $B \in \mathbb{R}^{N \times d_{in}}$ and calculate the expression as $(\text{diag}(f)A)^T B$, a simple matrix multiplication.

## 5. Analysis

*We provide a full exposition of our analysis with corresponding proofs in Appendices A to D.*

### 5.1. Rank Analysis

Our first analysis investigates how fast the Gaussian score approximation from Equation (5) converges to the true Gaussian ES matrix gradient in Equation (3) as rank $r$ increases. We make notation explicit in $r$ in this subsection, for example writing $E^r = A^r B^{r\top}$. Recall, we denote the true full-rank Gaussian ES gradient as $\nabla_\mu J(\mu)$. Our first theoretical result characterises the error rate between the Gaussian approximate score function in the low-rank update $\hat{g}_{LR}^r$ from Equation (4) and the true gradient using the matrix Frobenius norm, demonstrating a fast $\mathcal{O}(1/r)$ rate:

**Theorem 5.1** (Rank Convergence Rate). *Let Assumption 4.1 hold and assume that $f(M)$ is bounded, that is*

$\sup_M |f(M)| < \infty$, *then:*

$$\|\hat{g}_{LR}^r - \nabla_\mu J(\mu)\|_F = \mathcal{O}\left(\frac{1}{r}\right).$$

We further validate this convergence rate empirically under the Monte Carlo sampling regime encountered in practice, confirming the predicted $\mathcal{O}(1/r)$ behaviour up to a floor noise term arising from finite-sample variance (see Section O for details)

### 5.2. High-Dimensional Gaussian ES

The analysis in Section 5.1 characterises convergence as $r$ grows, but for many EGGROLL applications rank is low, typically $r = 1$. To account for this, we now analyse how ES and EGGROLL updates behave for large parameter models. We first analyse ES in high dimensions before applying these results to the EGGROLL update. In Appendix D we provide an extensive analysis of the ES update for general fitness functions $f(x)$ with parameters in vector space $x \in \mathbb{R}^d$ under Gaussian perturbations. We summarise these results for the matrix ES update with $d = mn$.

Our assumptions are mild, which we formally outline in Appendix D.1. We assume with probability 1 with respect to the random initialisation of $\mu$ that there exists an arbitrarily small ball where $f(M)$ is locally $C^1$ and $\|\nabla_M f(M)\|_F$ is $\alpha$-Hölder, that is:

$$\|\nabla_M f(M_1) - \nabla_M f(M_2)\|_F \leq L\|M_1 - M_2\|_F^\alpha,$$

with $L = \mathcal{O}(1)$. We assume polynomially bounded fitness functions and $\|\nabla_M f(\mu)\|_F$ does not grow with $d$, as is standard and necessary for any high-dimensional analysis (Jacot et al., 2018; Lee et al., 2019; Chizat et al., 2019; Liu et al., 2020). These assumptions encompass essentially all objectives encountered in modern machine learning, including networks with finitely many ReLU activations, max- and hinge-based losses, and other piecewise-smooth or discontinuous models.

In high dimensions, the Gaussian annulus theorem (Vershynin, 2018; Wegner, 2024) proves that the probability mass of standard Gaussian distributions concentrates in thin shells of radius $\sqrt{d}$ as dimension $d = mn$ increases. To counter this, we let $\sigma_d$ depend on $d$ and analyse the critical decay rate of $\sigma_d$ that yields convergence of the ES updates.

**Theorem 5.2** (Convergence to Linearity). *Let Assumptions D.1, D.2, and D.3 hold and $\sigma_d = o\left(d^{-\frac{1}{2}}\right)$. Then:*

$$\|\nabla_\mu J(\mu) - \nabla_M f(\mu)\|_F = \Theta\left(\left(\sigma_d\sqrt{d}\right)^\alpha\right) = o(1),$$

*almost surely with respect to the distribution over $\mu$.*

Theorem 5.2 proves Gaussian ES has a *critical convergence rate* of $\sigma_d = o\left(d^{-\frac{1}{2}}\right)$ in high dimensions, and operates in three regimes.

**Regime I (Convergence):** For $\sigma_d = o\left(d^{-\frac{1}{2}}\right)$, ES converges to a linearised form, recovering a first-order gradient update $\nabla f(\mu)$. This result is analogous to neural tangent kernel (NTK) type theorems which prove that neural networks linearise in high dimensions (Jacot et al., 2018) but applies to a more general set of objectives including discontinuous architectures. Moreover, the rate at which ES converges is tight and cannot in general be improved upon without strengthening continuity or introducing specific structure into the objective to ensure the $L$ decays with $d$.

**Regime II (Critical):** For $\sigma_d \asymp d^{-\frac{1}{2}}$, ES converges to a nonlinear limiting update that may retain higher-order derivative terms when they exist (see Appendix D.2).

**Regime III (Divergence):** For $d^{-\frac{1}{2}} = o\left(\sigma_d\right)$, there exist counterexamples where ES is provably divergent in high dimensions. Our analysis is non-pathological; we study well-behaved $C^\infty$ smooth cubic polynomial objectives, which include a subclass of convex optimisation problems (see Appendix D.2 for details). In practice, $\sigma_d$ is absorbed into the ES update stepsize, and its scale is adjusted automatically as part of the hyperparameter regime to ensure stability.

### 5.3. High-Dimensional EGGROLL

We now extend our high-dimensional analysis to study the EGGROLL update using the Gaussian approximate score function $\hat{g}_{\text{LR}}$ from Equation (5). Taking $r$ as fixed, we analyse the effect of increasing $d = mn$, which explains the success of EGGROLL in high dimensions with rank as small as $r = 1$. Two key differences between full rank Gaussian ES and EGGROLL are that $\hat{g}_{\text{LR}}$ is an approximation to a true gradient and $P(E)$ may have heavier tails than a Gaussian. To account for these differences, we require a slightly stricter local continuity control assumption on the fitness function, replacing local $C^1$-continuity and Hölder control with local $C^2$-continuity and Lipschitz control over Hessians, that is $\nabla_x^2 f(x = \text{vec}(M))$ is $L_d$-Lipschitz locally in a small ball around $\mu$. We also assume that $p_0(\cdot)$ generates sub-Gaussian elements with uniform tail control. We formally state and discuss these assumptions in Appendix D.3. These assumptions still permit discontinuous objectives and low-rank search matrices $A$ and $B$ sampled from Gaussian, bounded, uniform and generalised Gaussian distributions with shape parameter greater than two.

Our final theoretical result proves linearisation of the EGGROLL low-rank update $\hat{g}_{\text{LR}}$ from Equation (4) using the Gaussian approximate score function as $d \to \infty$, which implies convergence to the true ES gradient $\nabla_\mu J(\mu)$.

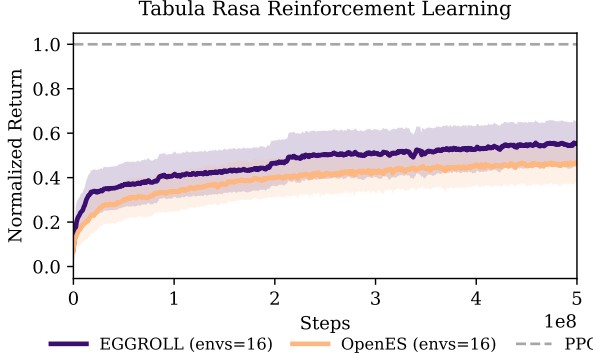

Tabula Rasa Reinforcement Learning

EGGROLL (envs=16)    OpenES (envs=16)    --- PPO

*Figure 3.* Comparison of reinforcement learning returns normalised by PPO performance across 16 environments for 10 seeds. The shaded region is the standard error of the mean.

**Theorem 5.3** (EGGROLL Convergence to Linearity). *Let Assumptions D.2, D.3, D.8 and D.9 hold, $\sigma_d = o(d^{-1/2})$, and $L_d(\sigma_d d)^2 = o(1)$. Then there exists some $K > 0$ such that:*

$$\|\hat{g}_{\text{LR}} - \nabla_M f(\mu)\|_F$$

$$= \mathcal{O}\left(L_d(\sigma_d d)^2\right) + \mathcal{O}\left(\frac{\sqrt{d}}{\sigma_d^2} \exp\left(-K\frac{\rho}{\sqrt{d}\sigma_d}\right)\right),$$

$$= o(1),$$

*and $\|\hat{g}_{\text{LR}} - \nabla_\mu J(\mu)\|_F = o(1)$ almost surely with respect to the distribution over $\mu$.*

For high-dimensional neural networks, standard parametrisations place training in the NTK regime, in which the network behaves approximately linearly in its parameters and gradient descent converges to a global minimum (Jacot et al., 2018; Lee et al., 2019; Chizat et al., 2019). Recent results show that the spectral norm of the Hessian decays polynomially with width, and that higher-order derivatives governing the variation of the Hessian also vanish (Liu et al., 2020). Consequently, the Lipschitz constant $L_d = o(1)$, typically at rate $d^{-\frac{1}{2}}$ or $d^{-1}$ depending on the network architecture. Substituting these rates into our bound yields convergence rates of $\mathcal{O}(\sigma_d^2 d^{\frac{3}{2}})$ or $\mathcal{O}(\sigma_d^2 d)$ respectively.

## 6. Experiments
### 6.1. Reinforcement Learning Tasks
In the following section we showcase the effectiveness of EGGROLL in a variety of tasks that position it as a strong alternative to back-propagation for the end-to-end training of foundation models.

### 6.2. Pure Integer Language Model Pretraining
To demonstrate the potential of EGGROLL as a general optimisation method, we apply it to language model pretraining. Since EGGROLL does not rely on gradients, we explicitly design a language model architecture to be effi-

cient and hardware-friendly at inference time. To highlight EGGROLL's flexibility, we train a nonlinear recurrent neural network (RNN) in pure integer datatypes with no explicit activation functions, relying only on the implicit nonlinearity of clipping in int8 operations. We call the resulting language model EGG, the Evolved Generative GRU, an EGGROLL-friendly architecture with all weights in int8. See Appendix G for more details on the architecture and motivation behind EGG.

We train an EGG model with 6 layers and hidden dimension 256 (6L-256D) to do character-level prediction on the minipile dataset (Kaddour, 2023). We update parameters after 100 tokens for each population member, applying truncated ES by keeping the hidden state and only resetting at document boundaries. We plot the test loss in Figure 1b over training steps across a range of population sizes with a fixed data batch size of 16 sequences per step, where the best test loss is 3.40 bits/byte. With a sufficiently large population size, EGG outperforms a dense 6L-256D Transformer trained with backprop SGD using the same data batch size. Note that larger population sizes require more parallel compute for the same amount of data; our largest population size of $2^{20} = 1048576$ requires around 180 times more GPU-hours than the backprop baseline, demonstrating the potential for compute-only scaling in limited data regimes using EGGROLL.

Our largest population size is $2^{20} = 1048576$, three orders of magnitude larger than the largest experiment done by Salimans et al. (2017) while only requiring a single GPU to train, highlighting EGGROLL's computational efficiency. We note that large population sizes are critical for pretraining; a population size of 2, analogous to MeZO (Malladi et al., 2023), significantly underperforms larger population sizes despite having access to the same data batch. We conduct more ablations in Appendix I, analysing the tradeoff between population size and data batch size.

To verify that low-rank perturbations do not change the optimisation behavior of ES in standard control settings, we benchmark EGGROLL against OpenES (Salimans et al., 2017) across 16 tabula rasa environments spanning Navix, Craftax, Brax, Kinetix, and Jumanji. We use a fixed 3-layer MLP policy (256 hidden units) and perform per-environment hyperparameter optimisation for each method before evaluating the selected configuration over 10 random seeds, reporting mean performance (normalised by PPO) and uncertainty. Overall, EGGROLL is competitive with OpenES on 7/16 environments, underperforms on 2/16, and outperforms on 7/16, while often delivering substantial wall-clock improvements due to its batched low-rank structure (full environment list, learning curves, timing comparisons, and complete HPO ranges/settings are provided in Appendix N.1). Figure 3 shows the averaged normalised return across

the 16 environments with 10 seeds per environment. We additionally report MARL results in Section N.2.

### 6.3. Foundation Model Fine-tuning

We apply EGGROLL to finetune an RWKV-7 (Peng et al., 2025) LLM on two reasoning tasks: countdown (Gandhi et al., 2024) and GSM8K (Cobbe et al., 2021). RWKV is a recurrent model that is better suited to parallelisation than transformers because any memory otherwise spent on the KV cache is used to evaluate population members. Figure 4b shows that EGGROLL fine-tuning on an RWKV-7 1.5B model converges to a higher validation accuracy of 35% (vs. 23%) given the same hardware and wall-clock time in the countdown task. Similarly, Figure 4a shows that EGGROLL outperforms GRPO on GSM8K fine-tuning. Our scoring function draws parallels to the group relative advantage of GRPO. In particular, to score a set of noise directions, $E \equiv \{E_1, \ldots, E_n\}$, we first compute their accuracies, $\{s_{1,q_i}, \ldots, s_{n,q_i}\}$, on $|q| = m$ questions, creating a matrix of scores $S \in \mathbb{R}^{m \times n}$. We then compute the normalised score per question, with the main difference that we use the global variance $\bar{\sigma}$, and average over all the questions to compute a score for the noise direction $E_i$: $\bar{s}_i = \frac{1}{m} \sum_{j=1}^{m} z_{i,q_j} = \frac{1}{m} \sum_{j=1}^{m} \frac{s_{i,j} - \mu_{q_j}}{\bar{\sigma}}$. We use this recipe to train a 14 billion parameter RWKV 7 model on the DeepScaleR dataset and evaluate in more challenging maths reasoning tasks. In this regime, GRPO is infeasible due to the extra memory used by the Adam optimiser (Kingma & Ba, 2014). Using a thinking budget of 5000 tokens for training and evaluation, our fine-tuned 14B model improves from 13% to 30% accuracy on AIME24, from 7% to 33% accuracy on AIME25 and from 11% to 13% accuracy on HMMT25 after training on 32 GPUs for 12 hours (Figure 13b). On 7B models, we outperform GRPO using 128 GPUs for 24 hours (Figure 5a).

In Section L, we achieve similar performance to GRPO when fine-tuning Qwen Transformer models, and additionally demonstrate that EGGROLL can directly optimise for pass@k, a known limitation of GRPO (Yue et al., 2025). Beyond language models, we also fine-tune a finance world model into an agent for high-frequency trading that directly optimises for PnL; see Section M for more details.

### 6.4. Fine-tuning Integer Quantised LLMs

We follow the same procedure as Jacob et al. (2017) to quantise the RWKV-7 family of models by dividing by the maximum *per-channel* value on each weight matrix and mapping into the int8 range of $[-127, 127]$. We then apply EGGROLL with Adam to do model distillation from the original, non-quantised RWKV-7, into the resulting int8 quantised model using examples from GSM8K. See Appendix K for full details about the specifics of quantisation and fine-tuning. The distillation is done by matching the

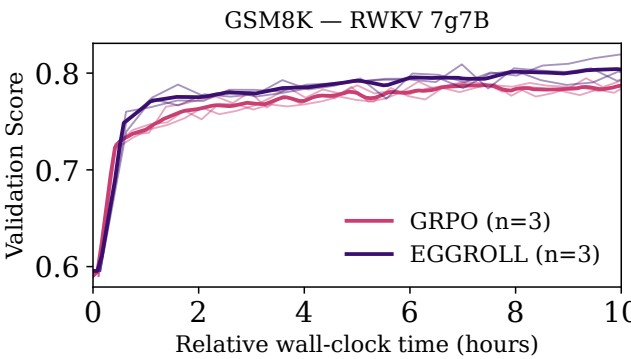
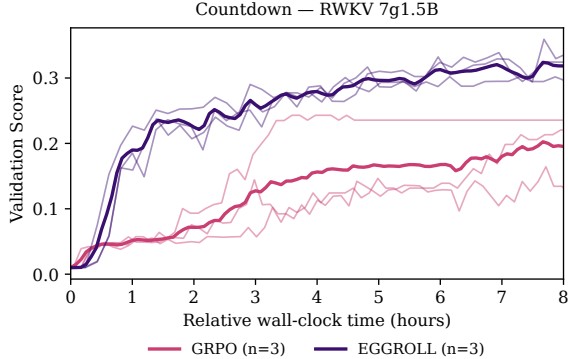

*Figure 4.* (a) Comparison of the validation score of 3 seeds of EGGROLL v.s. 3 seeds of GRPO in GSM8K task with an RWKV 7g7B model on 8 GPUs. EGGROLL allows 8192 parallel generations (1024 per GPU with 260 updates) whereas GRPO only 256 (32 per GPU with 340 updates). (b) Validation score of 3 seeds of EGGROLL v.s. 3 seeds of GRPO in countdown task with an RWKV 7g1.5B model on a single GPU. EGGROLL allows 1024 parallel generations per GPU (618 updates) whereas GRPO only 64 (915 updates).

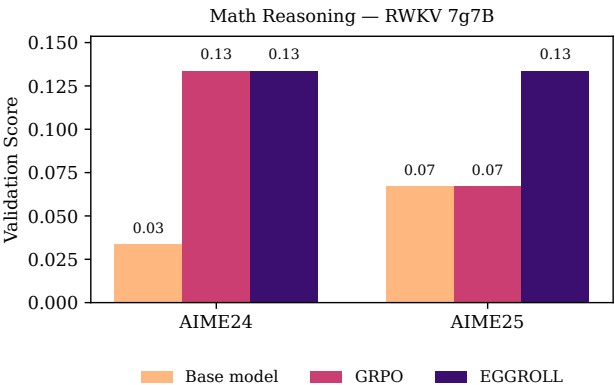
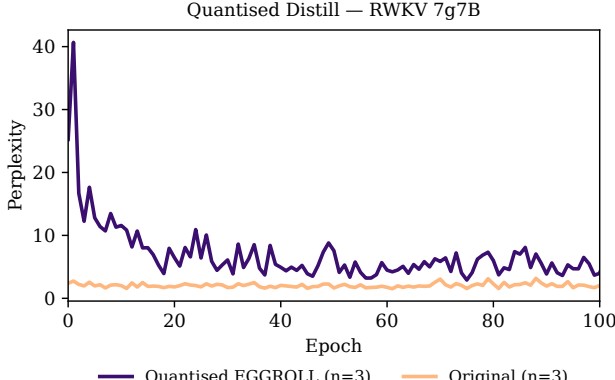

*Figure 5.* (a) Performance of our finetuned RWKV 7G 7 billion model on hard reasoning tasks using 128 GPUs for 12 hours. The model was trained using the DeepScaleR dataset and the best checkpoint was chosen by evaluating on AIME24. (b) Average per token perplexity during training of 3 seeds of a quantised int8 RWKV 7G 7 billion parameter model on distillation from the non-quantised model using examples from GSM8K.

distributions between the quantised and non-quantised models on teacher forced examples (with solutions) from the GSM8K dataset. Figure 5b shows the average per token perplexity of 3 seeds of a quantised RWKV 7G 7 billion parameter model compared to that of the original non-quantised model over the same sequence, as a baseline. Progressively, the quantised model recovers the capability to solve a subset of the GSM8K dataset (Figure 14).

## 7. Conclusion

We introduce EGGROLL, a powerful method for black-box optimisation that scales evolutionary strategies to billion-parameter models and beyond using low-rank search matrices. Our experiments demonstrate that EGGROLL is effective with a rank of 1, giving substantial computational and memory savings for negligible decrease in performance when compared to the full-rank perturbations. Empirically,

EGGROLL delivers large speedups over naïve ES in tabula rasa and multi-agent RL, and can power end-to-end training pipelines for foundation models. Our theoretical analysis shows that the EGGROLL update converges towards the Gaussian ES update with increasing rank $r$ and parameter dimension $d = mn$, and we provide a rigorous study of general ES at high dimensions, deriving necessary and sufficient conditions for convergence and linearisation. Looking forward, we can use EGGROLL for other problems beyond the reach of modern first-order gradient-based techniques. In particular, EGGROLL can enable the training of large scale end-to-end neurosymbolic systems (Sarker et al., 2021) with non-differentiable components. For instance, we can train neural networks that interface with symbolic modules for specific functions, like memory or calculations. We can also optimise end-to-end systems of language models, training them to be aware of inference-time harnesses and interactions with other agents in complex systems.

# Impact Statement

EGGROLL is an optimiser within the class of ES algorithms, and anything that can be optimised via EGGROLL could also be optimised by prior ES algorithms in principle. Nevertheless, making ES practical for large-scale pretraining and fine-tuning of powerful neural networks can have significant consequences, both positive and negative.

In large-scale backpropagation-based training, a key bottleneck is inter-chip memory latency and bandwidth (Fernandez et al., 2025); gradient information needs to be communicated between chips, limiting the potential for globally decentralised training. EGGROLL only requires communicating fitnesses between chips, which typically requires significantly less memory bandwidth than communicating model weights. This makes globally decentralised training practical, since fitnesses can be synchronised onto a common ledger with low bandwidth requirements, and training can be fully auditable under a zero-trust regime, because the model can be reconstructed at any optimisation step using the ledger and individual fitnesses can be recalculated. One potential consequence of this may be an increased demand for consumer GPUs for large-scale deep learning, since the high-bandwidth interconnects offered by enterprise GPUs would not be as pivotal when using EGGROLL.

Since ES can optimise arbitrary objectives, including those that are non-differentiable, it is possible to directly optimise neural networks for human preferences and judgement. This can have a positive impact since we can optimise in a way that humans deem ethical and democratically align the AI to societal beliefs. On the other hand, ES can also optimise for undesirable human propensities, like outrage or addiction, which may even emerge from well-intentioned fitness functions due to reward hacking.

### Acknowledgements

Compute for this project is graciously provided by the Isambard-AI National AI Research Resource, under the projects "FLAIR 2025 Moonshot Projects" and "Robustness via Self-Play RL." Some experiments used compute given by JASMIN, the UK's collaborative data analysis environment (https://www.jasmin.ac.uk).

Bidipta Sarkar is supported by the Clarendon Fund Scholarship in partnership with a Department of Engineering Science Studentship for his Oxford DPhil. Mattie Fellows is funded by a generous grant from the UKRI Engineering and Physical Sciences Research Council EP/Y028481/1. Juan Agustin Duque is supported by the St-Pierre-Larochelle Scholarship at the University of Montreal and by Aaron Courville's CIFAR AI Chair in Representations that Generalize Systematically. Jarek Liesen and Theo Wolf are supported by the EPSRC Centre for Doctoral Training in Autonomous Intelligent Machines & Systems EP/Y035070/1.

Jarek Liesen is also supported by Sony Interactive Entertainment Europe Ltd. Uljad Berdica is supported by the EPSRC Centre for Doctoral Training in Autonomous Intelligent Machines & Systems EP/S024050/1 and the Rhodes Scholarship. Lukas Seier is supported by the Intelligent Earth CDT with funding from the UKRI grant number EP/Y030907/1. Alexander D. Goldie is funded by the EPSRC Centre for Doctoral Training in Autonomous Intelligent Machines and Systems EP/S024050/1. Jakob Nicolaus Foerster is partially funded by the UKRI grant EP/Y028481/1 (originally selected for funding by the ERC). Jakob Nicolaus Foerster is also supported by the JPMC Research Award and the Amazon Research Award.

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

# Appendix

## A. Notation

In our proofs, we use the integral notation $\int$ to denote the integral over the corresponding $\mathbb{R}^d$ space, for example, for a matrix $E \in \mathbb{R}^{m \times n}$, $\int f(E)dE = \int_{\mathbb{R}^{m \times n}} f(E)dE$ and for a vector $E \in \mathbb{R}^{mn}$, $\int f(v)dv = \int_{\mathbb{R}^{mn}} f(v)dv$. For $f : \mathbb{R}^d \to \mathbb{R}$, we use $\nabla f(x)$ to denote the derivative of $f(\cdot)$ evaluated at $x$. We use $\mathrm{vec}(\cdot)$ to denote the vectorisation operator:

$$\mathrm{vec}(M) := [m_{1,1}, \ldots m_{m,1}, m_{1,2}, \ldots m_{m,n}]^\top.$$

For a vector $v \in \mathbb{R}^{mn}$, we define the mat operator as:

$$\mathrm{mat}(v) = \begin{bmatrix} v_1 & v_{m+1} & \cdots & v_{(n-1)m+1} \\ v_2 & v_{m+2} & \cdots & v_{(n-1)m+2} \\ \vdots & \vdots & \ddots & \vdots \\ v_m & v_{2m} & \cdots & v_{mn} \end{bmatrix},$$

so $\mathrm{mat}(\mathrm{vec}(M)) = M$. We will use the fact that the Frobenius norm becomes the $\ell_2$ norm in vector space:

$$\|M\|_F = \sqrt{\sum_{i,j} m_{i,j}^2} = \sqrt{\sum_k \mathrm{vec}(M)_k^2} = \|\mathrm{vec}(M)\|. \tag{7}$$

Our proofs make use of Fourier analysis. For a vector-valued function $f(v) : \mathbb{R}^d \to \mathbb{R}$, we define the Fourier transform as:

$$\tilde{f}(\omega) = \mathcal{F}[f](\omega) := \int f(v) \exp(-i\omega^\top v)dv,$$

and the inverse Fourier transform as:

$$f(v) = \mathcal{F}^{-1}[\tilde{f}](v) := \frac{1}{(2\pi)^d} \int \tilde{f}(\omega) \exp(i\omega^\top v)d\omega,$$

## B. ES Matrix Gradient Deviations

**Proposition 1.** *Using the Gaussian matrix policy $\pi(M|\mu) = \mathcal{N}(\mu, I_m\sigma, I_n\sigma)$ the ES objective can be written as:*

$$J(\mu) = \mathbb{E}_{E \sim P(E)} \left[ f(M = \mu + \sigma E) \right].$$

*and the matrix gradient of the ES objective is:*

$$\nabla_\mu J(\mu) = \frac{1}{\sigma} \mathbb{E}_{E \sim P(E)} \left[ E \cdot f(M = \mu + \sigma E) \right].$$

*where $P(E)$ is a zero-mean standard normal $p(E) = \mathcal{N}(0, I_m, I_n)$.*

*Proof.* We start by deriving the derivative of the ES objective from Equation (1). Taking the derivative with respect to $\mu$:

$$\nabla_\mu J(\mu) = \int f(M) \nabla_\mu \pi(M|\mu)dM,$$

$$= \int f(M) \nabla_\mu \log \pi(M|\mu)\pi(M|\mu)dM.$$

Now,

$$\nabla_\mu \log \pi(M|\mu) = -\frac{1}{2\sigma^2} \nabla_\mu \mathrm{tr} \left( (M - \mu)^\top (M - \mu) \right),$$

$$= \frac{1}{\sigma^2}(M - \mu),$$

where we have used Petersen & Pedersen (2012, Eq. 103) for the matrix derivative of the trace, hence:

$$\nabla_\mu J(\mu) = \frac{1}{\sigma} \int \frac{(M - \mu)}{\sigma} f(M) \pi(M|\mu) dM.$$

We make the transformation of variables $E = \frac{M - \mu}{\sigma}$:

$$\nabla_\mu J(\mu) = \frac{1}{\sigma} \int E \cdot f(M) p(E) dE,$$

where

$$
\begin{aligned}
p(E) &= \sigma^{mn} \pi(M = \mu + \sigma E | \mu) \\
&= \frac{1}{(2\pi)^{\frac{mn}{2}}} \exp\left(-\frac{1}{2} \text{tr}\left(E^\top E\right)\right), \\
&= \mathcal{N}(0, I_m, I_n),
\end{aligned}
$$

as required. Using the same transformation of variables, the alternative form of the matrix Gaussian ES objective follows immediately:

$$
\begin{aligned}
J(\mu) &= \int f(M) \pi(M|\mu) dM, \\
&= \int f(M = \mu + \sigma E) P(E) dE.
\end{aligned}
$$

$\square$

# C. Asymptotic Rank Analysis

For convenience, we work with random vectors in our analysis. We analyse the vector $v^r = \text{vec}(E^r)$, which is the vectorisation of the low-rank matrix $E^r$. We denote $v = \text{vec}(E)$, which is the vectorisation of the full rank matrix $E$. Note $v \sim \mathcal{N}(0, I_d)$ which we denote as $P(v)$. We write $v^r$ as a standardised sum of $r$ independent, zero-mean random vectors. Let

$$u_i = \text{vec}\left(a_i b_i^\top\right), \tag{8}$$

where recall $a_i$ and $b_i$ are the $i$th column vectors of $A$ and $B$ so:

$$v^r = \frac{1}{\sqrt{r}} \sum_{i=1}^{r} u_i.$$

Denoting the covariance matrix of $p(u)$ as $\Sigma_u$, the central limit theorem proves that the distribution of $v^r$ converges in distribution to a zero-mean Gaussian $\mathcal{N}(0, \Sigma_r)$. In Theorem C.1, we derive the covariance matrix for $\Sigma_u$, which we prove is the identity. Our analysis uses an Edgeworth expansion (Bhattacharya & Ranga Rao, 1976) to characterise precisely the rate at which $P(v^r)$ converges to the limiting Gaussian distribution. In Theorem C.2, we make an Edgeworth expansion of $P(v^r)$ to show that it is dominated by $\mathcal{O}\left(\frac{1}{r}\right)$ terms and higher. These are then used to prove Theorem C.4, which allows us to bound the integral of the remainder of the Edgeworth expansion, thereby characterising how fast $P(v^r)$ converges to the limiting Gaussian distribution.

**Lemma C.1.** *Let Assumption 4.1 hold and $u_i$ be defined in Equation* (8)*. Then the variable $u_i$ has identity covariance matrix:*

$$\Sigma_u := \mathbb{E}_{u_i \sim p(u_i)}[u_i u_i^\top] = I_d,$$

*has finite 4th-order absolute moments:*

$$\mathbb{E}_{u_i \sim p(u_i)}\left[\|u_i\|^4\right] < \infty,$$

*and the vector $v^r = vec(E^r)$ is zero-mean and has identity covariance matrix:*

$$\Sigma_v := \mathbb{E}_{v^r \sim P(v^r)}[v^r v^{r\top}] = I_d$$

*Proof.* Under the vec operator, the vector $u_i$ can be written element wise as:

$$u_i = [a_1 b_1, a_2 b_1, \ldots a_m b_1, a_1 b_2, \ldots a_m b_n]^\top.$$

We note that all elements in the vector $u_i$ have zero mean, and so the covariance matrix is the expectation of the outer product:

$$\Sigma_u = \mathbb{E}_{u_i \sim p(u_i)}\left[u_i u_i^\top\right].$$

The diagonal elements of $\Sigma_u$ are:

$$\mathbb{E}_{a_i, b_j}\left[(a_i b_j)^2\right] = \mathbb{E}_{a_i}\left[a_i^2\right] \mathbb{E}_{b_j}\left[b_j^2\right] = 1. \tag{9}$$

As all elements of $a$, $b$ and $\epsilon$ are zero-mean, off-diagonal elements are zero:

$$\mathbb{E}_{a_i, b_j, a_k, b_l}\left[a_i b_j a_k b_l\right] = 0 \quad i \neq k \text{ or } j \neq l. \tag{10}$$

Using Equations (9) and (10), our first result follows:

$$\Sigma_u = I_d.$$

Now, as $u_i$ is a vector of elements which are sums and products of variables which all have finite 4th order moments from Assumption 4.1, it immediately follows that $u$ has finite 4th order absolute moments.

For our final result, we can write $v^r$ as sum of independent variables:

$$v^r = \sum_{i=1}^{r} \left( \frac{1}{\sqrt{r}} u_i \right) = \sum_{i=1}^{r} x_i,$$

where $x_i := \frac{1}{\sqrt{r}} u_i$. As $v^r$ is a sum of zero-mean vectors, it is also zero-mean. We use the fact that the covariance of $r$ i.i.d. random variables is equal to the sum of the individual covariances, hence

$$
\begin{aligned}
\mathbb{E}_{v^r}[v^r v^r] &= r\mathbb{E}_{x_i}[x_i x_i^\top], \\
&= r\mathbb{E}_{u_i}\left[ \frac{1}{r} u_i u_i^\top \right], \\
&= \mathbb{E}_{u_i}\left[ u_i u_i^\top \right], \\
&= I_d,
\end{aligned}
$$

as required. $\qquad\square$

Using Theorem C.1, we see the asymptotic Gaussian density of $v^r$ is a standard normal:

$$g(v^r) = \frac{1}{\sqrt{(2\pi)^d}} \exp\left( -\frac{\|v^r\|^2}{2} \right). \tag{11}$$

which is the density of $P(v)$, where recall $v = \text{vec}(E)$, is the vectorisation of the full rank matrix $E$.

Although $P(v^r)$ does not have a density in the usual sense for low rank $r$, we can still approximate it with a distribution $\hat{p}(v^r)$ by making a Taylor series expansion of its characteristic function, which always exists regardless of whether $P(v^r)$ has a well-defined density or not. We now derive the 4th order Edgeworth expansion for $P(v^r)$. Our proof reveals that 3rd order cumulants control all terms in the expansion that decay at rate $\mathcal{O}\left( \frac{1}{\sqrt{r}} \right)$. As 3rd order cumulants are all zero due to symmetry in Assumption 4.1, the overall decay rate is controlled by $\mathcal{O}\left( \frac{1}{r} \right)$ terms associated with 4th order cumulants. It is for this reason that we obtain a faster convergence rate than the standard central limit theorem.

**Lemma C.2.** *Let Assumption 4.1 hold and let $v^r = \text{vec}(E^r)$ and $u_i$ be defined in Equation (8). Let $g(v^r)$ denote the limiting Gaussian density in Equation (11). Then, the 2nd order Edgeworth expansion of $v^r$ is a distribution $\hat{P}(v^r)$ defined by the approximate density:*

$$\hat{p}(v^r) = g(v^r) + \frac{1}{4! r} g(v^r) \sum_{i,j,k,l} \kappa_{i,j,k,l}^4 H_{i,j,k,l}(v^r),$$

*where:*

$$H_{i,j,k,l}(v^r) := \exp\left( \frac{\|v^r\|^2}{2} \right) \frac{\partial^4}{\partial v_i^r \partial v_j^r \partial v_k^r \partial v_l^r} \exp\left( -\frac{\|v^r\|^2}{2} \right)$$

*is a 4th order Hermite polynomial associated with $g(v^r)$ (Laplace, 1811; Hall, 1992; Withers, 2000).*

*Proof.* We denote the characteristic function of $P(u_i)$ as:

$$\varphi_U(\omega) = \int \exp\left( -i\omega^\top u \right) dP(u),$$

and the characteristic function of $P(v^r)$ as:

$$\varphi_r(\omega) = \int \exp\left( -i\omega^\top u \right) dP(v^r).$$

Recall $v^r = \frac{1}{\sqrt{r}} \sum_{i=1}^{r} u_i$ is the sum of $r$ i.i.d. copies of $\frac{1}{\sqrt{r}} u_i$. Using the scaling property of the Fourier transform, the characteristic function of $\frac{1}{\sqrt{r}} u_i$ is $\varphi_U\left( \frac{\omega}{\sqrt{r}} \right)$. The distribution of a sum of $r$ independent random variables is given by the

$r$-fold convolution of the individual distributions. As convolution in the spatial domain corresponds to multiplication in the frequency domain, the characteristic function of $v^r$ is (Bhattacharya & Ranga Rao, 1976):

$$\varphi_r(\omega) = \left( \varphi_U \left( \frac{\omega}{\sqrt{r}} \right) \right)^r.$$

Taking logarithms yields the log-characteristic function:

$$\log \varphi_r(\omega) = r \log \left( \varphi_U \left( \frac{\omega}{\sqrt{r}} \right) \right),$$
$$= r K_U \left( \frac{\omega}{\sqrt{r}} \right),$$

where $K_U(\omega) := \log \varphi_U(\omega)$. The cumulants are defined by

$$\kappa^{(n)}_{i_1,\ldots,i_n} := i^{-n} \left. \frac{\partial^n K_U(\omega)}{\partial \omega_{i_1} \cdots \partial \omega_{i_n}} \right|_{\omega=0}.$$

The Edgeworth expansion proceeds by a Taylor expansion of $r K_U \left( \frac{\omega}{\sqrt{r}} \right)$ about $\omega = 0$. A 4th order expansion yields:

$$r K_U \left( \frac{\omega}{\sqrt{r}} \right) \approx r K_U(0) + \sqrt{r} \sum_i \omega_i \kappa_i^1 + \frac{1}{2!} \sum_{i,j} \omega_i \omega_j \kappa_{i,j}^2$$
$$+ \frac{1}{3!\sqrt{r}} \sum_{i,j,k} \omega_i \omega_j \omega_k \kappa_{i,j,k}^3 + \frac{1}{4!r} \sum_{i,j,k,l} \omega_i \omega_j \omega_k \omega_l \kappa_{i,j,k,l}^4,$$

where $K_U(0) = 0$. Under Assumption C.6, $u_i$ is symmetric, hence all odd-order cumulants vanish: $\kappa^1 = \kappa^3 = 0$. The second-order cumulant satisfies

$$\sum_{i,j} \omega_i \omega_j \kappa_{i,j}^2 = -\omega^\top \Sigma_u \omega,$$

and from Theorem C.1 we have $\Sigma_u = I$. Substituting yields:

$$r K_U \left( \frac{\omega}{\sqrt{r}} \right) \approx -\frac{\|\omega\|^2}{2} + \frac{1}{4!r} \sum_{i,j,k,l} \omega_i \omega_j \omega_k \omega_l \kappa_{i,j,k,l}^4.$$

Exponentiating and expanding the exponential to first-order in $1/r$ gives:

$$\varphi_r(\omega) = \exp \left( r K_U \left( \frac{\omega}{\sqrt{r}} \right) \right),$$
$$\approx \exp \left( -\frac{\|\omega\|^2}{2} \right) \left( 1 + \frac{1}{4!r} \sum_{i,j,k,l} \omega_i \omega_j \omega_k \omega_l \kappa_{i,j,k,l}^4 \right).$$

Taking the inverse Fourier transform (with the convention $\mathcal{F}^{-1}(f)(v) = (2\pi)^{-d} \int e^{i\omega^\top v} f(\omega) d\omega$) yields:

$$\hat{p}(v^r) = g(v^r) + \frac{1}{4!r} \sum_{i,j,k,l} \kappa_{i,j,k,l}^4 \frac{\partial^4}{\partial v_i^r \partial v_j^r \partial v_k^r \partial v_l^r} g(v^r),$$

and using the identity $H_{i,j,k,l}(v^r) = g(v^r)^{-1} \frac{\partial^4}{\partial v_i^r \partial v_j^r \partial v_k^r \partial v_l^r} g(v^r)$, we recover the stated Edgeworth density. $\qquad \square$

We now apply key results from Bhattacharya & Ranga Rao (1976) to bound the difference in expectation between the low-rank distribution and the Edgeworth approximation as well as the difference in expectation between the true ES Gaussian distribution and the Edgeworth approximation. We require the following boundedness assumption:

**Assumption C.3.** Assume that $f(M)$ is bounded, that is $\sup_M |f(M)| < \infty$.

**Lemma C.4.** *Let $f(v) := f(M = \mu + \sigma mat(v))$, let $P(v) = \mathcal{N}(0, I_d)$, $P(v^r)$ be the distribution of $v^r$ and $\hat{P}(v^r)$ be the 2nd order Edgeworth expansion of $P(v^r)$. Let Assumptions 4.1 and C.3 hold and let $v^r = vec(E^r)$ and $u_i$ be defined in Equation (8). Then:*

$$\left\| \mathbb{E}_{v^r \sim P(v^r)} [v^r \cdot f(v^r)] - \mathbb{E}_{v^r \sim \hat{P}(v^r)} [v^r \cdot f(v^r)] \right\| = \mathcal{O}\left(\frac{1}{r}\right),$$

$$\left\| \mathbb{E}_{v \sim P(v)} [v \cdot f(v)] - \mathbb{E}_{v \sim \hat{P}(v)} [v \cdot f(v)] \right\| = \mathcal{O}\left(\frac{1}{r}\right).$$

*Proof.* From Theorem C.2, we have shown that the Edgeworth expansion for $P(v^r)$ is controlled by 4th order cumulants and higher, that is;

$$\hat{p}(v^r) = g(v^r) + \frac{1}{4!r} g(v^r) \sum_{i,j,k,l} \kappa^4_{i,j,k,l} H_{i,j,k,l}(v^r). \tag{12}$$

We show that the three assumptions needed to apply Bhattacharya & Ranga Rao (1976, Theorem 20.1) to obtain our result using Equation (12) hold. Firstly, the boundedness assumption of the integrand holds:

$$\sup_{v^r} \frac{\|f(v^r)v^r\|}{1 + \|v^r\|} \leq \sup_{v^r} |f(v^r)| < \infty.$$

Secondly, the sampling regularity assumption that $u_i$ (as defined in Equation (8)) is zero-mean i.i.d. (satisfied under Assumption 4.1) with finite 4th order moments (satisfied from Theorem C.1) holds. Let $\varphi_U(\omega)$ denote the characteristic function of $p(u)$, then the final assumption we need to verify is the Cramer condition: $\limsup_{\|\omega\| \to \infty} \varphi_U(\omega) < 1$, which is satisfied from the Riemann-Lebesgue lemma Folland (1999, Theorem 8.22) because $p_0(\cdot)$ is absolutely continuous under Assumption 4.1. Our first result thus follows from applying Bhattacharya & Ranga Rao (1976, Theorem 20.1):

$$\left\| \mathbb{E}_{v^r \sim P(v^r)} [v^r \cdot f(v^r)] - \mathbb{E}_{v^r \sim \hat{P}(v^r)} [v^r \cdot f(v^r)] \right\| = \mathcal{O}\left(\frac{1}{r}\right).$$

We now derive our second result.

$$\mathbb{E}_{v \sim \hat{P}(v)} [v \cdot f(v)] = \int v \cdot f(v) g(v) \left( 1 + \frac{1}{4!r} \sum_{i,j,k,l} \kappa^4_{i,j,k,l} H_{i,j,k,l}(v) \right) dv,$$

$$= \mathbb{E}_{v \sim P(v)} [v \cdot f(v)] - \int v \cdot f(v) g(v) \frac{1}{4!r} \sum_{i,j,k,l} \kappa^4_{i,j,k,l} H_{i,j,k,l}(v) dv,$$

hence

$$\|\mathbb{E}_{v \sim P(v)} [v \cdot f(v)] - \mathbb{E}_{v \sim \hat{P}(v)} [v \cdot f(v)]\| = \frac{1}{r} \left\| \int v \cdot f(v) \frac{1}{4!} \sum_{i,j,k,l} \kappa^4_{i,j,k,l} H_{i,j,k,l}(v) g(v) dv \right\|,$$

$$\leq \frac{1}{r} \int \|v\| \cdot |f(v)| \frac{1}{4!} \sum_{i,j,k,l} |\kappa^4_{i,j,k,l} H_{i,j,k,l}(v)| g(v) dv.$$

Now by definition, $H_{i,j,k,l}(v)$ is a 4th order Hermite polynomial and under Assumption C.3, $|f(v)|$ is bounded, hence $\|v\| \cdot |f(v)| \frac{1}{4!r} \sum_{i,j,k,l} |\kappa^4_{i,j,k,l} H_{i,j,k,l}(v)|$ has polynomial growth of order 5 and is bounded by:

$$\|v\| \cdot |f(v)| \frac{1}{4!} \sum_{i,j,k,l} |\kappa^4_{i,j,k,l} H_{i,j,k,l}(v)| \leq C(1 + \|v\|^5)$$

for some finite $C > 0$. As the expectation of a finite order polynomial under $\mathcal{N}(0, I_d)$ is bounded, it thus follows:

$$\|\mathbb{E}_{v \sim P(v)} [v \cdot f(v)] - \mathbb{E}_{v \sim \hat{P}(v)} [v \cdot f(v)]\| \leq \frac{1}{r} \int C(1 + \|v\|^5) g(v) dv = \mathcal{O}\left(\frac{1}{r}\right),$$

as required.

$\square$

Using Theorem C.4, we have all ingredients needed to derive our main convergence result, which follows after some simple algebra on the norm:

**Theorem C.5.** *Let Assumptions 4.1 and C.3 hold, then:*

$$\|\nabla_\mu J(\mu) - \hat{g}_{\text{LR}}^r\|_F = \mathcal{O}\left(\frac{1}{r}\right).$$

*Proof.* We start by converting the Frobenius norm to vector form using Equation (7):

$$\|\nabla_\mu J(\mu) - g_{\text{LR}}^r\|_F = \left\|\frac{1}{\sigma}\left(\text{vec}\left(\mathbb{E}_E\left[E \cdot f(M = \mu + \sigma E)\right]\right) - \text{vec}\left(\mathbb{E}_{E^r}\left[E^r \cdot f(M = \mu + \sigma E^r)\right]\right)\right)\right\|,$$

$$= \left\|\frac{1}{\sigma}\left(\mathbb{E}_E\left[\text{vec}(E)f(M = \mu + \sigma E)\right] - \mathbb{E}_{E^r}\left[\text{vec}(E^r)f(M = \mu + \sigma E^r)\right]\right)\right\|,$$

$$= \left\|\frac{1}{\sigma}\left(\mathbb{E}_v\left[vf(v)\right] - \mathbb{E}_{v^r}\left[v^r f(v^r)\right]\right)\right\|,$$

where $f(v) := f(M = \mu + \sigma\text{mat}(v))$ and $v = \text{vec}(E)$ is the vectorisation of variable $E$, which is distributed as $v \sim P(v) := \mathcal{N}(0, I_d)$. Let $\hat{P}(v)$ be the distribution for the 2nd order Edgeworth expansion, which we derived in Theorem C.2. Since $\hat{P}(v^r)$ and $\hat{P}(v)$ are identified as the same Edgeworth-expanded distribution on $\mathbb{R}^d$, we may equivalently write:

$$\mathbb{E}_{v^r \sim \hat{P}(v^r)}\left[v^r f(v^r)\right] = \mathbb{E}_{v \sim \hat{P}(v)}\left[v^r f(v)\right],$$

hence:

$$\mathbb{E}_v\left[vf(v)\right] - \mathbb{E}_{v^r}\left[v^r f(v^r)\right] = \mathbb{E}_v\left[vf(v)\right] - \mathbb{E}_{v \sim \hat{P}(v)}\left[vf(v)\right] + \mathbb{E}_{v^r \sim \hat{P}(v^r)}\left[v^r f(v^r)\right] - \mathbb{E}_{v^r}\left[v^r f(v^r)\right],$$

$$\implies \|\nabla_\mu J(\mu) - \hat{g}_{\text{LR}}^r\|_F \leq \frac{1}{\sigma}\left\|\mathbb{E}_v\left[vf(v)\right] - \mathbb{E}_{v \sim \hat{P}(v)}\left[vf(v)\right]\right\|$$

$$+ \frac{1}{\sigma}\left\|\mathbb{E}_{v^r \sim \hat{P}(v^r)}\left[v^r f(v^r)\right] - \mathbb{E}_{v^r}\left[v^r f(v^r)\right]\right\|.$$

Applying Theorem C.4 to each bound yields our desired result:

$$\|\nabla_\mu J(\mu) - \hat{g}_{\text{LR}}^r\|_F = \mathcal{O}\left(\frac{1}{r}\right).$$

$\square$

The convergence rate in Theorem C.5 is faster than the typical $\mathcal{O}\left(1/\sqrt{r}\right)$ rate dictated by the general parametric central limit theorem. Our analysis reveals that this is because for symmetric zero-mean distributions, all odd cumulants are zero (for the same reason that all odd moments of a symmetric distribution are zero). Hence, the rate of convergence to the limiting distribution is controlled by the 4th order term in the Edgeworth expansion, which has rate $\mathcal{O}\left(\frac{1}{r}\right)$. To demonstrate this fast convergence rate, we derive and we plot the negative density $\times$ score function $p(E_{i,j}^r)E_{i,j}^r$ for the marginal distribution $p(E_{i,j}^r)$ in Figure 6, which we derive in Appendix C.1. The figure shows that $p(E_{i,j}^r)E_{i,j}^r$ quickly converges to the limiting function $\frac{E_{i,j}^r}{\sqrt{2\pi}}\exp\left(-\frac{E_{i,j}^r{}^2}{2}\right)$, recovering the Gaussian form from the true Gaussian ES update. Even at $r = 1$, the function is not a poor approximation. After $r = 10$, the function has nearly converged, supporting the hypothesis that the low-rank approximation is accurate even for very low rank regimes $r \ll \min(m, n)$.

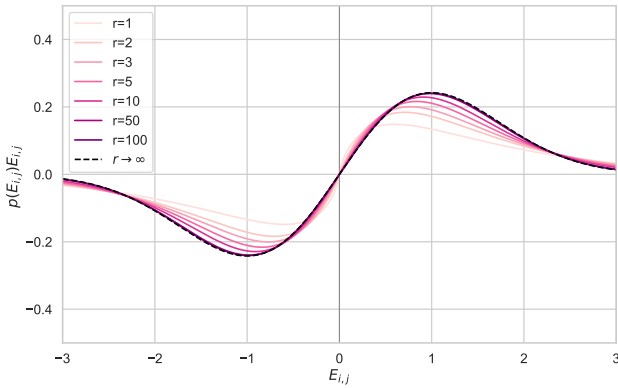

*Figure 6.* Plot of Marginal Score Multiplied by Density for Increasing $r$

## C.1. Mean Field Score Function Approximator

We will use $n$th order Bessel functions of the second kind $K_n(z)$ (Basset, 1888; Macdonald, 1899; Watson, 1944), which are conveniently represented by the integral equations:

$$K_n(z) = \int_0^\infty \exp(-z\cosh\theta)\cosh(n\theta)d\theta.$$

Bessel functions are the solutions to systems of differential equations that occur naturally in phenomena where there is strong radial symmetry, typically involving the propagation of spherical waves from points like the ripples formed from water droplets (Whitham, 1999). For our setting, Bessel functions describe the probability density of the product of rotationally invariant random variables, whose solution is analogous to the interference pattern of two spherical wave propagators.

Using the representation, we find the derivative of the zeroth order function takes the recursive form:

$$\frac{dK_0(z)}{dz} = -\int_0^\infty \exp(-z\cosh\theta)\cosh(\theta)d\theta = -K_1(z). \tag{13}$$

More generally, the derivative of the $n$th order Bessel function is Watson (1944, Section 3.71, Eq. 4):

$$\frac{dK_n(z)}{dz} = \frac{n}{z}K_n(z) - K_{n+1}(z). \tag{14}$$

## C.2. Derivation of Mean-field Approximators

To derive a mean-field approximation, we assume that the elements of $A$ and $B$ are drawn independently from the set of generalised Gaussian distributions (GGDs):

**Assumption C.6.** Assume each element $a_{i,j} \sim \mathcal{GG}(s,p)$ and $b_{i,j} \sim \mathcal{GG}(s,p)$ of $A$ and $B$ is independently distributed according to the zero-mean generalised Gaussian distribution $\mathcal{GG}(s,p)$ with density:

$$\mathcal{GG}(x|s,p) = \frac{p}{2s\Gamma\left(\frac{1}{p}\right)}\exp\left(-\left|\frac{x}{s}\right|^p\right),$$

where $0 < s < \infty$ is the scale parameter, $p > 0$ the shape parameter and $\Gamma(\cdot)$ is the gamma function.

We observe common distributions emerge from the set of GGDs including the Laplace for $p = 1$, the Gaussian for $p = 2$ and the uniform over $[-s, +s]$ in the limit $p \to \infty$.

If we make the assumption that all elements of $E$ are independent (this is true as $r$ grows) then we can write $p(E) \approx \hat{p}(E) := \prod_{i=1}^m \prod_{j=1}^n p(E_{i,j})$ as the product of the marginal distributions. Under this approximation, the score function can be defined element-wise as:

$$[\nabla_E \log p(E)]_{i,j} \approx \hat{S}(E_{i,j}) := \partial_{E_{i,j}} \log p(E_{i,j}).$$

Using this approximation we apply the score function $\hat{S}(\cdot)$ element-wise to the matrix $E$:

$$g_{\text{LR}} \approx \hat{g}_{\text{MF}} := -\frac{1}{\sigma}\mathbb{E}_{E \sim p(E)}\left[f(M = \mu + \sigma E) \cdot \hat{S} \odot (E)\right].$$

For $r = 1$, $\hat{S}(\cdot)$ has a convenient analytic form for all members of the set of GGDs:

**Theorem C.7.** *Let Assumption C.6 hold and $r = 1$. Then the distribution over marginals $p(E_{i,j})$ is:*

$$p(E_{i,j}) = \frac{p}{\left(s\Gamma\left(\frac{1}{p}\right)\right)^2}K_0\left(\frac{2|E_{i,j}|^{\frac{p}{2}}}{s^p}\right), \tag{15}$$

*where $K_0(\cdot)$ is the zeroth-order modified Bessel function of the second kind and the marginal score function is defined element-wise as:*

$$\hat{S}(E_{i,j}) = -\frac{K_1\left(\frac{2|E_{i,j}|^{\frac{p}{2}}}{s^p}\right)}{K_0\left(\frac{2|E_{i,j}|^{\frac{p}{2}}}{s^p}\right)} \cdot \frac{p|E_{i,j}|^{\frac{p}{2}-1}\text{sign}(E_{i,j})}{s^p}.$$

*Proof.* For $r = 1$, we denote the elements of vector $A$ as $a_i$ and elements of vector $B$ as $b_j$, then the elements of matrix $E = AB^\top$ are: $E_{i,j} = a_i b_j$. We now derive the distribution of the unnormalised variables: $E_{i,j}$ using the formula for the distribution of the product of two independent random variables (Rohatgi, 1976; Grimmett & Stirzaker, 1993):

$$p(E_{i,j}) = \int_{-\infty}^{\infty} p(a_i)p\left(b_j = \frac{E_{i,j}}{a_i}\right)\frac{1}{|a_i|}da_i,$$

$$= \left(\frac{p}{2s\Gamma\left(\frac{1}{p}\right)}\right)^2 \int_{-\infty}^{\infty} \exp\left(-\left|\frac{a_i}{s}\right|^p\right)\exp\left(-\left|\frac{E_{i,j}}{a_i s}\right|^p\right)\frac{1}{|a_i|}da_i,$$

$$= 2\left(\frac{p}{2s\Gamma\left(\frac{1}{p}\right)}\right)^2 \int_0^{\infty} \exp\left(-\left|\frac{a_i}{s}\right|^p\right)\exp\left(-\left|\frac{E_{i,j}}{a_i s}\right|^p\right)\frac{1}{|a_i|}da_i,$$

where we have used symmetry of the integrand about 0 to derive the final line. Now, making the substitution $x = \left(\frac{a_i}{s}\right)^p$, we have:

$$\frac{da_i}{dx} = \frac{sx^{\frac{1}{p}-1}}{p}, \quad a_i = sx^{\frac{1}{p}}$$

hence:

$$p(E_{i,j}) = \frac{p}{\left(s\Gamma\left(\frac{1}{p}\right)\right)^2}\frac{1}{2}\int_0^{\infty}\exp\left(-x - \frac{1}{x}\frac{|E_{i,j}|^p}{s^{2p}}\right)\frac{1}{x}dx.$$

Now, we use the identity (Temme, 1996, Theorem 9.42):

$$K_0(z) = \frac{1}{2}\int_0^{\infty}\exp\left(-x - \frac{z^2}{4x}\right)\frac{1}{x}dx,$$

with $z = \frac{2|E_{i,j}|^{\frac{p}{2}}}{s^p}$ to yield:

$$p(E_{i,j}) = \frac{p}{\left(s\Gamma\left(\frac{1}{p}\right)\right)^2}K_0\left(\frac{2|E_{i,j}|^{\frac{p}{2}}}{s^p}\right),$$

as required for Equation (15). Now we derive the marginal score function by applying the chain rule:

$$
\begin{aligned}
\partial_{E_{i,j}} \log p(E_{i,j}) &= \partial_{E_{i,j}} \log K_0 \left( \frac{2|E_{i,j}|^{\frac{p}{2}}}{s^p} \right), \\
&= \partial_z \log K_0 \left( z = \frac{2|E_{i,j}|^{\frac{p}{2}}}{s^p} \right) \partial_{E_{i,j}} \frac{2|E_{i,j}|^{\frac{p}{2}}}{s^p}, \\
&= \partial_z \log K_0 \left( z = \frac{2|E_{i,j}|^{\frac{p}{2}}}{s^p} \right) \frac{p|E_{i,j}|^{\frac{p}{2}-1}\mathrm{sign}(E_{i,j})}{s^p}, \\
&= \frac{\partial_z K_0 \left( z = \frac{2|E_{i,j}|^{\frac{p}{2}}}{s^p} \right)}{K_0 \left( z = \frac{2|E_{i,j}|^{\frac{p}{2}}}{s^p} \right)} \cdot \frac{p|E_{i,j}|^{\frac{p}{2}-1}\mathrm{sign}(E_{i,j})}{s^p}, \\
&= -\frac{K_1 \left( \frac{2|E_{i,j}|^{\frac{p}{2}}}{s^p} \right)}{K_0 \left( \frac{2|E_{i,j}|^{\frac{p}{2}}}{s^p} \right)} \cdot \frac{p|E_{i,j}|^{\frac{p}{2}-1}\mathrm{sign}(E_{i,j})}{s^p},
\end{aligned}
$$

where we have used the identity $\partial_z K_0(x) = -K_1(x)$ from Equation (13). $\qquad \square$

For $r > 1$ we can derive $\hat{S}(\cdot)$ for the Gaussian sampling case:

**Theorem C.8.** *Let Assumption C.6 hold and $p = 2$. Then the distribution over marginals $p(E_{i,j})$ is:*

$$
p(E_{i,j}) = \frac{2\sqrt{r}|\sqrt{r}E_{i,j}|^{\frac{r-1}{2}}}{s^{r+1}\sqrt{\pi}\Gamma\left(\frac{r}{2}\right)} \cdot K_{\frac{r-1}{2}} \left( \frac{2|\sqrt{r}E_{i,j}|}{s^2} \right).
$$

*and the score function is (for $E_{i,j} \neq 0$):*

$$
\hat{S}(E_{i,j}) = \frac{r-1}{E_{i,j}} - \frac{2\sqrt{r}\mathrm{sign}(E_{i,j})}{s^2} \frac{K_{\frac{r+1}{2}}\left( \frac{2|\sqrt{r}E_{i,j}|}{s^2} \right)}{K_{\frac{r-1}{2}}\left( \frac{2|\sqrt{r}E_{i,j}|}{s^2} \right)}.
$$

*Proof.* Each element $E_{i,j}$ is the sum of $r$ independent variables $u_{i,j,l} := a_{i,l}b_{j,l}$ distributed according to Equation (15) with $p = 2$:

$$
E_{i,j} = \frac{1}{\sqrt{r}} \sum_{l=1}^{r} a_{i,l}b_{j,l} = \frac{1}{\sqrt{r}} \sum_{l=1}^{r} u_{i,j,l}.
$$

Let $Z_{i,j} = \sqrt{r}E_{i,j}$, hence:

$$
Z_{i,j} = \sum_{l=1}^{r} u_{i,j,l}.
$$

We first find the density $p(Z_{i,j})$. From Equation (15), the distribution of each $u_{i,j,l}$ is:

$$
p(u_{i,j,l}) = \frac{2}{s^2\pi} K_0 \left( \frac{2|u_{i,j,l}|}{s^2} \right)
$$

We use the fact that the PDF of a sum of $r$ independent random variables (i.e. $Z_{i,j}$) is given by the $r$-fold convolution of the individual PDFs. As convolution in the spatial domain is equal to multiplication in the frequency domain, the PDF $p(Z_{i,j})$ follows by taking Fourier transform of $p(u_{i,j,l})$, taking the power $r$ and then taking the inverse Fourier transform:

$$
p(Z_{i,j}) = \left( \frac{2}{s^2\pi} \right)^r \mathcal{F}^{-1} \left[ \mathcal{F} \left[ K_0 \left( \frac{2|\cdot|}{s^2} \right) \right]^r \right] (Z_{i,j}),
$$

where recall from Section A with $d = 1$, $\mathcal{F}[f](\omega) := \int f(x) \exp(-i\omega x) dx$ denotes the Fourier transform and $\mathcal{F}^{-1}[\tilde{f}](x) := \frac{1}{2\pi} \int \tilde{f}(\omega) \exp(i\omega x) d\omega$, the inverse Fourier transform. Taking the Fourier transform of the Bessel function:

$$
\begin{aligned}
\mathcal{F}\left[K_0\left(\frac{2|\cdot|}{s^2}\right)\right](\omega) &= \int \exp(-i\omega x) K_0\left(\frac{2|x|}{s^2}\right) dx, \\
&= \int \cos(\omega x) K_0\left(\frac{2|x|}{s^2}\right) dx - i \int \sin(\omega x) K_0\left(\frac{2|x|}{s^2}\right) dx, \\
&= \int \cos(\omega x) K_0\left(\frac{2|x|}{s^2}\right) dx, \\
&= 2 \int_0^\infty \cos(\omega x) K_0\left(\frac{2x}{s^2}\right) dx,
\end{aligned}
\tag{16}
$$

where we have used the fact that $K_0\left(\frac{2|x|}{s^2}\right)$ is an even function of $x$ and so its integral with $\sin(\omega x)$ in the second line is zero. Using a standard result, we can evaluate the integral in Equation (16) Gradshteĭn et al. (2015, 6.671 Integral 14):

$$
\mathcal{F}\left[K_0\left(\frac{2|\cdot|}{s^2}\right)\right](\omega) = \frac{\pi}{\sqrt{\omega^2 + \left(\frac{2}{s^2}\right)^2}},
$$

hence:

$$
\begin{aligned}
p(Z_{i,j}) &= \left(\frac{2}{s^2\pi}\right)^r \mathcal{F}^{-1}\left[\frac{\pi^r}{\left(\omega^2 + \left(\frac{2}{s^2}\right)^2\right)^{\frac{r}{2}}}\right](Z_{i,j}), \\
&= \left(\frac{2}{s^2}\right)^r \mathcal{F}^{-1}\left[\left(\omega^2 + \left(\frac{2}{s^2}\right)^2\right)^{-\frac{r}{2}}\right](Z_{i,j}), \\
&= \left(\frac{2}{s^2}\right)^r \frac{1}{2\pi} \int \exp(i\omega Z_{i,j}) \left(\omega^2 + \left(\frac{2}{s^2}\right)^2\right)^{-\frac{r}{2}} d\omega, \\
&= \left(\frac{2}{s^2}\right)^r \frac{1}{2\pi} \left(\int \cos(\omega Z_{i,j}) \left(\omega^2 + \left(\frac{2}{s^2}\right)^2\right)^{-\frac{r}{2}} d\omega \right. \\
&\qquad\qquad\qquad \left. + i \int \sin(\omega Z_{i,j}) \left(\omega^2 + \left(\frac{2}{s^2}\right)^2\right)^{-\frac{r}{2}} d\omega\right), \\
&= \left(\frac{2}{s^2}\right)^r \frac{1}{2\pi} \int \cos(\omega Z_{i,j}) \left(\omega^2 + \left(\frac{2}{s^2}\right)^2\right)^{-\frac{r}{2}} d\omega, \\
&= \left(\frac{2}{s^2}\right)^r \frac{1}{\pi} \int_0^\infty \cos(\omega Z_{i,j}) \left(\omega^2 + \left(\frac{2}{s^2}\right)^2\right)^{-\frac{r}{2}} d\omega,
\end{aligned}
\tag{17}
$$

where we have used the fact that the integrand is an even function and so its integral with $\sin(\omega Z_{i,j})$ is zero to derive the penultimate line. To evaluate the integral in Equation (17) we apply Gradshteĭn et al. (2015, 3.771 Integral 2):

$$
\begin{aligned}
p(Z_{i,j}) &= \left(\frac{2}{s^2}\right)^r \cdot \frac{1}{\sqrt{\pi}\Gamma\left(\frac{r}{2}\right)} \left(\frac{s^2|Z_{i,j}|}{4}\right)^{\frac{r-1}{2}} \cdot K_{\frac{r-1}{2}}\left(\frac{2|Z_{i,j}|}{s^2}\right), \\
&= \frac{2|Z_{i,j}|^{\frac{r-1}{2}}}{s^{r+1}\sqrt{\pi}\Gamma\left(\frac{r}{2}\right)} \cdot K_{\frac{r-1}{2}}\left(\frac{2|Z_{i,j}|}{s^2}\right).
\end{aligned}
$$

Using the transformation of variables $E_{i,j} = \frac{1}{\sqrt{r}} Z_{i,j}$ yields our desired results:

$$p(E_{i,j}) = \sqrt{r} p(Z_{i,j} = \sqrt{r} E_{i,j}),$$

$$= \frac{2\sqrt{r}|\sqrt{r}E_{i,j}|^{\frac{r-1}{2}}}{s^{r+1}\sqrt{\pi}\Gamma\left(\frac{r}{2}\right)} \cdot K_{\frac{r-1}{2}}\left(\frac{2|\sqrt{r}E_{i,j}|}{s^2}\right).$$

Now, we derive the score function:

$$\partial_{E_{i,j}} \log p(E_{i,j}) = \frac{r-1}{2} \cdot \partial_{E_{i,j}} \log|\sqrt{r}E_{i,j}| + \partial_{E_{i,j}} \log K_{\frac{r-1}{2}}\left(\frac{2|\sqrt{r}E_{i,j}|}{s^2}\right),$$

$$= \frac{r-1}{2E_{i,j}} + \frac{2\partial_{E_{i,j}}|\sqrt{r}E_{i,j}|}{s^2} \frac{\partial_x K_{\frac{r-1}{2}}\left(x = \frac{2|\sqrt{r}E_{i,j}|}{s^2}\right)}{K_{\frac{r-1}{2}}\left(\frac{2|\sqrt{r}E_{i,j}|}{s^2}\right)},$$

$$= \frac{r-1}{2E_{i,j}} + \frac{2\sqrt{r}\,\text{sign}(E_{i,j})}{s^2} \frac{\partial_x K_{\frac{r-1}{2}}\left(x = \frac{2|\sqrt{r}E_{i,j}|}{s^2}\right)}{K_{\frac{r-1}{2}}\left(\frac{2|\sqrt{r}E_{i,j}|}{s^2}\right)},$$

Now, from Equation (14) for $E_{i,j} \neq 0$:

$$\frac{\partial_x K_{\frac{r-1}{2}}(x)}{K_{\frac{r-1}{2}}(x)} = \frac{\frac{r-1}{2x} K_{\frac{r-1}{2}}(x) - K_{\frac{r+1}{2}}(x)}{K_{\frac{r-1}{2}}(x)},$$

$$= \frac{r-1}{2x} - \frac{K_{\frac{r+1}{2}}(x)}{K_{\frac{r-1}{2}}(x)},$$

$$\implies \partial_{E_{i,j}} \log p(E_{i,j}) = \frac{r-1}{2E_{i,j}} + \frac{(r-1)\text{sign}(E_{i,j})}{2|E_{i,j}|} - \frac{2\sqrt{r}\,\text{sign}(E_{i,j})}{s^2} \frac{K_{\frac{r+1}{2}}\left(\frac{2|\sqrt{r}E_{i,j}|}{s^2}\right)}{K_{\frac{r-1}{2}}\left(\frac{2|\sqrt{r}E_{i,j}|}{s^2}\right)},$$

$$= \frac{r-1}{2E_{i,j}} + \frac{(r-1)}{2E_{i,j}} - \frac{2\sqrt{r}\,\text{sign}(E_{i,j})}{s^2} \frac{K_{\frac{r+1}{2}}\left(\frac{2|\sqrt{r}E_{i,j}|}{s^2}\right)}{K_{\frac{r-1}{2}}\left(\frac{2|\sqrt{r}E_{i,j}|}{s^2}\right)},$$

$$= \frac{r-1}{E_{i,j}} - \frac{2\sqrt{r}\,\text{sign}(E_{i,j})}{s^2} \frac{K_{\frac{r+1}{2}}\left(\frac{2|\sqrt{r}E_{i,j}|}{s^2}\right)}{K_{\frac{r-1}{2}}\left(\frac{2|\sqrt{r}E_{i,j}|}{s^2}\right)},$$

as required. $\qquad\square$

## D. High-Dimensional Analysis

In this section, we analyse ES in high dimensions. Let $f : \mathbb{R}^d \to \mathbb{R}$ be a fitness function and define the Gaussian ES objective as:

$$J(\mu) = \mathbb{E}_{x \sim \pi(x|\theta)} [f(x)]$$

for $\pi(x|\theta) = \mathcal{N}(\mu, I_d \sigma^2)$. We treat $\sigma_d$ as a non-differentiable parameter, hence $\theta = \mu$. The ES update under $\pi(x|\theta)$ can be written using an expectation under a standard normal distribution (Wierstra et al., 2011; Salimans et al., 2017):

$$\nabla_\mu J(\mu) = \mathbb{E}_{x \sim \mathcal{N}(\mu, I_d)} [\nabla_\mu \log \pi(x|\theta) \cdot f(x)]$$
$$= \frac{1}{\sigma_d} \mathbb{E}_{v \sim \mathcal{N}(0, I_d)} [v \cdot f(x = \mu + \sigma_d v)].$$

The matrix form in Equation (3) is recovered with $E = \text{mat}(v)$, $M = \text{Mat}(x)$, $d = mn$ and the fact sampling a matrix $X \sim \mathcal{N}(M, U, V)$ from a matrix Gaussian distribution is equivalent to sampling a vector $\text{vec}(X) \sim \mathcal{N}(\mu, \Sigma)$ from a multivariate Gaussian distribution with mean $\mu = \text{vec}(M)$ and covariance matrix $\Sigma = V \otimes U$ where $\otimes$ denotes the Kronecker product. For isotropic matrix Gaussian distributions with covariance matrices $U = \sigma I_m$ and $V = \sigma I_n$, the equivalent multivariate Gaussian distribution is also isotropic with $\Sigma = \sigma^2 I_{mn}$. We overload notation slightly by using $\mu$ for both the matrix mean $\mu$ and its vectorisation $\text{vec}(\mu)$, however its meaning is always clear from context.

### D.1. High-Dimensional ES and Convergence

We make the following mild regularity assumptions:

**Assumption D.1** (Locally Continuous Fitness). With probability 1 with respect to the random initialisation of $\mu$, assume there exists a ball $B_\rho(\mu) := \{h| \|h - \mu\| < \rho\}$ of fixed radius $\rho > 0$ where $f(h)$ is $C^1$-continuous for all $h \in B_\rho(\mu)$. Within this ball, let $\nabla f(h)$ be $\alpha$-Hölder continuous, i.e., $\|\nabla f(h) - \nabla f(h')\| \leq L\|h - h'\|^\alpha$ for all $h, h' \in B_\rho(\mu)$, $\alpha \in (0, 1]$ and $L = \mathcal{O}(1)$.

Assumption D.1 *does not restrict the fitness to be globally continuous*; with probability one with respect to the initialisation distribution there must exist an arbitrarily small $C^1$-continuous ball around $\mu$. In particular, discontinuities, kinks, and non-differentiable regions may exist in the domain, provided they are not encountered with nonzero probability in the local region explored by the algorithm. $\alpha$-Hölder is the weakest simple, dimension-robust assumption that guarantees vanishing local gradient variation under Gaussian perturbations; it is weaker than Lipschitz continuity, which is recovered with $\alpha = 1$.

**Assumption D.2** (Global Polynomial Growth). Assume that there exists some constant $0 < C < \infty$ that is $\mathcal{O}(1)$ in $d$ and finite polynomial degree $p \geq 0$ such that $|f(\mu + \sigma_d v)| \leq C(1 + \|\mu + \sigma_d v\|^p)$ and $\|\nabla f(\mu + \sigma_d v)\| \leq C(1 + \|\mu + \sigma_d v\|^p)$ almost surely, or equivalently, in expectation $v \sim \mathcal{N}(0, I_d)$.

Unlike Assumption D.1, this is a *global* assumption. Again, discontinuities can exist. The assumption is weaker than boundedness, is satisfied by essentially all fitness functions used in ES, and ensures that both the objective and its gradient are integrable under Gaussian perturbations; objectives violating this condition typically exhibit super-polynomial growth and derivative growth, which leads to ill-defined or highly unstable ES updates. Moreover, if the condition is not satisfied almost surely, then the function and its gradients are undefined in regions that have nonzero Gaussian measure.

**Assumption D.3** (Bounded Derivative). With probability 1 with respect to the random initialisation of $\mu$, assume that $\|\mu\| = \mathcal{O}(1)$ and $\|\nabla f(\mu)\| = \mathcal{O}(1)$, i.e. $\|\mu\|$ and $\|\nabla f(\mu)\|$ do not grow with increasing $d$.

This assumption is standard and necessary for any high-dimensional analysis proving convergence to linearity as we could not prove convergence to $\|\nabla f(\mu)\|$ if $\|\nabla f(\mu)\|$ diverges. Moreover, the ES update as a whole can diverge if Assumption D.3 is not satisfied. It can be ensured by scaling, typically by scaling networks parameters by $d^{-\frac{1}{2}}$ or using an appropriate scaled initialisation, commonly Gaussian initialisation $\mu \sim \mathcal{N}\left(0, \frac{1}{d} I_d\right)$. This is precisely the scaling employed in the neural tangent kernel (NTK) regime (Jacot et al., 2018; Lee et al., 2019; Chizat et al., 2019), where it guarantees dimension-independent gradients and stable training dynamics.

We use insights from the Gaussian annulus theorem when investigating the convergence properties of high-dimensional ES: our proof relies on the fact that all probability mass converges to the interior of the ball $B_\epsilon(\mu)$ where $\epsilon = \frac{\rho}{2}$ in the limit $d \to \infty$, meaning we only need to consider the smooth region around $\mu$ in the limit. Our first result proves that the mass outside of the ball for any polynomially bounded function tends to zero at an exponential rate.

**Lemma D.4** (Polynomial Tail Bounds). *Let $g(x)$ be polynomial bounded as:*

$$\|g(\mu + \sigma_d v)\| \leq C\|v\|^q (1 + \|\mu + \sigma_d v\|^p),$$

*for some finite polynomial of orders $p$ and $q$ and constant $C > 0$. Let $A_d := \{\|\sigma_d v\| \geq \epsilon\}$ denote the event that a mutation lies outside the a local ball of radius $\epsilon$ around $\mu$. Assume $\sigma_d = o(d^{-1/2})$. Then for some constant $K > 0$:*

$$\left\|\mathbb{E}_{v \sim \mathcal{N}(0,I_d)}\left[g(\mu + \sigma_d v)\mathbb{1}(A_d)\right]\right\| = \mathcal{O}\left(d^{\frac{q}{2}} \exp\left(-K\left(\frac{\epsilon}{\sigma_d}\right)^2\right)\right),$$

*and in particular the right-hand side is $o(1)$ as $d \to \infty$.*

*Proof.* We start by bounding the integrand using the polynomial bound. Denote $\mathbb{P}(A_d) := \mathbb{E}_{v \sim \mathcal{N}(0,I_d)}[\mathbb{1}(A_d)]$. Then, by Jensen's inequality in the first line, polynomial boundedness in the second and $\|a + b\|^p \leq 2^{p-1}(\|a\|^p + \|b\|^p)$ in the third:

$$
\begin{aligned}
\left\|\mathbb{E}_{v \sim \mathcal{N}(0,I_d)}\left[g(\mu + \sigma_d v)\mathbb{1}(A_d)\right]\right\| &\leq \mathbb{E}_{v \sim \mathcal{N}(0,I_d)}\left[\|g(\mu + \sigma_d v)\|\mathbb{1}(A_d)\right], \\
&\leq C\,\mathbb{E}_{v \sim \mathcal{N}(0,I_d)}\left[\|v\|^q(1 + \|\mu + \sigma_d v\|^p)\mathbb{1}(A_d)\right], \\
&\leq C\,\mathbb{E}_{v \sim \mathcal{N}(0,I_d)}\left[\|v\|^q(1 + 2^{p-1}\|\mu\|^p)\mathbb{1}(A_d) + 2^{p-1}\sigma_d^p\|v\|^{p+q}\mathbb{1}(A_d)\right], \\
&= C'\mathbb{E}_{v \sim \mathcal{N}(0,I_d)}\left[\|v\|^q\mathbb{1}(A_d)\right] + C''\sigma_d^p\mathbb{E}_{v \sim \mathcal{N}(0,I_d)}\left[\|v\|^{p+q}\mathbb{1}(A_d)\right].
\end{aligned}
$$

where $C' = C(1 + 2^{p-1}\|\mu\|^p)$ and $C'' = C2^{p-1}$ are constants independent of $d$. Applying the Cauchy–Schwarz inequality to the second expectation gives:

$$\mathbb{E}_{v \sim \mathcal{N}(0,I_d)}[\|v\|^{p+q}\mathbb{1}(A_d)] \leq \sqrt{\mathbb{E}_{v \sim \mathcal{N}(0,I_d)}[\|v\|^{2(p+q)}]} \cdot \sqrt{\mathbb{P}(A_d)}.$$

Now, the variable $\|v\|$ is $\chi_d$-distributed. Using the formula for the $i$-th central moment of $\|v\|$ about the origin (Forbes et al., 2011, Chapter 11.3) yields:

$$\mathbb{E}_{v \sim \mathcal{N}(0,I_d)}\left[\|v\|^i\right] = 2^{\frac{i}{2}}\frac{\Gamma\left(\frac{1}{2}(d+i)\right)}{\Gamma\left(\frac{1}{2}d\right)}.$$

Applying the identity $\frac{\Gamma(z+a)}{\Gamma(z+b)} \sim z^{a-b}$ (Askey & Roy, 2020-2026, Eq. 5.11.12):

$$\mathbb{E}_{v \sim \mathcal{N}(0,I_d)}\left[\|v\|^i\right] \sim 2^{\frac{i}{2}}\left(\frac{d}{2}\right)^{\frac{i}{2}} = d^{\frac{i}{2}}, \tag{18}$$

where $\sim$ denotes asymptotic equivalence. For $i = 2(p + q)$, this yields the bound:

$$\mathbb{E}_{v \sim \mathcal{N}(0,I_d)}[\|v\|^{2(p+q)}] = \mathcal{O}(d^{p+q}),$$

hence:

$$
\begin{aligned}
\left\|\mathbb{E}_{v \sim \mathcal{N}(0,I_d)}\left[g(\mu + \sigma_d v)\mathbb{1}(A_d)\right]\right\| &\leq C'd^{\frac{q}{2}}\sqrt{\mathbb{P}(A_d)} + C''\sigma_d^p d^{\frac{p+q}{2}}\sqrt{\mathbb{P}(A_d)}, \\
&= (C' + C''\sigma_d^p d^{\frac{p}{2}})d^{\frac{q}{2}}\sqrt{\mathbb{P}(A_d)},
\end{aligned}
\tag{19}
$$

We use the Gaussian concentration inequality for the Euclidean norm (Vershynin, 2018, Theorem 3.1.1), which states that for $x \sim \mathcal{N}(0, I_d)$ there exists an absolute constant $K > 0$ such that for all $t \geq 0$,

$$\mathbb{P}\left(\left|\|x\| - \sqrt{d}\right| \geq t\right) \leq 2\exp(-Kt^2).$$

In our setting, we need to bound:

$$\mathbb{P}(A_d) = \mathbb{P}(\|\sigma_d v\| \geq \epsilon) = \mathbb{P}\left(\|v\| \geq \frac{\epsilon}{\sigma_d}\right) = \mathbb{P}\left(\|v\| - \sqrt{d} \geq \frac{\epsilon}{\sigma_d} - \sqrt{d}\right).$$

Setting $t = \frac{\epsilon}{\sigma_d} - \sqrt{d}$, the assumption $\sqrt{d}\sigma_d = o(1)$ implies for sufficiently large $d$ that $\sqrt{d}\sigma_d \leq \epsilon$ and therefore $t \geq 0$, so we can apply the concentration bound to obtain:

$$\mathbb{P}(A_d) = \mathbb{P}\left(\|v\| - \sqrt{d} \geq t\right) \leq \mathbb{P}\left(\left|\|v\| - \sqrt{d}\right| \geq t\right), \tag{20}$$

$$= \mathcal{O}\left(\exp\left(-K\left(\frac{\epsilon}{\sigma_d} - \sqrt{d}\right)^2\right)\right) = \mathcal{O}\left(\exp\left(-K\left(\frac{\epsilon}{\sigma_d}\right)^2\left(1 - \frac{\sigma_d\sqrt{d}}{\epsilon}\right)^2\right)\right).$$

Now, as $\sqrt{d}\sigma_d = o(1)$, it follows $\frac{\sigma_d\sqrt{d}}{\epsilon} = o(1)$, yielding:

$$\mathbb{P}(A_d) = \mathcal{O}\left(\exp\left(-K\left(\frac{\epsilon}{\sigma_d}\right)^2\right)\right),$$

$$\implies \sqrt{\mathbb{P}(A_d)} = \mathcal{O}\left(\exp\left(-\frac{K}{2}\left(\frac{\epsilon}{\sigma_d}\right)^2\right)\right),$$

$$\implies d^{\frac{q}{2}}\sqrt{\mathbb{P}(A_d)} = \mathcal{O}\left(d^{\frac{q}{2}}\exp\left(-\frac{K}{2}\left(\frac{\epsilon}{\sigma_d}\right)^2\right)\right),$$

Applying these results to Equation (19), along with $\sigma_d^p d^{\frac{p}{2}} = \mathcal{O}(d^{\frac{-p}{2}})d^{\frac{p}{2}} = \mathcal{O}(1)$, yields our desired result:

$$\left\|\mathbb{E}_{v\sim\mathcal{N}(0,I_d)}\left[g(\mu + \sigma_d v)\mathbb{1}(A_d)\right]\right\| \leq C'\mathcal{O}\left(d^{\frac{q}{2}}\exp\left(-\frac{K}{2}\left(\frac{\epsilon}{\sigma_d}\right)^2\right)\right) + C''\mathcal{O}(1)\mathcal{O}\left(d^{\frac{q}{2}}\exp\left(-\frac{K}{2}\left(\frac{\epsilon}{\sigma_d}\right)^2\right)\right),$$

$$= \mathcal{O}\left(d^{\frac{q}{2}}\exp\left(-K\left(\frac{\epsilon}{\sigma_d}\right)^2\right)\right).$$

where we have absorbed the factor of $\frac{1}{2}$ into the constant $K$. $\qquad\square$

Our proof in Lemma D.4 reveals the necessity of the condition $\sigma_d\sqrt{d} = o(1)$ for convergence as we can only apply the Gaussian concentration inequality in Equation (20) for $\sigma_d\sqrt{d} = o(1)$; this is a direct consequence of the Gaussian annulus theorem, as for slower rates $1 = o(\sigma_d\sqrt{d})$, the Gaussian probability mass will exit any local ball around $\mu$ and flood the tail, meaning that the tail probability will grow with increasing $d$. Having bounded the tail, convergence to linearity follows by proving convergence within the ball, which allows us to exploit the local $C^1$ smoothness of $f(x)$:

**Theorem D.5** (Convergence to Linearity). *Let Assumptions D.1, D.2 and D.3 hold and $\sigma_d = o\left(d^{-\frac{1}{2}}\right)$. Then:*

$$\|\nabla_\mu J(\mu) - \nabla f(\mu)\| = \Theta\left(\left(\sigma_d\sqrt{d}\right)^\alpha\right) = o(1),$$

*almost surely with respect to the distribution over $\mu$.*

*Proof.* We start with the definition of the ES update:

$$\nabla_\mu J(\mu) = \frac{1}{\sigma_d}\mathbb{E}_{v\sim\mathcal{N}(0,I_d)}\left[v \cdot f(\mu + \sigma_d v)\right].$$

Now let $\epsilon = \frac{\rho}{2}$ where $\rho$ is the radius of the ball from Assumption D.1. Consider the hinge function:

$$\phi(x) = \begin{cases} 1, & \|x\| \leq \epsilon, \\ 2 - \frac{\|x\|}{\epsilon}, & \epsilon < \|x\| < 2\epsilon, \\ 0, & \|x\| \geq 2\epsilon, \end{cases}$$

which interpolates between 1 and 0 in the region $\epsilon < \|x\| < 2\epsilon$. Our first goal is to use $\phi(x)$ to generate a function $\tilde{f}(x)$ that is absolutely continuous and has integrable derivatives outside of $B_\rho(\mu)$ to allow us to apply Stein's lemma (Stein, 1972). We define $\tilde{f}(x)$ as:

$$\tilde{f}(x) = f(x) \cdot \phi(x - \mu)$$

Consider the closed ball $B_\epsilon(\mu) := \{h | \|h - \mu\| \leq \epsilon\}$. We note that within the ball $f(\mu + \sigma_d v)$ remains unchanged:

$$\tilde{f}(\mu + \sigma_d v) = \begin{cases} f(\mu + \sigma_d v), & \|\sigma_d v\| \leq \epsilon, \\ f(\mu + \sigma_d v) \cdot \left(2 - \frac{\|\sigma_d v\|}{\epsilon}\right), & \epsilon < \|\sigma_d v\| < 2\epsilon, \\ 0, & \|\sigma_d v\| \geq 2\epsilon. \end{cases} \tag{21}$$

The derivative of the function with respect to $v$ is:

$$\nabla_v \tilde{f}(\mu + \sigma_d v) = \begin{cases} \sigma_d \nabla f(\mu + \sigma_d v), & \|\sigma_d v\| \leq \epsilon, \\ \sigma_d \nabla f(\mu + \sigma_d v) \cdot \left(2 - \frac{\|\sigma_d v\|}{\epsilon}\right) - \frac{\sigma_d v}{\epsilon \|v\|} \cdot f(\mu + \sigma_d v), & \epsilon < \|\sigma_d v\| < 2\epsilon, \\ 0, & \|\sigma_d v\| \geq 2\epsilon. \end{cases} \tag{22}$$

where the gradient fails to exist only on the sets $\|\sigma_d v\| \in \{\epsilon, 2\epsilon\}$, which have Lebesgue measure zero. We start by using this function to decompose $J(\mu)$ into a smoothed part and a remainder:

$$\nabla_\mu J(\mu) = \underbrace{\frac{1}{\sigma_d} \mathbb{E}_{v \sim \mathcal{N}(0, I_d)} \left[v \cdot \tilde{f}(\mu + \sigma_d v)\right]}_{:=\nabla_\mu \tilde{J}(\mu)} + \underbrace{\frac{1}{\sigma_d} \mathbb{E}_{v \sim \mathcal{N}(0, I_d)} \left[v \cdot (f(\mu + \sigma_d v) - \tilde{f}(\mu + \sigma_d v))\right]}_{:=\Delta(\mu)},$$

Hence:

$$\|\nabla_\mu J(\mu) - \nabla f(\mu)\| \leq \left\|\nabla_\mu \tilde{J}(\mu) - \nabla f(\mu)\right\| + \|\Delta(\mu)\|. \tag{23}$$

Consider the smoothed part:

$$\nabla_\mu \tilde{J}(\mu) := \frac{1}{\sigma_d} \mathbb{E}_{v \sim \mathcal{N}(0, I_d)} \left[v \cdot \tilde{f}(\mu + \sigma_d v)\right].$$

Our goal is to apply Stein's lemma (Stein, 1972) in its multivariate form (Liu, 1994, Lemma 1). The assumptions of (Liu, 1994, Lemma 1) require that the partial derivatives $\partial_{v_i} \tilde{f}(\mu + \sigma_d v)$ are absolutely continuous almost everywhere and:

$$\mathbb{E}_{v \sim \mathcal{N}(0, I_d)} \left[|\partial_{v_i} \tilde{f}(\mu + \sigma_d v)|\right] < \infty.$$

These two conditions are satisfied by construction. Indeed, under Assumption D.1, $f(\cdot)$ is $C^1$ continuous on $B_\rho(\mu)$, hence from Equation (21), $\tilde{f}(\cdot)$ coincides with a compactly supported, piecewise $C^1$ function whose gradient (Equation (22)) exists almost everywhere. Moreover, under Assumption D.2. both $f(\mu + \sigma_d v)$ and $\nabla f(\mu + \sigma_d v)$ are polynomially bounded, and since $\nabla \tilde{f}(\mu + \sigma_d v)$ is supported on $\|\sigma_d v\| \leq 2\epsilon$, it follows that:

$$\mathbb{E}_{v \sim \mathcal{N}(0, I_d)} \left[\|\nabla \tilde{f}(\mu + \sigma_d v)\|\right] < \infty.$$

Applying (Liu, 1994, Lemma 1):

$$\frac{1}{\sigma_d} \mathbb{E}_{v \sim \mathcal{N}(0, I_d)} \left[v \cdot \tilde{f}(\mu + \sigma_d v)\right] = \frac{1}{\sigma_d} \mathbb{E}_{v \sim \mathcal{N}(0, I_d)} \left[\nabla_v \tilde{f}(\mu + \sigma_d v)\right], \tag{24}$$

$$= \mathbb{E}_{v \sim \mathcal{N}(0, I_d)} \left[\nabla \tilde{f}(\mu + \sigma_d v)\right],$$

$$\implies \|\nabla_\mu \tilde{J}(\mu) - \nabla f(\mu)\| = \left\|\mathbb{E}_{v \sim \mathcal{N}(0, I_d)} \left[\nabla \tilde{f}(\mu + \sigma_d v) - \nabla f(\mu)\right]\right\|$$

$$\leq \mathbb{E}_{v \sim \mathcal{N}(0, I_d)} \left[\left\|\nabla \tilde{f}(\mu + \sigma_d v) - \nabla f(\mu)\right\|\right].$$

Let $\{\mu + \sigma_d v \in B_\epsilon(\mu)\} = \{\|\sigma_d v\| \leq \epsilon\}$ denote the event that a mutation lies within the ball $B_\epsilon(\mu)$. We now split the integral into two regions, the first within the ball and the second outside:

$$\mathbb{E}_{v \sim \mathcal{N}(0, I_d)} \left[\|\nabla f(\mu + \sigma_d v) - \nabla f(\mu)\|\right] = \underbrace{\mathbb{E}_{v \sim \mathcal{N}(0, I_d)} \left[\left\|\nabla \tilde{f}(\mu + \sigma_d v) - \nabla f(\mu)\right\| \mathbb{1}(\|\sigma_d v\| \leq \epsilon)\right]}_{:=I_{\text{loc}}}$$

$$+ \underbrace{\mathbb{E}_{v \sim \mathcal{N}(0, I_d)} \left[\left\|\nabla \tilde{f}(\mu + \sigma_d v) - \nabla f(\mu)\right\| \mathbb{1}(\|\sigma_d v\| > \epsilon)\right]}_{:=I_{\text{tail}}}.$$

Consider the region inside the ball, $I_{\text{loc}}$. From Equation (22), $\nabla \tilde{f}(\mu + \sigma_d v) = \nabla f(\mu + \sigma_d v)$ within this region. Using the local $\alpha$-Hölder continuity from Assumption D.1:

$$
\begin{aligned}
I_{\text{loc}} &= \mathbb{E}_{v \sim \mathcal{N}(0, I_d)} \left[ \|\nabla f(\mu + \sigma_d v) - \nabla f(\mu)\| \, \mathbb{1}(\|\sigma_d v\| \leq \epsilon) \right], \\
&\leq L \mathbb{E}_{v \sim \mathcal{N}(0, I_d)} \left[ \|\sigma_d v\|^\alpha \, \mathbb{1}(\|\sigma_d v\| \leq \epsilon) \right], \\
&\leq \sigma_d{}^\alpha L \mathbb{E}_{v \sim \mathcal{N}(0, I_d)} \left[ \|v\|^\alpha \right].
\end{aligned}
$$

Now, applying the identity $\mathbb{E}_{v \sim \mathcal{N}(0, I_d)} \left[ \|v\|^i \right] \sim d^{\frac{i}{2}}$, from Equation (18):

$$
I_{\text{loc}} = \mathcal{O}\left( \left( \sigma_d \sqrt{d} \right)^\alpha \right).
$$

We now bound the tail region outside the ball:

$$
\begin{aligned}
I_{\text{tail}} &= \mathbb{E}_{v \sim \mathcal{N}(0, I_d)} \left[ \left\| \nabla \tilde{f}(\mu + \sigma_d v) - \nabla f(\mu) \right\| \, \mathbb{1}(\|\sigma_d v\| > \epsilon) \right], \\
&\leq \mathbb{E}_{v \sim \mathcal{N}(0, I_d)} \left[ \left\| \nabla \tilde{f}(\mu + \sigma_d v) - \nabla f(\mu) \right\| \, \mathbb{1}(\|\sigma_d v\| \geq \epsilon) \right].
\end{aligned}
$$

Now, as $\|\nabla f(\mu)\| = \mathcal{O}(1)$ from Assumption D.3 and we have established that $\|\nabla \tilde{f}(\mu + \sigma_d v)\|$ is polynomial bounded under Assumption D.2 when applying Stein's lemma, it follows that $\left\| \nabla \tilde{f}(\mu + \sigma_d v) - \nabla f(\mu) \right\|$ is also polynomial bounded, that is there exists some constant $C > 0$ and finite polynomial order $p$ such that:

$$
\left\| \nabla \tilde{f}(\mu + \sigma_d v) - \nabla f(\mu) \right\| \leq C(1 + \|\mu + \sigma_d v\|^p).
$$

Applying Lemma D.4, it follows:

$$
I_{\text{tail}} = \mathcal{O}\left( \exp\left( -K \left( \frac{\epsilon}{\sigma_d} \right)^2 \right) \right),
$$

for some constant $K > 0$. Together, this yields:

$$
\begin{aligned}
\|\nabla_\mu \tilde{J}(\mu) - \nabla f(\mu)\| &= I_{\text{loc}} + I_{\text{tail}}, \\
&= \mathcal{O}\left( \left( \sigma_d \sqrt{d} \right)^\alpha \right) + \mathcal{O}\left( \exp\left( -K \left( \frac{\epsilon}{\sigma_d} \right)^2 \right) \right).
\end{aligned}
$$

As $\exp(-x) = o\left( x^{-a} \right)$ for any $a > 0$, we take $a = \alpha/2$ to obtain a weakened bound matching the first term:

$$
\exp\left( -K \left( \frac{\epsilon}{\sigma_d} \right)^2 \right) = o\left( \left( \frac{\sigma_d}{\epsilon} \right)^\alpha \right) = o\left( \left( \sigma_d \sqrt{d} \right)^\alpha \right).
$$

This yields the upper bound:

$$
\|\nabla_\mu \tilde{J}(\mu) - \nabla f(\mu)\| = \mathcal{O}\left( \left( \sigma_d \sqrt{d} \right)^\alpha \right). \tag{25}
$$

Returning to Equation (23), we must bound the remainder term:

$$
\begin{aligned}
\|\Delta(\mu)\| &= \left\| \frac{1}{\sigma_d} \mathbb{E}_{v \sim \mathcal{N}(0, I_d)} \left[ v \cdot (f(\mu + \sigma_d v) - \tilde{f}(\mu + \sigma_d v)) \right] \right\|, \\
&\leq \frac{1}{\sigma_d} \mathbb{E}_{v \sim \mathcal{N}(0, I_d)} \left[ \|v\| \cdot \left| (f(\mu + \sigma_d v) - \tilde{f}(\mu + \sigma_d v)) \right| \right].
\end{aligned}
$$

Again, from Assumption D.2, it follows that $\left| (f(\mu + \sigma_d v) - \tilde{f}(\mu + \sigma_d v)) \right|$ is polynomially bounded, that is there exists some constant $C' > 0$ and finite polynomial order $p'$ such that:

$$
\left| (f(\mu + \sigma_d v) - \tilde{f}(\mu + \sigma_d v)) \right| \leq C'(1 + \|\mu + \sigma_d v\|^p).
$$

Applying Lemma D.4 with $q = 1$:

$$\|\Delta(\mu)\| = \mathcal{O}\left(\frac{d^{\frac{1}{2}}}{\sigma_d}\exp\left(-K\left(\frac{\epsilon}{\sigma_d}\right)^2\right)\right).$$

Now, as $\exp(-x) = o\left(x^{-1}\right)$ for $x \to \infty$, it follows:

$$\exp\left(-K\left(\frac{\epsilon}{\sigma_d}\right)^2\right) = o\left(\sigma_d^2\right),$$

$$\implies \|\Delta(\mu)\| = \mathcal{O}\left(\frac{d^{\frac{1}{2}}}{\sigma_d}\exp\left(-K\left(\frac{\epsilon}{\sigma_d}\right)^2\right)\right),$$

$$= o(\sigma_d\sqrt{d}),$$

$$= o\left((\sigma_d\sqrt{d})^\alpha\right), \tag{26}$$

where the final line follows from the fact $\sqrt{d}\sigma_d = o(1)$. Assembling our bounds using Ineq. 23 yields our desired result:

$$\|\nabla_\mu J(\mu) - \nabla f(\mu)\| \le \underbrace{\left\|\nabla_\mu \tilde{J}(\mu) - \nabla f(\mu)\right\|}_{=O\left((\sigma_d\sqrt{d})^\alpha\right),\ \text{Eq. } 25} + \underbrace{\|\Delta(\mu)\|}_{=o\left(((\sigma_d\sqrt{d})^\alpha\right),\ \text{Eq. } 26} = O\left((\sigma_d\sqrt{d})^\alpha\right).$$

We now show that the bound is tight. Consider the function $f(x) = \frac{L}{2}\sum_{i=1}^d x_i|x_i| + a^\top x$ where $\|a\| = \mathcal{O}(1)$. Taking partial derivatives:

$$\partial_i f(x) = L|x_i| + a_i, \tag{27}$$

hence:

$$\|\nabla f(x) - \nabla f(y)\| = L\sqrt{\sum_{i=1}^d (|x_i| - |y_i|)^2}$$

Applying the reverse triangle inequality $||x_i| - |y_i|| \le |x_i - y_i| \implies (|x_i| - |y_i|)^2 \le (x_i - y_i)^2$:

$$\|\nabla f(x) - \nabla f(y)\| \le L\sqrt{\sum_{i=1}^d (x_i - y_i)^2} = L\|x - y\|.$$

We have thus shown that $f(x)$ is $C^1$-continuous and its gradient has Lipschitz constant $L$, i.e. $\alpha = 1$ with Hölder constant $L$. It is also bounded by a polynomial of order 2. Without loss of generality, we take a deterministic initialisation $\mu = 0$ to simplify algebra, yielding;

$$\nabla f(\mu) = a \implies \|\nabla f(\mu)\| = \|a\| = \mathcal{O}(1).$$

$f(x)$ thus satisfies Assumptions D.1, D.2 and D.3. Using $f(x)$ as the fitness:

$$\nabla_\mu J(\mu) - \underbrace{\nabla f(\mu)}_{=a} = \frac{1}{\sigma_d}\mathbb{E}_{v\sim\mathcal{N}(0,I_d)}\left[v \cdot f(\sigma_d v)\right] - a,$$

$$= \mathbb{E}_{v\sim\mathcal{N}(0,I_d)}\left[\nabla f(\sigma_d v)\right] - a;$$

Taking expectations element-wise and using Equation (27):

$$[\nabla_\mu J(\mu) - \nabla f(\mu)]_i = \mathbb{E}_{v\sim\mathcal{N}(0,I_d)}\left[\partial_i f(\sigma_d v)\right] - a_i,$$

$$= \sigma_d L\mathbb{E}_{v_i\sim\mathcal{N}(0,1)}[|v_i|].$$

Applying Equation (18):

$$\mathbb{E}_{v_i \sim \mathcal{N}(0,1)}[|v_i|] = \sqrt{2}\frac{\Gamma(1)}{\Gamma(\frac{1}{2})} = \sqrt{\frac{2}{\pi}},$$

Hence:

$$\|\nabla_\mu J(\mu) - \nabla f(\mu)\| = \sigma_d \sqrt{d} \cdot L \sqrt{\frac{2}{\pi}},$$

thereby attaining the upper bound rate of $\sigma_d \sqrt{d}$. $\qquad\square$

### D.2. Critical Convergence Rate

To show that our rate is critical, we investigate the space of functions that can be represented by cubic polynomials of the form:

$$f(x) = a^\top x + \frac{1}{2}x^\top Bx + \frac{1}{6}C(x,x,x),$$

where $a \in \mathbb{R}^d$, $B \in \mathbb{R}^{d \times d}$ and $C(x,x,x) = \sum_{i,j,k} c_{i,j,k} x_i x_j x_k$ denotes a symmetric 3-linear map represented by the 3-tensor $C \in \mathbb{R}^{d \times d \times d}$.

These are non-pathological well-behaved, analytic $C^\infty$ smooth functions and include a rich subclass of convex optimisation problems, for instance, cubic perturbations of strictly convex quadratics. Moreover, any convex $C^3$ objective admits a local third-order Taylor expansion of this form around a minimiser. We make the following regularity assumption:

**Assumption D.6.** Assume $\|a\| = \mathcal{O}(1)$, $\|B\| = \mathcal{O}(1)$, $\|C\| = \mathcal{O}(1)$ and $\|\mu\| = \mathcal{O}(1)$ where $\|\cdot\|$ denotes operator norm for $i$-tensor $T(x_1, \ldots x_i)$:

$$\|T\| := \sup_{\|x_1\|=\cdots=\|x_i\|=1} |T(x_1, \ldots x_i)|.$$

Since our theory depends on analysing the local stability of a smooth ball for a fitness function, stability over this class is necessary for convergence on more general objectives. We show that once $\sigma_d$ decays slower than the critical rate, divergence already occurs within this subclass, establishing the sharpness of the rate.

**Theorem D.7** (Exact divergence for cubic objectives). *Let*

$$f(x) = a^\top x + \frac{1}{2}x^\top Bx + \frac{1}{6}C(x,x,x),$$

*where $a \in \mathbb{R}^d$, $B \in \mathbb{R}^{d \times d}$ is symmetric, and $C : \mathbb{R}^d \times \mathbb{R}^d \times \mathbb{R}^d \to \mathbb{R}$ is a fully symmetric 3-linear map. Let Assumption D.6 hold, then:*

$$\nabla J(\mu) = \nabla f(\mu) + \frac{\sigma_d^2}{2}\mathbb{E}_{v \sim \mathcal{N}(0,I_d)}[C(v,v,\cdot)].$$

*Moreover:*

$$\left\|\mathbb{E}_{v \sim \mathcal{N}(0,I_d)}[C(v,v,\cdot)]\right\| = \Theta(d) \implies \|\nabla J(\mu) - \nabla f(\mu)\| = \Theta(\sigma_d^2 d).$$

*Proof.* We start by taking derivatives of $f(x)$:

$$\nabla f(x) = a + Bx + \frac{1}{2}C(x,x,\cdot).$$

Substituting this into the definition of $\nabla J(\mu)$ and using Equation (24):

$$
\begin{aligned}
\nabla J(\mu) &= \frac{1}{\sigma_d} \mathbb{E}_{v \sim \mathcal{N}(0,I_d)} \left[ v f(\mu + \sigma_d v) \right], \\
&= \mathbb{E}_{v \sim \mathcal{N}(0,I_d)} \left[ \nabla f(\mu + \sigma_d v) \right], \\
&= \mathbb{E}_{v \sim \mathcal{N}(0,I_d)} \left[ a + B(\mu + \sigma_d v) + \frac{1}{2} C(\mu + \sigma_d v, \mu + \sigma_d v, \cdot) \right], \\
&= a + B\mu + \sigma_d B \underbrace{\mathbb{E}_{v \sim \mathcal{N}(0,I_d)}[v]}_{=0} + \frac{1}{2} \mathbb{E}_{v \sim \mathcal{N}(0,I_d)} \left[ C(\mu + \sigma_d v, \mu + \sigma_d v, \cdot) \right], \\
&= a + B\mu + \frac{1}{2} \mathbb{E}_{v \sim \mathcal{N}(0,I_d)} \left[ C(\mu, \mu, \cdot) + \sigma_d C(v, \mu, \cdot) + \sigma_d C(\mu, v, \cdot) + \sigma_d^2 C(v, v, \cdot) \right], \\
&= \underbrace{a + B\mu + \frac{1}{2} C(\mu, \mu, \cdot)}_{=\nabla f(\mu)} + \frac{1}{2} \mathbb{E}_{v \sim \mathcal{N}(0,I_d)} \left[ 2\sigma_d C(v, \mu, \cdot) + \sigma_d^2 C(v, v, \cdot) \right], \\
&= \nabla f(\mu) + \sigma_d \mathbb{E}_{v \sim \mathcal{N}(0,I_d)} \left[ C(v, \mu, \cdot) \right] + \frac{\sigma_d^2}{2} \mathbb{E}_{v \sim \mathcal{N}(0,I_d)} \left[ C(v, v, \cdot) \right],
\end{aligned}
$$

where we have used the fact $C(v, \mu, \cdot) = C(\mu, v, \cdot)$ by definition of the symmetry of C. As $C(v, \mu, \cdot)$ is linear in $v$, its expectation under $\mathcal{N}(0, I_d)$ is zero, hence:

$$
\nabla J(\mu) = \nabla f(\mu) + \frac{\sigma_d^2}{2} \mathbb{E}_{v \sim \mathcal{N}(0,I_d)} \left[ C(v, v, \cdot) \right],
$$

proving our first result. Now, it follows that $\|C(v, v, \cdot)\| \le \|C\| \|v\|^2$ and from Assumption D.6 $\|C\| = \mathcal{O}(1)$, hence:

$$
\begin{aligned}
\left\| \mathbb{E}_{v \sim \mathcal{N}(0,I_d)} \left[ C(v, v, \cdot) \right] \right\| &\le \|C\| \mathbb{E}_{v \sim \mathcal{N}(0,I_d)} \left[ \|v\|^2 \right], \\
&= \mathcal{O}(\mathbb{E}_{v \sim \mathcal{N}(0,I_d)} \left[ \|v\|^2 \right])
\end{aligned}
$$

Now as $v$ is unit Gaussian: $\mathbb{E}_{v \sim \mathcal{N}(0,I_d)} \left[ \|v\|^2 \right] = d$, hence:

$$
\left\| \mathbb{E}_{v \sim \mathcal{N}(0,I_d)} \left[ C(v, v, \cdot) \right] \right\| = \mathcal{O}(d).
$$

We now show that the bound is tight. Consider the function $f(x) = u^\top x \|x\|^2$ for $u^\top = \frac{1}{\sqrt{d}}[1, \ldots 1]$. The factor of $\frac{1}{\sqrt{d}}$ ensures that the gradient of the function $\nabla_x f(x) = \mathcal{O}(1)$. We can write $\|x\|^2$ as the tensor contraction:

$$
\|x\|^2 = I_d(x, x),
$$

where $I_d$ is the identity matrix and:

$$
u^\top(x) = u(x),
$$

hence we write $f(x)$ as a tensor contraction as:

$$
f(x) = C(x, x, x),
$$

where $C := \mathrm{Sym}(u \otimes I_d)$. Using this function:

$$
\begin{aligned}
C(v, v, \cdot) &= \nabla_v(u^\top v \|v\|^2) \\
&= u\|v\|^2 + 2vu^\top v, \\
\implies \mathbb{E}_{v \sim \mathcal{N}(0,I_d)} \left[ C(v, v, \cdot) \right] &= u \underbrace{\mathbb{E}_{v \sim \mathcal{N}(0,I_d)} \left[ \|v\|^2 \right]}_{=d} + 2 \underbrace{\mathbb{E}_{v \sim \mathcal{N}(0,I_d)} \left[ vv^\top \right]}_{=I_d} u, \\
&= u(d + 2),
\end{aligned}
$$

hence $\|\mathbb{E}_{v \sim \mathcal{N}(0, I_d)} [C(v, v, \cdot)]\| = d + 2$, achieving the upper bound rate of $\mathcal{O}(d)$ which implies:

$$\|\mathbb{E}_{v \sim \mathcal{N}(0, I_d)} [C(v, v, \cdot)]\| = \Theta(d).$$

Our final result follows immediately:

$$\|\nabla J(\mu) - \nabla f(\mu)\| = \frac{(\sigma_d)^2}{2} \|\mathbb{E}_{v \sim \mathcal{N}(0, I_d)} [C(v, v, \cdot)]\| = \Theta(\sigma_d^2 d). \quad \square$$

Theorem D.7 establishes three distinct regimes for ES in high dimensions, separated by the critical rate $\sigma_d = C d^{-1/2}$. For rates slower than the critical scaling, $d^{-1/2} = o(\sigma_d)$, there exist smooth objective functions for which the ES update diverges due to the cubic contribution $\|\mathbb{E}_{v \sim \mathcal{N}(0, I_d)} [C(v, v, \cdot)]\| \to \infty$, implying $\|\nabla J(\mu)\| \to \infty$. At the critical rate $\sigma_d \asymp d^{-1/2}$, the cubic contribution is $\Theta(1)$ and balances the bounded term $\nabla f(\mu)$, so higher-order effects persist in the update. For faster rates, $\sigma_d = o(d^{-1/2})$, the cubic contribution vanishes asymptotically, leaving only the first-order derivative.

### D.3. EGGROLL Linearisation

We now study the effect of EGGROLL in high dimensions. Like in Section C, we introduce the notation $v = \text{vec}(E)$ to denote the vectorisation of the low-rank matrix $E = \frac{1}{\sqrt{r}} A B^\top$ and work in vector space. The EGGROLL vector update $v$ can thus be written as sum of independent variables:

$$v = \sum_{i=1}^{r} \frac{1}{\sqrt{r}} u_i$$

with:

$$u_i = \text{vec}\left(a_i b_i^\top\right),$$

where recall $a_i$ and $b_i$ are the $i$th column vectors of $A$ and $B$. The key difference in notation between this section and Section C is we don't explicitly condition on $r$ (i.e. we don't write $v^r$) as our analysis treats $r$ as a constant and $d = mn$ as the asymptotic quantity. As before, we overload notation slightly by using $\mu$ for both the matrix mean $\mu$ and its vectorisation $\text{vec}(\mu)$, however its meaning is always clear from context. Using Equation (7), we can convert between results as:

$$\|\text{vec}(\hat{g}_{\text{LR}}) - \nabla f(\mu)\| = \|\hat{g}_{\text{LR}} - \nabla_M f(\mu)\|_F,$$
$$\|\text{vec}(\hat{g}_{\text{LR}}) - \nabla J(\mu)\| = \|\hat{g}_{\text{LR}} - \nabla_\mu J(\mu)\|_F.$$

A crucial difference between our analysis of full rank Gaussian ES and EGGROLL is that we cannot apply a key identity known as Stein's lemma (Stein, 1972) because $\frac{1}{\sigma}\mathbb{E}[Ef(M = \mu + \sigma E)] \neq \mathbb{E}[\nabla_M f(M = \mu + \sigma E)]$ using the low-rank approximation. To account for this, we introduce the following slightly stricter local continuity assumption on the fitness function:

**Assumption D.8** (EGGROLL Locally Continuous Fitness). With probability 1 with respect to the random initialisation of $\mu$, assume there exists a ball $B_\rho(\mu) := \{h \mid \|h - \mu\| < \rho\}$ of fixed radius $\rho > 0$ where $f(h)$ is $C^2$-continuous for all $h \in B_\rho(\mu)$ and $\|\nabla^2 f(\mu)\|$ be polynomial bounded in $d$. Within this ball, let $\nabla^2 f(h)$ be Lipschitz continuous, i.e. $\|\nabla^2 f(h) - \nabla^2 f(h')\| \leq L_d \|h - h'\|$ for all $h, h' \in B_\rho(\mu)$.

To extend our analysis, we need to ensure that all polynomial moments of $P(v)$ are finite and grow at most polynomially in the dimension $d = mn$. In particular, such tail bounds are sufficient to dominate polynomial error terms in our analysis. To introduce sub-Gaussian variables, we follow the exposition of Vershynin (2018) and results therein. A random variable $x_i \in \mathbb{R}$ is sub-Gaussian if there exists some finite constant $C > 0$ such that for all $t > 0$:

$$\mathbb{P}(|x_i| > t) \leq 2 \exp(-C t^2),$$

meaning their their tails decay like Gaussians. This is equivalent to any of the following three properties holding (Vershynin, 2018, 2.6.1): There exist constants $C_1, C_2, C_3 > 0$ that differ at most by an absolute constant factor such that:

$$(\mathbb{E}[|x_i|^p])^{\frac{1}{p}} \leq C_1 \sqrt{p}, \quad \forall p \geq 1,$$
$$\mathbb{E}\left[\exp\left(\frac{x_i^2}{C_2^2}\right)\right] \leq 2,$$

and if $\mathbb{E}[x_i] = 0$:

$$\mathbb{E}\left[\exp(\lambda x_i)\right] \leq \exp(C_3^2 \lambda^2), \quad \forall \lambda \in \mathbb{R}.$$

A random vector $x \in \mathbb{R}^d$ is sub-Gaussian if all one-dimensional marginals of $x$ are sub-Gaussian, i.e. $x^\top u$ is sub-Gaussian for all $u \in \mathbb{R}^d$. The sub-Gaussian norm is defined as:

$$\|x\|_{\psi_2} := \inf_K \left\{ K \,\middle|\, \mathbb{E}\left[\exp\left(u^\top(x - \mathbb{E}[x])\right)\right] \leq \exp\left(\frac{K^2\|u\|^2}{2}\right), \quad \forall u \in \mathbb{R}^d \right\}.$$

which returns the smallest universal sub-Gaussian constant for all marginals. This motivates our final assumption:

**Assumption D.9** (Sub-Gaussian)**.** In addition to Assumption 4.1, assume that $p_0(\cdot)$ generates variables that have sub-Gaussian tails, i.e. for $x_i \sim p_0(x_i)$

$$\mathbb{P}(|x_i| > t) \leq 2\exp(-Ct^2),$$

for some $C < \infty$ that does not depend on $d, m, n$.

The assumption is trivially satisfied for Gaussian distributions $a \sim \mathcal{N}(0, I_m)$ and $b \sim \mathcal{N}(0, I_n)$, and holds more generally, for example for bounded distributions, uniform distributions and generalised Gaussian distributions with shape parameter greater than two. This flexibility is particularly relevant for 'finite' models Section 6.2, where heavier-shouldered distributions may be preferred over the Gaussian.

A key property of sub-Gaussian vectors that we use in our proofs is the sub-Gaussian concentration inequality for the Euclidean norm (Vershynin, 2018, Theorem 3.1.1), which states that for if $x$ is a sub-Gaussian vector with $\mathbb{E}[x_i^2] = 1$ and $K = \|x\|_{\psi_2}$, there exists an absolute constant $C > 0$ such that for all $t \geq 0$,

$$\mathbb{P}\left(\big|\|x\| - \sqrt{m}\big| \geq t\right) \leq 2\exp\left(-\frac{Ct^2}{K^4}\right). \tag{28}$$

We also use a weaker form of control, that replaces the Gaussian-like tail decay with an exponential decay, but all other properties are defined similarly. In this paper, we use the definition that a variable $x$ is known as sub-exponential if there exists a $K > 0$ such that for all $t \geq 0$:

$$\mathbb{P}\left(|x| \geq t\right) \leq 2\exp\left(-\frac{t}{K}\right).$$

Our first result derives a bound on the expected value of the norms $\|a\|^i$ and $\|b\|^i$:

**Lemma D.10.** *Let Assumption D.9 hold. Let $P(a)$ denote the distribution over columns of $A$ and $P(b)$ denote the distribution over columns of $B$. Then:*

$$\mathbb{E}_{a \sim P(a)}[\|a\|^i] = \mathcal{O}(m^{\frac{i}{2}}), \quad \mathbb{E}_{b \sim P(b)}[\|b\|^i] = \mathcal{O}(n^{\frac{i}{2}}).$$

*Proof.* It suffices to prove $\mathbb{E}_{a \sim P(a)}[\|a\|^i] = \mathcal{O}(m^{\frac{i}{2}})$ as $\mathbb{E}_{b \sim P(b)}[\|b\|^i] = \mathcal{O}(n^{\frac{i}{2}})$ follows automatically from the same assumptions. We start by using the 'layer cake' representation of the expectation Lieb & Loss (2010 - 2010, Theorem 1.13):

$$\mathbb{E}_{a \sim P(a)}[\|a\|^i] = i \int_0^\infty t^{i-1} \mathbb{P}(\|a\| > t) dt.$$

Let $t_m = C\sqrt{m}$ for any $C > 1$. We split the integral into two regions:

$$i \int_0^\infty t^{i-1} \mathbb{P}(\|a\| > t) dt = \int_0^{t_m} it^{i-1} \mathbb{P}(\|a\| > t) dt + \int_{t_m}^\infty it^{i-1} \mathbb{P}(\|a\| > t) dt.$$

For the first integral:

$$\int_0^{t_m} it^{i-1} \mathbb{P}(\|a\| > t) dt \leq \int_0^{t_m} it^{i-1} dt,$$
$$= (t_m)^i,$$
$$= C^i m^{\frac{i}{2}}.$$

For the second integral, we wish to bound $\mathbb{P}(\|a\| > t)$ for the region $t \geq t_m = C\sqrt{m}$. Setting $t' = t - \sqrt{m} > 0$, the assumption $C > 1$ implies $t' \geq 0$ in this region, hence

$$\mathbb{P}(\|a\| > t) = \mathbb{P}(\|a\| - \sqrt{m} > t') \leq \mathbb{P}(|\|a\| - \sqrt{m}| > t').$$

We bound this using the sub-Gaussian concentration inequality from Equation (28). Under Assumption D.9, $a$ is a sub-Gaussian vector with $\|x\|_{\psi_2} \leq \infty$, hence there exists an absolute constant $C' > 0$ such that for all $t' \geq 0$,

$$\mathbb{P}\left(|\|a\| - \sqrt{m}| \geq t'\right) \leq 2\exp\left(-C't'^2\right).$$

This implies:

$$\mathbb{P}\left(\|a\| \geq t\right) \leq 2\exp\left(-C'(t - \sqrt{m})^2\right),$$

for all $t \geq t_m$. Substituting yields:

$$\int_{t_m}^{\infty} it^{i-1}\mathbb{P}(\|a\| > t)dt \leq \int_{t_m}^{\infty} it^{i-1}\exp\left(-C'(t - \sqrt{m})^2\right)dt.$$

Let $x = t - \sqrt{m} \implies dt = dx$:

$$\int_{t_m}^{\infty} it^{i-1}\mathbb{P}(\|a\| > t)dt \leq \int_{\sqrt{m}(C-1)}^{\infty} i(x + \sqrt{m})^{i-1}\exp\left(-C'x^2\right)dx.$$

Now, $\sqrt{m} \leq \frac{x}{C-1}$ for all $x \geq \sqrt{m}(C-1)$, hence:

$$
\begin{aligned}
\int_{t_m}^{\infty} it^{i-1}\mathbb{P}(\|a\| > t)dt &\leq \int_{\sqrt{m}(C-1)}^{\infty} ix^{i-1}\left(1 + \frac{1}{C-1}\right)^{i-1}\exp\left(-C'x^2\right)dx, \\
&= i\left(1 + \frac{1}{C-1}\right)^{i-1}\int_{\sqrt{m}(C-1)}^{\infty} x^{i-1}\exp\left(-C'x^2\right)dx, \\
&\leq i\left(1 + \frac{1}{C-1}\right)^{i-1}\int_{0}^{\infty} x^{i-1}\exp\left(-C'x^2\right)dx, \\
&\leq i\left(1 + \frac{1}{C-1}\right)^{i-1}\frac{1}{2}(C')^{-i/2}\Gamma\left(\frac{i}{2}\right), \\
&= \mathcal{O}(1).
\end{aligned}
$$

Combining the two bounds yields:

$$\mathbb{E}_{a\sim P(a)}[\|a\|^i] = \mathcal{O}(m^{\frac{i}{2}}),$$

as required. $\qquad\square$

Using this result, we now bound the whole vector $v = \frac{1}{\sqrt{r}}\sum_{i=1}^{r}\text{vec}(a_ib_i^\top)$

**Lemma D.11.** *Let $i \geq 1$. Under Assumption D.9:*

$$\mathbb{E}_{v\sim P(v)}\left[\|v\|^i\right] = \mathcal{O}\left((rmn)^{\frac{i}{2}}\right)$$

*Proof.* For any vectors $a, b$:

$$\|\text{vec}(ab^\top)\| = \sqrt{\sum_{j=1}^{m}\sum_{k=1}^{n}(a_jb_k)^2} = \sqrt{\sum_{j=1}^{m}a_j^2\sum_{k=1}^{n}b_k^2} = \|a\|\|b\|,$$

$$\implies \|\text{vec}(ab^\top)\|^i = \|a\|^i\|b\|^i.$$

Applying Lemma D.10 under Assumption D.9 for each summand of $v = \frac{1}{\sqrt{r}} \sum_{l=1}^{r} \text{vec}(a_l b_l^\top)$:

$$\mathbb{E}_{v \sim P(v)} \left[ \|\text{vec}(a_l b_l^\top)\|^i \right] = \mathbb{E}_{a_l \sim P(a_l)} \left[ \|a_l\|^i \right] \mathbb{E}_{b_l \sim p(b_l)} \left[ \|b_l\|^i \right],$$
$$= \mathcal{O}\left( (mn)^{\frac{i}{2}} \right).$$

Applying the triangle inequality:

$$\mathbb{E}_{v \sim P(v)} \left[ \|v\|^i \right] = \mathbb{E}_{v \sim P(v)} \left[ \left\| \frac{1}{\sqrt{r}} \sum_{l=1}^{r} \text{vec}(a_l b_l^\top) \right\|^i \right],$$
$$\leq \mathbb{E}_{v \sim P(v)} \left[ \left( \frac{1}{\sqrt{r}} \sum_{l=1}^{r} \|\text{vec}(a_l b_l^\top)\| \right)^i \right],$$
$$= r^{\frac{i}{2}} \mathbb{E}_{v \sim P(v)} \left[ \left( \frac{1}{r} \sum_{l=1}^{r} \|\text{vec}(a_l b_l^\top)\| \right)^i \right].$$

Now, as $i \geq 1$, we can apply Jensen's inequality:

$$\left( \frac{1}{r} \sum_{l=1}^{r} \|\text{vec}(a_l b_l^\top)\| \right)^i \leq \frac{1}{r} \sum_{l=1}^{r} \|\text{vec}(a_l b_l^\top)\|^i,$$

yielding:

$$\mathbb{E}_{v \sim P(v)} \left[ \|v\|^i \right] \leq r^{\left(\frac{i}{2}-1\right)} \sum_{l=1}^{r} \mathbb{E}_{v \sim P(v)} \left[ \|\text{vec}(a_l b_l^\top)\|^i \right] = r^{\frac{i}{2}} \mathcal{O}\left( (mn)^{\frac{i}{2}} \right) = \mathcal{O}\left( (rmn)^{\frac{i}{2}} \right). \qquad \square$$

Our proof borrows techniques used to prove linearisation of the ES update in Section D.1 by bounding the tail probability of any polynomial under the low-rank distribution outside of the ball $B_\rho(\mu)$. To apply the concentration inequality that would generalise Lemma D.4, we show that $v$ has an exponentially decaying tail:

**Lemma D.12** (Exponential Tail Bound). *Let $r < \infty$ and Assumption D.9 hold. Then all elements of $v$ are sub-exponential and for $\sqrt{d}\sigma_d = o(1)$ there exists some constant $C > 0$ such that:*

$$\mathbb{P}(\|\sigma_d v\| \geq \rho) \leq 2d \exp\left( -C \frac{\rho}{\sqrt{d}\sigma_d} \right).$$

*Proof.* In matrix form:

$$E = \frac{1}{\sqrt{r}} \sum_{i=1}^{r} a_i b_i^\top.$$

The elements of $E$ are thus:

$$E_{j,k} = \frac{1}{\sqrt{r}} \sum_{i=1}^{r} a_{ij} b_{ik}.$$

As $a_{ij}$ and $b_{ik}$ are independent sub-Gaussian random variables with zero mean, it follows from Vershynin (2018, Lemma 2.8.6) that their product $a_{ij}b_{ik}$ is a zero-mean sub-exponential variable with a uniform norm $\|a_{ij}b_{ik}\|_{\psi_1} < \infty$. Finally, a finite sum of sub-exponential variables is sub-exponential (Wainwright, 2019, Eq. (2.18)) with a uniform norm, so all elements of $E$ and hence $v = \text{vec}(E)$ are sub-exponential and zero-mean with a uniform $\psi_1$-norm $K < \infty$.

We now bound $\mathbb{P}(\|\sigma_d v\| \geq \rho) = \mathbb{P}(\|v\| \geq \frac{\rho}{\sigma_d})$. For the vector $v$, it follows for $t \geq 0$:

$$\|v\| \geq t \implies \max_j |v_j| \geq \frac{t}{\sqrt{d}}.$$

This is easily proven via the contrapositive: if $\max_j |v_j| < \frac{t}{\sqrt{d}}$ then

$$\|v\|^2 = \sum_{j=1}^{d} v_j^2 < d\frac{t^2}{d} = t^2,$$

implying $\|v\| < t$. This means for $t \geq 0$:

$$\mathbb{P}(\|v\| \geq t) \leq \mathbb{P}\left(\max_j |v_j| \geq \frac{t}{\sqrt{d}}\right),$$

$$\leq \sum_{j=1}^{d} \mathbb{P}\left(|v_j| \geq \frac{t}{\sqrt{d}}\right). \tag{29}$$

As $v_j$ is a sub-exponential variable with finite uniform sub-exponential norm, by definition (Vershynin, 2018, Proposition 2.8.1) there exists a finite $K$ such that for all $j$:

$$\mathbb{P}\left(|v_j| \geq \frac{t}{\sqrt{d}}\right) \leq 2\exp\left(-\frac{t}{\sqrt{d}K}\right).$$

Applying to Equation (29) yields:

$$\mathbb{P}(\|v\| \geq t) \leq 2d\exp\left(-\frac{t}{\sqrt{d}K}\right).$$

Now, using $t = \frac{\rho}{\sigma_d}$ and $C = \frac{1}{K}$ yields:

$$\mathbb{P}(\|\sigma_d v\| \geq \rho) \leq 2d\exp\left(-C\frac{\rho}{\sqrt{d}\sigma_d}\right). \qquad \square$$

We now use these results to assemble into our key polynomial tail bound:

**Lemma D.13** (EGGROLL Polynomial Tail Bounds). *Let Assumption D.9 hold. Let $g(x)$ be polynomial bounded as:*

$$\|g(x)\| \leq C(1 + \|x\|^p),$$

*for some finite polynomial of order $p$ and constant $C > 0$. Consider the ball $B_\rho(\mu) := \{h\,|\,\|h - \mu\| < \rho\}$. Let $\{\mu + \sigma_d v \in B_\rho(\mu)\} = \{\|\sigma_d v\| < \rho\}$ denote the event that a mutation lies outside the ball. Assume $\sigma_d = o(d^{-1/2})$. Then for some constant $K > 0$ independent of $d$:*

$$\left\|\mathbb{E}_{v \sim P(v)}\left[g(\mu + \sigma_d v)\mathbb{1}(A_d)\right]\right\| = \mathcal{O}\left(\sqrt{d}\exp\left(-K\frac{\rho}{\sqrt{d}\sigma_d}\right)\right),$$

*and in particular the right-hand side is $o(1)$ as $d \to \infty$.*

*Proof.* Let

$$A_d := \{\mu + \sigma_d v \in B_\rho(\mu)\}$$

and denote $\mathbb{P}(A_d) := \mathbb{E}_{v \sim P(v)}[\mathbb{1}(A_d)]$. Our proof proceeds as in Lemma D.4 to obtain:

$$\left\|\mathbb{E}_{v \sim P(v)}\left[g(\mu + \sigma_d v)\mathbb{1}(A_d)\right]\right\| \leq C'\mathbb{P}(A_d) + C''\sigma_d^p\mathbb{E}_{v \sim P(v)}\left[\|v\|^p\mathbb{1}(A_d)\right].$$

where $C' = C(1 + 2^{p-1}\|\mu\|^p)$ and $C'' = C2^{p-1}$ are constant in $d$. Applying the Cauchy–Schwarz inequality to the second expectation gives:

$$\mathbb{E}_{v \sim P(v)}[\|v\|^p\mathbb{1}(A_d)] \leq \sqrt{\mathbb{E}_{v \sim P(v)}[\|v\|^{2p}]} \cdot \sqrt{\mathbb{P}(A_d)}.$$

Applying Lemma D.11 with fixed $r$ and $d = mn$:

$$\sqrt{\mathbb{E}_{v \sim P(v)}[\|v\|^{2p}]} = \mathcal{O}\left(d^{\frac{p}{2}}\right).$$

Now, $\mathbb{P}(A_d) = \mathbb{P}(\|\sigma_d v\| \geq \rho)$. From Lemma D.12, there exists some $K > 0$ such that:

$$\mathbb{P}(\|\sigma_d v\| \geq \rho) \leq 2d \exp\left(-K\frac{\rho}{\sqrt{d}\sigma_d}\right),$$

$$\implies \sqrt{\mathbb{P}(A_d)} = \mathcal{O}\left(\sqrt{d}\exp\left(-K\frac{\rho}{\sqrt{d}\sigma_d}\right)\right),$$

where we have absorbed the factor of $\frac{1}{2}$ into $K$, hence:

$$\mathbb{E}_{v \sim P(v)}[\|v\|^p \mathbb{1}(A_d)] = \mathcal{O}\left(d^{\frac{p+1}{2}}\exp\left(-K\frac{\rho}{\sqrt{d}\sigma_d}\right)\right).$$

Now, as $\sqrt{d}\sigma_d = o(1)$, $\sigma_d^p d^{\frac{p}{2}} = o(1)$, hence:

$$\sigma_d^p \mathbb{E}_{v \sim P(v)}[\|v\|^p \mathbb{1}(A_d)] = \mathcal{O}\left(\sqrt{d}\exp\left(-K\frac{\rho}{\sqrt{d}\sigma_d}\right)\right).$$

Applying our bounds yields our desired result:

$$\left\|\mathbb{E}_{v \sim P(v)}\left[g(\mu + \sigma_d v)\mathbb{1}(A_d)\right]\right\| = \mathcal{O}\left(\sqrt{d}\exp\left(-K\frac{\rho}{\sqrt{d}\sigma_d}\right)\right) = o(1).$$

where the $o(1)$ bound follows from the fact that the exponential factor dominates $\sqrt{d}$ and $\sqrt{d}\sigma_d = o(1)$. $\qquad\square$

**Theorem D.14** (EGGROLL Convergence to Linearity). *Let Assumptions D.2, D.3, D.8 and D.9 hold and $\sigma_d = o(d^{-1/2})$ and $L_d(\sigma_d d)^2 = o(1)$. Then there exists some $K > 0$ such that:*

$$\|\text{vec}(\hat{g}_{LR}) - \nabla f(\mu)\| = \mathcal{O}\left(L_d(\sigma_d d)^2\right) + \mathcal{O}\left(\frac{\sqrt{d}}{\sigma_d^2}\exp\left(-K\frac{\rho}{\sqrt{d}\sigma_d}\right)\right) = o(1),$$

$$\|\text{vec}(\hat{g}_{LR}) - \nabla J(\mu)\| = \mathcal{O}\left(\sigma_d\sqrt{d}\cdot\left(1 + L_d\sigma_d d^{\frac{3}{2}}\right)\right) = o(1).$$

*almost surely with respect to the distribution over $\mu$.*

*Proof.* We start with the definition of the vectorised EGGROLL update:

$$\text{vec}(\hat{g}_{\text{LR}}) - \nabla f(\mu) = \frac{1}{\sigma_d}\mathbb{E}_{v \sim P(v)}\left[v \cdot f(\mu + \sigma_d v)\right] - \nabla f(\mu),$$

$$= \frac{1}{\sigma_d}\mathbb{E}_{v \sim P(v)}\left[v \cdot f(\mu + \sigma_d v)\right] - \frac{1}{\sigma_d}\underbrace{\mathbb{E}_{v \sim P(v)}[v]}_{=0} \cdot f(\mu) - \underbrace{\mathbb{E}_{v \sim P(v)}[vv^\top]}_{I_d}\nabla f(\mu)$$

$$+ \frac{1}{2\sigma_d}\underbrace{\mathbb{E}_{v \sim P(v)}[\sigma_d^2 vv^\top \nabla^2 f(\mu)v]}_{=0},$$

$$= \frac{1}{\sigma_d}\mathbb{E}_{v \sim P(v)}\left[v \cdot \underbrace{\left(f(\mu + \sigma_d v) - f(\mu) - \sigma_d v^\top \nabla f(\mu) + \frac{\sigma_d^2}{2}v^\top \nabla^2 f(\mu)v\right)}_{:=T_d(v)}\right],$$

$$= \frac{1}{\sigma_d}\mathbb{E}_{v \sim P(v)}\left[v \cdot T_d(v)\right],$$

where we have used the fact that the expectation of an odd function under a symmetric, zero mean distribution is always zero, and $P(v)$ satisfies this under Assumption D.9, hence $\mathbb{E}_{v \sim P(v)}[vv^\top \nabla^2 f(\mu)v] = 0$, and $\mathbb{E}_{v \sim P(v)}[vv^\top] = I_d$ from

**Lemma C.1.** Consider the ball $B_\rho(\mu) := \{h|\|h - \mu\| < \rho\}$. We now split the integral into two regions, the first within the ball and the second outside:

$$\frac{1}{\sigma_d}\mathbb{E}_{v \sim P(v)}\left[v \cdot (f(\mu + \sigma_d v) - f(\mu))\right] = \underbrace{\frac{1}{\sigma_d}\mathbb{E}_{v \sim P(v)}\left[v \cdot T_d(v)\mathbb{1}(\|\sigma_d v\| < \rho)\right]}_{:=I_{\text{loc}}}$$

$$+ \underbrace{\frac{1}{\sigma_d}\mathbb{E}_{v \sim P(v)}\left[v \cdot T_d(v)\mathbb{1}(\|\sigma_d v\| \geq \rho)\right]}_{:=I_{\text{tail}}}.$$

Consider the region inside the ball:

$$\|I_{\text{loc}}\| = \frac{1}{\sigma_d}\left\|\mathbb{E}_{v \sim P(v)}\left[v \cdot T_d(v)\mathbb{1}(\|\sigma_d v\| < \rho)\right]\right\|,$$

$$\leq \frac{1}{\sigma_d}\mathbb{E}_{v \sim P(v)}\left[\|v\|\,|T_d(v)|\,\mathbb{1}(\|\sigma_d v\| < \rho)\right]. \tag{30}$$

Within this region, $f(\mu + \sigma_d v)$ is $C^2$ continuous under Assumption D.8. We can thus write $f(\mu + \sigma_d v)$ using a first-order Taylor expansion about $\mu$ with a Hessian (second order derivative) remainder within the ball:

$$f(\mu + \sigma_d v) = f(\mu) + \sigma_d \nabla f(\mu)^\top v + \sigma_d^2 v^\top \left(\int_0^1 (t-1)\nabla^2 f(\mu + t\sigma_d v)dt\right)v,$$

$$\implies T_d(v) = \sigma_d^2 v^\top \left(\int_0^1 (t-1)\nabla^2 f(\mu + t\sigma_d v)dt\right)v + \frac{\sigma_d^2}{2}v^\top \nabla^2 f(\mu)v,$$

$$= \sigma_d^2 v^\top \left(\int_0^1 (t-1)(\nabla^2 f(\mu + t\sigma_d v) - \nabla^2 f(\mu))dt\right)v.$$

Applying the Lipschitz bound on the Hessian from Assumption D.8:

$$|T_d(v)| \leq \sigma_d^2 \|v\|^2 \left\|\int_0^1 (t-1)(\nabla^2 f(\mu + t\sigma_d v) - \nabla^2 f(\mu))dt\right\|,$$

$$\leq \sigma_d^2 \|v\|^2 \int_0^1 (t-1)\left\|\nabla^2 f(\mu + t\sigma_d v) - \nabla^2 f(\mu)\right\| dt,$$

$$\leq \sigma_d^2 \|v\|^2 \int_0^1 (t-1)L_d \|t\sigma_d v\| dt,$$

$$= \sigma_d^3 \|v\|^3 L_d \left|\int_0^1 (t-1)t\,dt\right|,$$

$$= \frac{L_d}{6}\sigma_d^3 \|v\|^3.$$

Using this to bound Equation (30):

$$\|I_{\text{loc}}\| \leq \frac{L_d}{6}\sigma_d^2 \mathbb{E}_{v \sim P(v)}\left[\|v\|^4 \mathbb{1}(\|\sigma_d v\| < \rho)\right],$$

$$\leq \frac{L_d}{6}\sigma_d^2 \mathbb{E}_{v \sim P(v)}\left[\|v\|^4\right].$$

Now, (for fixed $r$) we apply the identity $\mathbb{E}_{v \sim P(v)}\left[\|v\|^4\right] = \mathcal{O}\left((mn)^2\right)$ with $mn = d$ from Lemma D.11:

$$\|I_{\text{loc}}\| = \mathcal{O}(L_d(\sigma_d d)^2).$$

We now bound the tail region outside the ball:

$$I_{\text{tail}} = \frac{1}{\sigma_d} \mathbb{E}_{v \sim P(v)} \left[ v \cdot T_d(v) \mathbb{1} (\|\sigma_d v\| \geq \rho) \right],$$

$$\leq \frac{1}{\sigma_d} \mathbb{E}_{v \sim P(v)} \left[ \|v\| |T_d(v)| \mathbb{1} (\|\sigma_d v\| \geq \rho) \right],$$

$$= \frac{1}{\sigma_d^2} \mathbb{E}_{v \sim P(v)} \left[ \|\sigma_d v\| |T_d(v)| \mathbb{1} (\|\sigma_d v\| \geq \rho) \right].$$

Now under Assumptions D.2, D.3 and D.8, $f(\mu + \sigma_d v)$ is polynomial bounded, $\|\nabla f(\mu)\| = \mathcal{O}(1)$ and $\|\nabla^2 f(\mu)\|$ is polynomial bounded hence there exists some finite constant $C > 0$ and finite polynomial order $p$ such that:

$$\|\sigma_d v\| |T_d(v)| \leq C(1 + \|\mu + \sigma_d v\|^p).$$

We thus apply Lemma D.13:

$$\frac{1}{\sigma_d^2} \mathbb{E}_{v \sim P(v)} \left[ \|\sigma_d v\| |T_d(v)| \mathbb{1} (\|\sigma_d v\| \geq \rho) \right] = \mathcal{O} \left( \frac{\sqrt{d}}{\sigma_d^2} \exp \left( -K \frac{\rho}{\sqrt{d} \sigma_d} \right) \right),$$

$$= \mathcal{O} \left( \frac{d\sqrt{d}}{d\sigma_d^2} \exp \left( -K \frac{\rho}{\sqrt{d} \sigma_d} \right) \right).$$

Now, as $\sigma_d \sqrt{d} = o(1)$, the exponential term dominates the prefactor $\frac{d\sqrt{d}}{d\sigma_d^2}$, we conclude:

$$\frac{1}{\sigma_d^2} \mathbb{E}_{v \sim P(v)} \left[ \|\sigma_d v\| |T_d(v)| \mathbb{1} (\|\sigma_d v\| \geq \rho) \right] = o(1)$$

Our final result follows from:

$$\|\text{vec}(\hat{g}_{\text{LR}}) - \nabla J(\mu)\| = \|\text{vec}(\hat{g}_{\text{LR}}) - \nabla f(\mu) + \nabla f(\mu) - \nabla J(\mu)\|,$$

$$\leq \|\text{vec}(\hat{g}_{\text{LR}}) - \nabla f(\mu)\| + \|\nabla f(\mu) - \nabla J(\mu)\|.$$

We have already shown $\|\text{vec}(\hat{g}_{\text{LR}}) - \nabla f(\mu)\| = o(1)$ and under the assumptions for this theorem, Theorem D.5 holds and so $\|\nabla f(\mu) - \nabla_\mu J(\mu)\| = o(1)$. $\qquad \square$

## E. EGGROLL Speed

All timings were done on a single GPU on a GH200 (equivalent to a single H100) for a linear model with dimension 8192 in bfloat16, allowing a maximum batch size of 1024. For the graph in Figure 1a, we pre-generate the noises instead of integrating the noise generation into the forward pass.

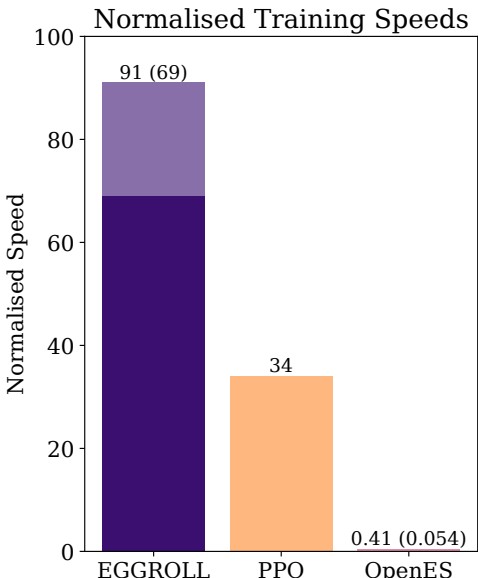

*Figure 7.* Relative speed of EGGROLL, when including jax noise regeneration.

In Figure 7, we consider the impact of regenerating noises on-the-fly using jax PRNG. The darker area and value in parenthesis for EGGROLL and OpenES indicate the speed when regenerating noises on-the-fly, while the full bar indicates the speed when the noises are already generated.

We regenerate noises on the fly in our primary jax codebase, but pre-generating the EGGROLL perturbations beforehand is also a practical possibility since low-rank perturbations only require a small amount of memory, proportional to the square root of the size of the original parameter matrices.

# F. Arithmetic Intensity Analysis

In this section, we derive the arithmetic intensity of standard batched inference, Gaussian matrix ES, and EGGROLL. We calculate arithmetic intensity as the number of operations divided by the total number of bytes read from or written to. For context, for the (b)float16 datatype on an H100 GPU, there are approximately 1000 teraFLOPS of compute (without sparsity) and 3.35 TB/s of GPU memory bandwidth, meaning that the roofline threshold is approximately 300 ops/byte, defined as the minimum for computation needed for it to be the bottleneck instead of memory movement.

In the following subsections, we are considering a single linear layer with mean parameter $\mu \in \mathbb{R}^{d_{out} \times d_{in}}$ and a batch of inputs $x \in \mathbb{R}^{B \times d_{in}}$. All operations occur with a precision of $s$ bytes per element.

## F.1. Arithmetic Intensity of Standard Batched Inference

In standard batched inference, we wish to simply calculate $x\mu^T$. The total bytes read as input are $B \times d_{in} \times s$ (for $x$) and $d_{out} \times d_{in} \times s$ (for $\mu$), and the total bytes written as output are $B \times d_{out} \times s$. The total number of operations are $B \times d_{in} \times d_{out} \times 2$ since matrix multiplication requires both multiplications and additions for each element of $x$ across all of $d_{out}$. Therefore, the arithmetic intensity is:

$$\frac{B \times d_{in} \times d_{out} \times 2}{B \times d_{in} \times s + B \times d_{out} \times s + d_{out} \times d_{in} \times s}.$$

When $s = 2$ (for (b)float16) and $d_{out} = d_{in} = m$, the arithmetic intensity simplifies to

$$\frac{Bm}{2B + m}.$$

The batch size needed to achieve a desired arithmetic intensity of $A$ is derived as follows:

$$Bm = 2AB + Am$$
$$Bm - 2AB = Am$$
$$B = \frac{Am}{m - 2A}$$

Therefore, achieving an arithmetic intensity of 300 ops/byte with $m = 8192$ requires a minimum batch size of 324.

## F.2. Arithmetic Intensity of Gaussian Matrix ES

In Gaussian matrix ES, we assume access to pre-generated perturbations of shape $\mathbb{R}^{B \times d_{out} \times d_{in}}$. The total bytes read as input are $B \times d_{in} \times s$ (for $x$) and $B \times d_{out} \times d_{in} \times s$ (for $\mu$), and the total bytes written as output are $B \times d_{out} \times s$. Otherwise, the total number of operations is identical to standard batched inference, giving us an arithmetic intensity of

$$\frac{B \times d_{in} \times d_{out} \times 2}{B \times d_{in} \times s + B \times d_{out} \times s + B \times d_{out} \times d_{in} \times s} = \frac{d_{in} \times d_{out} \times 2}{d_{in} \times s + d_{out} \times s + d_{out} \times d_{in} \times s}.$$

When $s = 2$ (for (b)float16) and $d_{out} = d_{in} = m$, the arithmetic intensity simplifies to

$$\frac{m}{2 + m}.$$

This means that arithmetic intensity is always strictly less than 1, regardless of batch size or dimensionality. The common way to increase arithmetic intensity is to bring it closer to standard batched inference, reusing the same perturbation across multiple inputs. For instance, when $m = 8192$, achieving an arithmetic intensity of 300 ops/byte requires that each perturbation is reused at least 324 times, and smaller values of $m$ need to be reused even more often.

### F.3. Arithmetic Intensity of EGGROLL

For EGGROLL, we assume access to the pre-generated decomposed perturbations $A \in \mathbb{R}^{B \times d_{out} \times r}$ and $B \in \mathbb{R}^{B \times d_{in} \times r}$. Therefore, the bytes read as pure input are $B \times d_{in} \times s + B \times (d_{in} + d_{out}) \times r \times s + d_{out} \times d_{in} \times s$ and the bytes written as pure output are $B \times d_{out} \times s$. However, the efficient low-rank perturbation calculation requires writing and reading an intermediate matrix of shape $B \times r$, so the total bytes read are

$$(B \times d_{in} + B \times (d_{in} + d_{out} + 2) \times r + d_{out} \times d_{in} + B \times d_{out}) \times s.$$

The total number of operations includes the amount for standard batch inference, $B \times d_{in} \times d_{out} \times 2$, along with the rank-$r$ perturbations, $B \times (d_{in} + d_{out}) \times r \times 2$, and the final sum between the main calculation and perturbation $B \times d_{out}$. Therefore, the arithmetic intensity is

$$\frac{B \times d_{in} \times d_{out} \times 2 + B \times (d_{in} + d_{out}) \times r \times 2 + B \times d_{out}}{(B \times d_{in} + B \times (d_{in} + d_{out} + 2) \times r + d_{out} \times d_{in} + B \times d_{out}) \times s}.$$

When $s = 2$ (for (b)float16) and $d_{out} = d_{in} = m$, the arithmetic intensity simplifies to

$$\frac{Bm + 2Br + \frac{B}{2}}{B + Br(2 + \frac{2}{m}) + m + B}$$

$$= \frac{m + 2r + \frac{1}{2}}{2 + r(2 + \frac{2}{m}) + \frac{m}{B}}.$$

The batch size needed to achieve a desired arithmetic intensity of $A$ is derived as follows:

$$2A + rA(2 + \frac{2}{m}) + \frac{Am}{B} = m + 2r + \frac{1}{2}$$

$$\frac{Am}{B} = m + 2r + \frac{1}{2} - 2A - rA(2 + \frac{2}{m})$$

$$B = \frac{Am}{m - 2A + 2r + \frac{1}{2} - rA(2 + \frac{2}{m})}$$

Note that the only difference with the critical batch size of standard batched inference is the additional $2r + \frac{1}{2} - rA(2 + \frac{2}{m})$ in the denominator. Therefore, achieving an arithmetic intensity of 300 ops/byte with $m = 8192$ and $r = 1$ requires a minimum batch size of 352, compared to 324 for standard batched inference. This means that EGGROLL can saturate compute with unique perturbations per input, unlike Gaussian matrix ES.

Note that there is an overhead of $Bm(4r + 1)$ flops relative to standard batched inference, resulting in an additional compute rate of $\frac{Bm(4r+1)}{2Bm^2} = \frac{4r+1}{2m}$, which is effectively negligible for large enough matrices.

# G. EGG Architecture

In the following section, we detail the design of our EGG model, which follows the high-level structure of modern pre-layernorm decoder-only language models, but replaces self-attention with a modified minGRU and standard layernorms with a custom variant to enable pure integer training. See Algorithm 2 for an overview of the forward pass of the EGG architecture.

---

**Algorithm 2** EGG forward pass

---

**Require:** Input token $t \in \mathbb{U}_8$, input state $s \in \mathbb{I}_8^{l \times D}$, network parameters $\theta$
**Ensure:** Output vector $y \in \mathbb{I}_8^D$ and output state $s' \in \mathbb{I}_8^{l \times D}$
  $s' \leftarrow \mathbb{I}_8^{l \times D}$ initialised to 0
  $y \leftarrow \text{EMBED}(\theta_{\text{emb}}, t)$
  **for** $i \in \{0, \ldots, l-1\}$ **do**
    $y', s'_i \leftarrow \text{GRU}(\theta_{\text{gru},i}, \text{LN}(\theta_{\text{ln1},i}, y), s_i)$
    $y \leftarrow \mathbb{I}_8(\mathbb{I}_{32}(y') + \mathbb{I}_{32}(y))$
    $y' \leftarrow \text{MLP}(\theta_{\text{mlp},i}, \text{LN}(\theta_{\text{ln2},i}, y))$
    $y \leftarrow \mathbb{I}_8(\mathbb{I}_{32}(y') + \mathbb{I}_{32}(y))$
  **end for**
  $y \leftarrow \text{LN}(\theta_{\text{lnout},i}, y) @ \theta_{\text{head}}^T$

---

## G.1. Motivation

Since EGGROLL does not rely on gradients, we can explicitly design a language model architecture to be efficient and hardware-friendly at inference time. In particular, we design EGG under the following constraints to emphasise the flexibility of EGGROLL:

**Pure Integer Training:** On H100 systems, int8 is the fastest datatype and int8 matrix multiplication with int32 accumulation is the fastest tensor core operation. Furthermore, integer datatypes are much simpler to implement in hardware, providing massive energy savings for high-throughput systems (Horowitz, 2014). Therefore, we keep all weights in int8 and all activations in integer formats, *never* casting to floating point at any point during training. This stands in contrast to the standard approach for language model quantisation through "quantisation aware training" with backpropagation, where floating point activations are still necessary (Wang et al., 2023).

**Nonlinear RNN:** Modern language models use sequence-parallel architectures like Transformers and SSMs, since they enable stable gradients without backpropagation through time. However, most of these architectures cannot handle simple state tracking (Merrill et al., 2024), whereas classic recurrent networks like LSTMs and GRUs can do so with a single layer. Since EGGROLL does not require backpropagation through time, we can train on unbounded sequence lengths (Li et al., 2023) with nonlinear RNNs of broader complexity classes. Specifically, we develop a variant of the minGRU model (Heck & Salem, 2017) that performs all operations in integer formats.

**Removal of all Activation Functions:** Inspired by Foerster (2017), we remove all activation functions, like the rectified linear unit and hyperbolic tangent, due to the nonlinearity present in the int8 datatype. Specifically, the saturated addition of int8 values provides sufficient nonlinearity due to the implicit clipping of values to the int8 dynamic range, which evolution strategies can exploit.

## G.2. Notation and Operations

We use the constant $l \in \mathbb{Z}^+$ to denote the number of layers of the model and $D = 4^d$ as the hidden dimension of the model, where $d \in \mathbb{Z}^+$.

We use $\mathbb{I}_n$ to denote an $n$-bit signed integer and $\mathbb{U}_n$ to denote an $n$-bit unsigned integer. We denote casting vector $\vec{x}$ to format $\mathbb{I}_n$ as $\mathbb{I}_n(\vec{x})$, which implicitly includes clipping to the bounds of the datatype. To ensure symmetry between positive and negative values of each datatype, we consider the value $-2^{n-1}$ to be invalid for datatype $\mathbb{I}_n$; for instance, for 8-bit signed integers we only allows value from -127 to 127.

We use the following operations:

- $\vec{x}@M$ indicating scaled vector-matrix multiplication of $\mathbb{I}_8^n \times \mathbb{I}_8^{n,m} \to \mathbb{I}_8^m$, corresponding to int8 tensor core multiplication with int32 accumulation and scaling. The details of this operation are described in Section G.4.

- $a \cdot b$ indicates dot product with int32 accumulation, $\mathbb{I}_8^n \times \mathbb{I}_8^n \to \mathbb{I}_{32}$, and $a \odot b$ indicates the Hadamard (elementwise) product.

- Standard integer operations: $+$ for addition, $-$ for subtraction, and $\odot$ for element-wise multiplication.

- $|x|$ indicates taking the element-wise absolute value of $x$, $\mathbb{I}^n \to \mathbb{I}^n$.

- $\text{sign}(x)$ indicates taking the element-wise sign of $x$, giving 1 for positive values, -1 for negative values, and 0 for zero.

- $\text{sum}(x)$ indicates taking the sum of all elements in $x$ (casting to $\mathbb{I}_{32}$ to prevent overflow): $\mathbb{I}^n \to \mathbb{I}_{32}$.

- $x \gg n$ indicates an elementwise bitwise right shift by $n$, which is typically equivalent to $2^{-n}x$. Similarly, $x \ll n$ indicates a bitwise left shift by $n$, which is typically equivalent to $2^n x$.

- Square-bracket indexing. For instance $M[x, y]$ extracts the element at index $x$ in axis 0 and index $y$ in axis 1, following the zero-based indexing convention.

### G.3. Parameter Initialisation
The standard initialisation for matrix parameters in our model is rounding 16 times a sample from the standard normal, and casting to $\mathbb{I}_8$. This can be precomputed on a CPU since this is only done once at the start of training.

The egg model has the following parameters (where an additional subscript of $i$ indicates that there is a version of this parameter for each layer of the model):

- $\theta_{\text{emb}} \in \mathbb{I}_8^{256 \times D}$, following standard initialisation.

- $\theta_{\text{head}} \in \mathbb{I}_8^{256 \times D}$, following standard initialisation.

- $\theta_{\text{lnout}} \in \mathbb{I}_8^D$, initialised to 16 for each element.

- $\theta_{\text{ln1},i}, \theta_{\text{ln2},i} \in \mathbb{I}_8^D$, initialised to 16 for each element

- $\theta_{\text{mlp},i,1} \in \mathbb{I}_8^{4D \times D}$ and $\theta_{\text{mlp},i,2} \in \mathbb{I}_8^{D \times 4D}$, following standard initialisation.

- $\theta_{\text{GRU},i,[\text{Wf,Uf,Wh,Uh}]} \in \mathbb{I}_8^{D \times D}$, following standard initialisation.

- $\theta_{\text{GRU},i,[\text{bfm bh}]} \in \mathbb{I}_8^D$, initialised to 0 for each element.

In total there are $513D + l(4D + 12D^2)$ parameters in the model.

### G.4. Matrix Multiplication
Tensor cores in GPUs are able to calculate fast vector-matrix multiplications with int32 accumulation as $xM \in \mathbb{I}_{32}^m$ where $x \in \mathbb{I}_8^n$ and $M \in \mathbb{I}_8^{n \times m}$. For our purposes, we define $x@M$ as a scaled multiplication:

$$x@M := \mathbb{I}_8 \left( \frac{xM}{16\sqrt{n}} \right).$$

Note that when $n = 4^d$, the division operation just becomes a right-shift by $4 + d$, which is fast to calculate.

We choose this scaled matrix multiplication because we initialise $M$ to 16 times standard normal samples for each element, so dividing by $16\sqrt{n}$ preserves the magnitude of $x$ for the output. In particular, if all elements of $x$ and $M$ are drawn from independently from the standard normal distribution multiplied by 16, the central limit theorem tells us that the expected value per element of the output will be $256\sqrt{n}$, so dividing by $16\sqrt{n}$ preserves the standard deviation of 16.

### G.5. Embedding
Our embedding function takes as input an embedding matrix $\theta_{\text{emb}} \in \mathbb{I}_8^{256 \times D}$ and an input token $t \in \mathbb{U}_8$, and simply outputs the vector corresponding to that token: $\theta_{\text{emb}}[t] \in \mathbb{I}_8^D$.

### G.6. Layer Normalisation (LN)

Our layer normalisation operation involves multiplying our input $x \in \mathbb{I}_8^D$ with a weight $\theta_{\text{ln}} \in \mathbb{I}_8^D$ before dividing by the mean absolute value of $x$.

We decide to divide by the mean absolute value of the input instead of the more more common root-mean-squared since square roots are expensive on integers. Note that the $L1$ norm after dividing the input by the mean absolute value (when using real numbers) is $D$ instead of 1, which we intentionally choose to preserve more bits of information given the limited range of $\mathbb{I}_8$.

We calculate the mean absolute value of input $x$ as:

$$x_{\text{mav}} = \mathbb{I}_8(\text{sum}(|x|) \gg (2d)),$$

Note that we can safely cast the mean absolute value to an $\mathbb{I}_8$ without overflow given the properties of the mean of a set, though we lose precision due to truncating the fractional component.

The output of layernorm is calculated as:

$$\text{DIVIDE}(\mathbb{I}_{16}(x) \odot \mathbb{I}_{16}(\theta_{\text{ln}}), x_{\text{mav}}).$$

Since division is an expensive operation, we precompute it using a lookup table. Note that the product of two $\mathbb{I}_8$ values will always remain in the dynamic range of $\mathbb{I}_{16}$, so our lookup table will be of shape $2^{16} \times 2^8$.

### G.7. MLP

Each MLP block consists of two weight parameters: $\theta_1 \in \mathbb{I}_8^{4D \times D}$ and $\theta_2 \in \mathbb{I}_8^{D \times 4D}$. Given an input $x \in \mathbb{I}_8^D$, we calculate the output as:

$$(x @ \theta_1^T) @ \theta_2^T.$$

Note that we do not use an activation function, because the @ operation is already nonlinear due to the saturated conversion from $\mathbb{I}_{32}$ to $\mathbb{I}_8$

### G.8. GRU

Each GRU block accepts an input vector and state $x, s \in \mathbb{I}_8^D$ consists of 6 weight parameters: $\theta_{\text{Wf}}, \theta_{\text{Uf}}, \theta_{\text{Wh}}, \theta_{\text{Uh}} \in \mathbb{I}_8^{D \times D}$ and $\theta_{\text{bf}}, \theta_{\text{bh}} \in \mathbb{I}_8^D$.

Using these weight matrices, we calculate the following vectors:

$$f = \sigma(\mathbb{I}_8(\mathbb{I}_{32}(x @ \theta_{\text{Wf}}^T) + \mathbb{I}_{32}(s @ \theta_{\text{Uf}}^T) + \mathbb{I}_{32}(\theta_{\text{bf}}))),$$
$$\hat{f} = \mathbb{I}_8(((\mathbb{I}_{32}(f) + 127) \odot \mathbb{I}_{32}(s)) \gg 8),$$
$$\hat{h} = \phi(\mathbb{I}_8(\mathbb{I}_{32}(x @ \theta_{\text{Wh}}^T) + \mathbb{I}_{32}(\hat{f} @ \theta_{\text{Uh}}^T) + \mathbb{I}_{32}(\theta_{\text{bh}}))),$$
$$h = s + \mathbb{I}_8(((\mathbb{I}_{32}(f) + 127) \odot (\mathbb{I}_{32}(\hat{h}) - \mathbb{I}_{32}(s))) \gg 8),$$

where $h$ is the output and the new hidden state. In the typical GRU, $\sigma$ stands for the sigmoid function while $\phi$ stands for the hyperbolic tangent, but we find that setting these as identity operations is sufficient due to the nonlinearity already present in the clipped addition. One can view this clipped addition operation as scaled and shifted version of the "hard" tanh and sigmoid operators.

To explain why we perform these operations, we can analyse this relative to the original GRU. The $f$ vector for the standard GRU has all elements between 0 and 1 due to the sigmoid, but our elements are between -127 and 127. Therefore, to calculate $\hat{f}$ (which is typically just $f \odot s$), we first add 127 to $f$, getting the range between 0 and 254 before multiplying by $s$ before bit-shifting right by 8 again to bring our values back to the $\mathbb{I}_8$ dynamic range. We apply similar logic to calculate the final $h$, which is typically just $h = s + f \odot (\hat{h} - s)$ but needs to be rescaled to keep the int8 dynamic range.

### G.9. Fitness Calculation in Integer Types

The "fitness" used in language model pretraining is the log-likelihood of correctly generating the next token, treating the outputs of the language model as logits (unnormalised log probabilities). If $t' \in \mathbb{U}_8$ is the next token to predict and $y \in \mathbb{I}_8^{256}$ are the logits, we can calculate the log likelihood as follows:

$$y' = \mathbb{I}_{32}(y) + 128,$$
$$o = y'[t'] - \text{LOG2}[\text{sum}(\text{EXP2}[y'])],$$

where $o$ is the loss for one token. We implement EXP2 and LOG2 as lookup tables, where

$$\text{EXP2}[i] = 16 \times 2^{i/16},$$
$$\text{LOG2}[i] = 16 \times \log_2(i/16).$$

Note that each element in EXP2 for any $\mathbb{U}_8$ input requires at most 20 bits, so the sum of exponents across all possible choices is at most 28 bits, meaning we have to precompute LOG2 for $2^{28}$ values.

## H. EGG Pretraining with Integer EGGROLL

The core ideas of EGGROLL still apply in this integer-based training setting, but we have to make some modifications to ensure it only uses integer operations.

### H.1. Adding EGGROLL Perturbations

For parameter $\theta \in \mathbb{I}_8^{m \times n}$ that represents a matrix multiplication, we first sample rank-1 perturbation vectors for each index in the batch: $A \in \mathbb{I}_8^m$ and $B \in \mathbb{I}_8^n$. We sample these vectors from the standard random normal multiplied by 16 and rounded to the nearest $\mathbb{I}_8$ (clipping if necessary). To prevent the use of floating-point arithmetic on the accelerator, we pre-generate a large matrix of these random values, randomly indexing into it to get the perturbation vectors.

Given an input $x \in \mathbb{I}_8^n$, instead of calculating $x@\theta^T$, we calculate

$$\mathbb{I}_8 \left( \frac{x\theta^T + ((x \cdot B)\mathbb{I}_{32}(A) \gg (4 + \hat{\sigma}))}{16\sqrt{n}} \right).$$

The value of $\hat{\sigma}$ is a hyperparameter, related to the $\sigma$ in the main paper as $\sigma = 2^{-\hat{\sigma}}$. Note that the batched forward pass remains efficient since it still simply performs a batched vector-vector dot product in int8 (with int32 accumulate) and a batched vector-scalar product in int32.

We apply this same logic to the embedding matrix, since we can interpret $\theta[t]$ as one_hot$(t)\theta$ and still apply our rank-1 updates in that context. In practice, this means replacing $x \cdot B$ with $B[t]$.

### H.2. Fitness Shaping

We employ a simple fitness shaping scheme based on antithetical pairs. Specifically, given raw fitnesses $s^+, s^-$, for the positive and negative sample of the antithetical pair respectively, the transformed fitness for the noise is:

$$\text{sign}(s^+ - s^-),$$

Note that the only possible values for the fitness after shaping are $\{-1, 0, 1\}$.

### H.3. Parameter Update

For parameter $\theta \in \mathbb{I}_8^{m \times n}$ that represents a matrix multiplication (or embedding vector), suppose the sampled batch of rank-1 perturbation vectors are $A \in \mathbb{I}_8^{N \times m}$ and $B \in \mathbb{I}_8^{N \times n}$, and let the fitnesses after shaping be $F \in \mathbb{I}_8^N$. Then we calculate an intermediate value $E \in \mathbb{I}_{32}^{m \times n}$ as:

$$E = (\text{diag}(F)A)^T B.$$

We use $E$ to determine if each element of $\theta$ should be increased or decreased. In particular, when the absolute value of $E$ is above a pre-specified threshold we move $\theta$ by one discrete bin in the direction of the sign of $E$. Since there are only 255 unique values for each element in $\mathbb{I}_8$, restricting updates to single bins improves stability without compromising the ability for a parameter to get to any other value with relatively few updates. In particular, we have a real-valued hyperparameter, $\alpha \in (0, 1)$ such that the threshold equals

$$\mathbb{I}_{32}\left(16 \times \Phi^{-1}\left(\frac{1-\alpha}{2}\right)\right) \times 16\sqrt{N},$$

where $\Phi$ is the normal cumulative distribution function. Note that this threshold can be precalculated on a CPU. We observe that $\alpha$ approximately equals the fraction of parameters that are updated at each step.

We currently do not incorporate any momentum or other optimiser states, but this remains critical future work to improve the speed of convergence for pure integer training.

Across model sizes and population size, we find that setting $\hat{\sigma}$ to 4 and letting $\alpha$ decay over training steps as $\frac{1}{.015t+1}$ gives consistently strong results.

## I. EGG Ablations

In our main experiments, we use a fixed data batch size of 16 sequences for population sizes 2 and powers of 4 ranging from 4 to $4^{10} = 1048576$. In this section, we vary the batch size by powers of 4, ranging from 4 to $4^5 = 1024$, while varying population size by powers of 4 from 16 to 1048576. When the batch size, $b$ is greater than half of the population size, $N$, we give each antithetical pair $\frac{2b}{N}$ sequences, functionally giving a cleaner fitness signal to each member of the population. This also means that the number of parallel "inferences" required is $\max(2b, N)$.

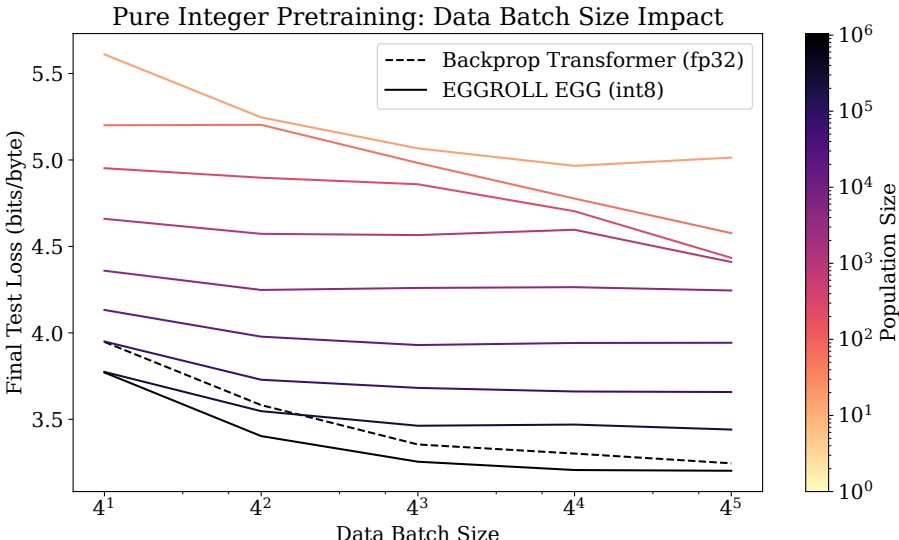

*Figure 8.* Test loss curves when varying data batch size and population size.

In Figure 8, we observe that the final test loss for each population size is relatively constant beyond a specific data batch size threshold. At the top right of the figure, we observe a decrease in loss for small population sizes after $b > \frac{N}{2}$, which is an artifact of the increased compute usage necessary to use the full data batch. Ignoring this artifact, the minimum batch size for near-optimal performance at a given population size $N$ appears to be $\frac{N}{4^6}$. We see that large population sizes need larger data batches for improved performance, since a batch size of 4 results in nearly identical performance for population sizes $4^9 = 262144$ and $4^{10} = 1048576$, but this diverges as data batch size increases.

# J. Distributed EGGROLL Framework

To facilitate the large-scale experiments, where we scale population sizes beyond 1M, we develop a lightweight distributed training framework designed to minimise network overhead.

## J.1. Base-3 Fitness Packing and Bandwidth Efficiency

A key bottleneck in distributed training is the communication of gradients or results. We address this via a custom base-3 packing scheme for fitness vectors. Since workers evaluate perturbations in antithetic pairs, the raw signal is discretised into ternary values $\{+1, 0, -1\}$. These are mapped to $\{0, 1, 2\}$ and packed five at a time into a single byte:

$$byte = \sum_{i=0}^{4} v_i \cdot 3^i$$

This yields an effective bitrate of $1.6$ bits per value (near the $\log_2 3 \approx 1.585$ theoretical limit). Consequently, the network payload per chunk is approximately $52 + \text{chunk\_size}/10$ bytes, rendering bandwidth usage independent of model size.

## J.2. System Architecture

The system employs a Coordinator-Worker topology. The Coordinator maintains the global state and assigns population chunks to Workers. Workers calculate fitness on GPU, apply signal shaping (chunk mean filtering, adaptive thresholding), and return only the packed ternary fitness, minimising traffic significantly compared to standard gradient transmission.

# K. Fine-tuning of Integer Quantised Models

## K.1. Quantisation Procedure

To maximise population throughput and reduce device memory during EGGROLL fine-tuning, we represent the large matrix-multiplication parameters of RWKV in an int8 weight format while keeping non-matmul parameters (e.g., small biases / bookkeeping tensors) in floating point, bf16. Following Jacob et al. (2017), for each weight matrix $W \in \mathbb{R}^{d_{\text{in}} \times d_{\text{out}}}$, we use symmetric per-channel int8 quantisation with an absmax scale. For each output channel we first compute:

$$s_i = \max\left( \frac{\max_j |W_{i,j}|}{127}, \epsilon \right),$$

where $\epsilon$ is some small scalar. Then, we store each $s_i$ in bf16, and quantise weights as

$$Q_{i,j} = \text{clip}\left( \text{round}\left( \frac{W_{i,j}}{s_i} \right), -127, 127 \right) \in \mathbb{I}_8.$$

Every matrix parameter is stored as a dictionary containing the quantised weight matrix $Q$, the scale parameters per channel $\{s_i\} \forall i \in 1, \ldots, d_{\text{out}}$ and an input scale factor $s_x$ in bf16 precision. At runtime, the forward pass is computed by scaling the input vector by $s_x$ and the quantised matrix $Q$ with the scales per channel, $[s_1, \ldots, s_{d_{\text{out}}}]$,

$$x_{n+1} = (x_n \odot s_x)^T (W \odot [s_1, \ldots, s_{d_{\text{out}}}]).$$

## K.2. Integrating integer-quantised EGGROLL with Adam

EGGROLL performs black-box (ES) optimisation directly over the parameter representation used in the forward pass, including integer quantised weights. We integrate this with the Adam optimiser (Kingma & Ba, 2014) by maintaining Adam's moment estimates in bf16, while enforcing that all quantised tensors remain on the int8 lattice.

**ES gradients.** EGGROLL estimates gradients via antithetic ES perturbations and score-weighted averaging. This yields a bf16 gradient estimate for: (i) floating-point parameters (when present), (ii) quantised matrix parameters via a low-rank perturbation pathway, and (iii) scale parameters $\{s_i\} \forall i \in 1, \ldots, d_{\text{out}}$ and $s_x$ via explicit scale perturbations. We then pass these gradients to Adam (Optax), which produces an update tensor $u$ for each parameter leaf.

**Adam updates for int8 tensors (discretised).** For integer parameters (notably int8), Adam produces a real-valued proposal $u$ (stored in bf16). Since the parameter itself must remain int8, we convert this proposal into a sparse unit-step update using a normalised thresholding rule. Let $Q \in \mathbb{Z}_8^{m \times n}$ be an int8 tensor and $u \in \mathbb{R}^{m \times n}$ be Adam's proposed update. We compute a per-tensor z-score normalisation

$$z = \frac{u - \mu(u)}{\sigma(u) + 10^{-8}},$$

then apply a threshold $\tau$ to form the integer step

$$\Delta = \text{sign}(z) \cdot \mathbb{1}\{|z| \geq \tau\} \in \{-1, 0, +1\}^{m \times n}.$$

Finally we update by unit increments and clip to the valid int8 range:

$$Q \leftarrow \text{clip}(Q + \Delta, -127, 127).$$

Intuitively, Adam supplies a magnitude- and history-aware proposal, while the discretisation enforces the integer constraint and yields a stable, sparse update pattern (only entries with sufficiently large normalised updates are modified).

**Memory considerations.** We store Adam's optimiser state (moments) in bf16 for all array-valued leaves to reduce memory footprint, while keeping scalar bookkeeping in full precision. This keeps the dominant memory cost of optimisation close to that of the parameters themselves, which is particularly important when fine-tuning large models with large ES populations.

**Model distillation.** We distil a non-quantised model into the quantised RWKV-7 model by matching the two distributions in teacher forced examples from GSM8k. More specifically, the fitness for a given set of parameters, $\mu_i$, is computed as follows:

$$f_{\mu_i}(x_{1:T}) = \sum_{t=1}^{T} \text{KL}\left(p_t || q_t(\cdot; \mu_i)\right),$$

where $x_{1:T}$ is a subsequence of tokens taken from the solutions of GSM8K and $\text{KL}\left(p_t || q_t(\cdot; \mu_i)\right)$ is the Kullback-Leibler divergence between the distribution of the non-quantised model, $p_t$, and the distribution of the quantised model $q_t$ over the vocabulary at token $t$.

## L. Fine-tuning Pretrained Transformer LLMs with Verifiable Rewards

This section compares EGGROLL to standard RL from Verifiable Rewards (RLVR). We first describe our experimental results, before including details of the infrastructure used to run these experiments.

### L.1. Results

Here we demonstrate that EGGROLL can be used to fine-tune pre-trained LLMs on verifiable rewards. We use the vLLM library (Kwon et al., 2023) for efficient, high-throughput inference. More infrastructure detail is given in Section L.2.

We fine-tune Qwen3-4B separately on DeepScaleR (Agentica Organization et al., 2025) (40k math questions) and ORZ57K (Hu et al., 2025) (57k math questions), and report the results for both fine-tuned models. As in standard RLVR, the model generates a chain-of-thought (CoT) followed by a final answer. Fitness is then simply calculated by extracting the final answer and comparing it to the ground truth answer (Shao et al., 2025). We evaluate performance on Math500 (Hendrycks et al., 2021), OlympiadBench (He et al., 2024), AIME24 (Zhang & Math-AI, 2024), AIME25 (Zhang & Math-AI, 2025), and MinervaMath (Lewkowycz et al., 2022).

We compare the performance with the untrained model and with GRPO, trained to match hardware and effective wall-clock training time. In particular, we ran EGGROLL with 256 population members per GPU, and then scaled the number of GPUs for larger population sizes. Our results with an EGGROLL population size of 256 are obtained by training on a single GH200 GPU for 9 hours, hence showing EGGROLL can match GRPO within the same compute budget. Furthermore, we found that GRPO consistently plateaued, and hence these results are effectively the strongest GRPO performances for this setup (context length, initial model, sampling parameters), even with unlimited compute budget. By contrast, EGGROLL continued to improve with more compute to significantly outperform GRPO, as we show with population sizes 4k and 16k.

| Parameter | EGGROLL | GRPO |
|---|---|---|
| Population size | 256, 4096, 16384 | n/a |
| Sigma | 1e-3 | n/a |
| Learning rate | 2e-4, (1e-3) | 1e-6, (1e-7, 5e-7) |
| Max response length | 8192, (4096) | 8192, (4096) |
| Prompt batch size | 16 | 32 |
| Temperature | 0.0 | 0.6 |
| Sampling top-$p$ | n/a | 0.95 |
| Sampling top-$k$ | n/a | 20 |
| Samples per prompt | 1 | 8 |
| LoRA rank | 1 | n/a |
| LoRA reuse steps | 4 | n/a |
| Normalise fitnesses with std | Yes | No, (Yes) |
| Scale lr in grad | Yes | n/a |
| Training steps | 3 | 200 |

*Table 1.* Hyperparameters for the EGGROLL and GRPO training runs, used for both Qwen3-4B and Qwen3-4B-Base. Parentheses indicate alternate settings used in some runs. For Qwen3-4B, we used a maximum response length of 8192, whereas for Qwen3-4B-Base, we used a maximum response length of 4096. When using population size 256, we use 1e-3 as the learning rate; for larger population sizes, the learning rate is set to 2e-4.

| Method | Dataset | Math500 | Minerva | OlympiadBench | AIME24 | AIME25 | Average |
|---|---|---|---|---|---|---|---|
| No Training | n/a | 82.0 | 29.8 | 33.3 | 21.1 | 18.9 | 37.0 |
| GRPO (ours) | DeepScaler | 89.4 | 34.7 | 37.8 | 32.1 | **43.6** | 47.5 |
| EGGROLL (256 population) | DeepScaler | 85.9 | 42.2 | 43.0 | 44.5 | 33.6 | 49.8 |
| EGGROLL (4k population) | DeepScaler | 89.8 | 43.0 | 46.1 | **55.5** | 23.4 | 51.6 |
| EGGROLL (16k population) | DeepScaler | **91.4** | **44.5** | **51.6** | 54.7 | 41.4 | **56.7** |
| GRPO (ours) | ORZ | 89.0 | 35.5 | 38.0 | 26.1 | 32.4 | 44.2 |
| EGGROLL (256 population) | ORZ | **93.0** | 41.4 | 45.3 | 35.9 | 26.6 | 48.4 |
| EGGROLL (4k population) | ORZ | 91.4 | **44.5** | 49.2 | 56.2 | 38.3 | 55.9 |
| EGGROLL (16k population) | ORZ | 90.6 | 43.8 | **52.3** | **64.8** | **41.4** | **58.6** |

*Table 2.* Evaluation results for Qwen3-4B across maths benchmarks. All runs use 8192 maximum token budget. Bold indicates the best result within each dataset group. For EGGROLL, to ensure a conservative compute-matched comparison, we report results after the third training step, even if we observed even better results at further training steps. For GRPO, we report results at step 200 of training, since we found that performance beyond this point plateaues or decreases.

The hyperparameters used for these runs can be found in Table 1, the vLLM settings can be found in Table 5 and the results can be found in Table 2.

We also fine-tune the Qwen3-4B-Base model on DeepScaleR, using EGGROLL and GRPO, and again we observe that across benchmarks EGGROLL outperforms GRPO. The results can be found in Table 3.

| Method | Math500 | Minerva | OlympiadBench | AIME24 | AIME25 | Average |
|---|---|---|---|---|---|---|
| No Training | 50.2 | 28.0 | 24.4 | 10.0 | 7.4 | 24.0 |
| GRPO (ours) | 72.8 | 30.0 | 30.7 | 7.6 | 10.0 | 30.2 |
| EGGROLL (16k population) | **83.6** | **31.3** | **35.2** | **18.0** | **19.5** | **37.5** |

*Table 3.* Evaluation results for Qwen3-4B-Base fine-tuned on DeepScaleR, using a 4096 maximum token budget. Bold indicates the best result for each benchmark.

Lastly, since EGGROLL can be used to optimise non-differentiable objectives, we evaluate its performance when optimising directly for pass@k. While zero-shot (pass@1) is a differentiable objective, the pass@k objective is not, as it depends on multiple samples from the model. This means it cannot be optimised easily with RL. In Figure 9 we fine-tune the Qwen3-1.7B model on the DeepScaleR dataset with a population size of 256, LoRA rank 1, and $K = 4$ (full hyperparameters can be found in Table 4). We see that EGGROLL successfully optimises both the pass@1 (differentiable) and pass@k (non-differentiable) objectives. In Figure 9 *(right)* we plot the number of distinct answers in 4 samples from the model. We

see that when optimising for pass@k the answer diversity sampled by the model increases over training, whereas when optimising for zero-shot (pass@1) the model collapses towards a single final answer.

| Parameter | Value |
| --- | --- |
| Population size | 256 |
| Sigma | 1e-3 |
| Learning Rate | 1e-3 |
| Max Response Length | 4096 |
| Temperature | 0.7 |
| Samples Per Prompt | 4 |
| LoRA Rank | 1 |
| LoRA Reuse Steps | 4 |

*Table 4.* Hyperparameters for the pass@k fine-tuning experiments in Figure 9.

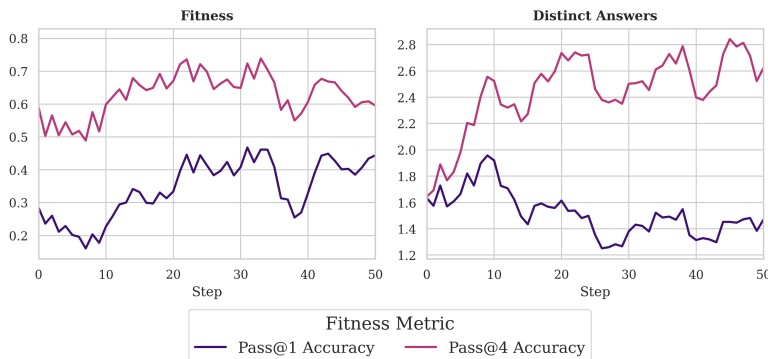

*Figure 9.* Using EGGROLL to optimise non-differentiable objectives. *Left*: Fitness curves comparing training with pass@1 (differentiable) versus pass@k (non-differentiable), where $K = 4$. *Right*: The mean number of unique final answers generated per 4-sample set. We observe that answer diversity increases when optimising for pass@k, and decreases when optimising for zero-shot accuracy (pass@1).

### L.2. Training Infrastructure for Large-Scale Transformer LLMs

EGGROLL facilitates the fine-tuning of transformer-based LLMs at scale. We achieve this by repurposing the vLLM inference engine, leveraging its high-throughput kernel implementations and native support for multi-LoRA serving. The system utilises vLLM's native Tensor Parallelism (TP) to shard the model weights across the GPUs within a node, while cross-node parallelisation is employed for the concurrent evaluation of the LoRA population.

To render ES-based optimisation feasible and efficient across a wide range of model sizes, we implement several critical systems-level optimisations:

**Custom `WorkerExtension` and Sharding-Aware Updates**   By implementing a custom `WorkerExtension`, we effectively convert the vLLM inference engine into a training-capable runtime. This extension allows the optimisation logic to reside within the GPU process space, enabling direct, in-place manipulation of the model's weights. A significant complexity of this integration is vLLM's internal tensor parallelism, which frequently fuses weights (e.g. combining `q_proj`, `k_proj`, and `v_proj` into a single `qkv_proj` tensor). Our update mechanism is explicitly "sharding-aware"; it constructs a dictionary which maps individual LoRA updates to the specific fused slices held by each local GPU rank. This ensures that the global ES update is mathematically consistent across all distributed shards.

**Layer-wise Memory Management**   To prevent out-of-memory (OOM) errors during the update phase, the `WorkerExtension` performs the ES weight application in a streaming, layer-wise fashion. By processing one layer at a time and clearing temporary buffers, the memory overhead of the update remains independent of the total model depth. This allows for the fine-tuning of models of very different sizes with a VRAM footprint barely exceeding that of standard inference.

**Direct GPU-to-GPU Weight Synchronisation**    After computing the ES update on the primary rank, we broadcast the updated parameters to all model instances using NCCL via `PyNcclCommunicator`. This approach bypasses CPU-based communication and instead uses hardware interconnects to transfer weights directly between GPUs, preventing synchronisation from becoming a bottleneck when scaling to more nodes.

**Meta-Device Blueprinting**    To initialise models that exceed the physical RAM of the control node, we employ Meta-Device Initialisation. Using `init_empty_weights` from `accelerate` (Gugger et al., 2022), we instantiate a "meta" version of the model to derive the weight shapes and sharding requirements for the LoRA adapters. This allows the system to generate a complete parameter blueprint for models of arbitrary size without ever allocating the full weight tensors in system memory.

**vLLM Engine Settings**    Throughout the different experiments with vLLM, we use the engine settings from Table 5. These generally allow us to reach a maximum throughput of around 4300 tokens/second, although the throughput varies slightly depending on the stage of training, the size of the model and other hyperparameters, such as the number of LoRAs per engine.

| Parameter | Value |
|---|---|
| Tensor parallel size | 1 |
| Data type | auto |
| Enable prefix caching | True |
| Enforce eager execution | False |
| Enable LoRA | True |
| Max LoRAs | $\lceil \text{population\_size}/\text{num\_engines} \rceil$ |
| GPU memory utilisation | 0.90 |
| Max number of sequences | 512 |
| Max model length | $\max(1024, 512 + \text{max\_tokens})$ |
| Max batched tokens | $\text{prompt\_batch\_size} \times 2048$ |
| Load format | auto |

*Table 5.* vLLM engine configuration parameters used for the results in Table 2 and Table 3

## M. Fine-tuning Time Series Foundation Model: High-Frequency Trading

The preceding experiments demonstrate the effectiveness of EGGROLL on natural language reasoning tasks. We now investigate whether EGGROLL can effectively fine-tune pretrained foundation models on a fundamentally different data modality: structured time series. We focus on high-frequency trading (HFT) for two reasons. First, HFT generates data at an unprecedented scale. The S&P 500 constituents alone produced trillions of tokens of order flow data since 2016, comparable to current natural language corpora. Second, the domain presents a well-defined downstream task (order execution) with a natural reward signal: the realised profit and loss, also known as PnL, making it amenable to fine-tuning via evolution strategies.

Order execution takes place in limit order books (LOBs), which are the mechanism upon which modern financial exchanges operate (Gould et al., 2013; Bouchaud et al., 2018). They allow market participants to submit limit orders that specify the details of intended transactions. Specifically, each limit order contains the order type, direction, price, and quantity. The continuous stream of these orders is known as the order flow. LOBs aggregate the limit orders that have not been matched yet. Unlike natural language, where tokens are purely symbolic, order flow messages comprise both categorical values (e.g., order type, direction) and numerical values (e.g., price, quantity) in which magnitude carries semantic meaning. This structure provides a distinct test of EGGROLL's ability to fine-tune foundation models on time series sequential data.

A central objective in this context is order execution, which consists of buying or selling a specified quantity of an asset within a given time window. The goal is to maximise profit by transacting at favourable prices. In prior reinforcement learning approaches to this problem, the action space is usually simplified (Frey et al., 2023; Mohl et al., 2025; Ning et al., 2021). In contrast, we aim to give the model full flexibility in choosing limit orders, i.e., to freely choose the order type, direction, price, and quantity. We achieve this by tokenising the limit order book messages and providing the model with a token-level action space.

Foundation models have recently been used to generate synthetic order flow (Nagy et al., 2023; Li et al., 2025) and have been shown to replicate realistic market behaviour (Nagy et al., 2025) through next-token prediction. We therefore first pretrain a foundation model on tokenised limit order book messages, and then fine-tune it using EGGROLL for the order execution task. The pretraining follows the approach of Nagy et al. (2023): we employ an S5 model architecture (Smith et al., 2023) that generates next-token probabilities, with cross-entropy as the training loss. The pretraining is conducted on the LOBSTER data set (Huang & Polak, 2011) for eight S&P 500 stocks (GOOG, AAPL, NVDA, AMZN, META, TSLA, MSFT, AMD) over 2022–2025, for approximately 64 billion tokens.

Subsequently, we fine-tune the model using EGGROLL. The training parameters are summarised in Table 6. The task is to execute a sell order of $Q = 30$ shares within a horizon of $T = 10$ steps. In each episode, the LOB is initialised based on a LOB snapshot followed by 10 warm-up background messages. In each step, the population members generate their messages, which are then followed by 50 real background data messages. The orders are executed using the Jax-LOB (Frey et al., 2023) simulator. We perform the fine-tuning on a single fixed GOOG window in January 2026. Following (Galim et al., 2025), we apply LoRA on all projection matrices while freezing SSM parameters and layer norms, and we compare rank 1 versus rank 4. Performance is evaluated using PnL based on the executed prices and the initial mid price. Specifically, for a sell task of total quantity $Q$, the PnL is computed as

$$\sum_{i=1}^{N} q_i p_i - Q\, P_{\text{mid}}^{\text{init}},$$

where $q_i$ and $p_i$ denote the quantity and price of the $i$-th executed trade and $P_{\text{mid}}^{\text{init}}$ is the mid-price at the beginning of the execution window. If the agent does not execute the entire quantity by the end of the episode, an automatic order is submitted at the end of the episode selling the remaining quantity. To improve robustness to outliers, fitness is defined as a rank-based transformation of the PnL. Specifically, for a population of size $M$, the PnL values

$$\mathcal{P} = \{\text{PnL}_1, \ldots, \text{PnL}_M\},$$

are mapped to the interval $[-0.5, 0.5]$, where $\text{rank}(\text{PnL}_i) \in \{0, \ldots, M-1\}$ denotes the rank of the $i$-th individual's PnL:

$$F_i = \frac{1}{2} - \frac{\text{rank}(\text{PnL}_i)}{M-1},$$

Training curves over 15 h of wall-clock are shown in Figure **??**. The baseline policy ($\sigma = 0$), corresponding to the unmodified pretrained model, achieves a mean PnL of approximately 7.66 cents. EGGROLL fine-tuning ($\sigma = 0.01$) lifts

| Hyperparameter | Value |
| --- | --- |
| Model | LOBS5-360M |
| Parallel generations per GPU | 2,048 |
| Total parallel generations | 65,536 |
| LoRA rank | $\{1, 4\}$ |
| Sigma | 0.01 |
| Learning rate $\eta$ | 0.001 |

*Table 6.* Model and EGGROLL fine-tuning settings for high-frequency trading.

the mean PnL to approximately 20 cents, a $\sim 2.7\times$ improvement over the baseline. The two LoRA-rank conditions reach very similar final PnL. Per epoch, $r=4$ improves slightly faster than $r=1$, but $r=4$ is also more expensive per epoch, so in wall-clock terms $r=1$ ends up slightly ahead at the 15 h mark (20.65 cents for $r=1$ vs 19.99 cents for $r=4$).

## N. Experimental Details

### N.1. Reinforcement Learning Experiments

Next, we compare the performance of EGGROLL against standard OpenES as implemented in (Salimans et al., 2017) on reinforcement learning tasks. Given the small network sizes, we can use OpenES at this scale, but as network sizes increase, the use of vanilla OpenES becomes computationally infeasible. We use the standard formulation of simply optimising for the final return in the environment. For both EGGROLL and OpenES, we perform hyperparameter optimisation (HPO) separately for each environment. For each algorithm–environment pair, we define plausible ranges for all key hyperparameters based on prior work and preliminary experiments. We then perform 20 random search trials, where each trial corresponds to a single training run with a randomly sampled hyperparameter configuration. Each configuration is evaluated based on the final return achieved by the mean policy parameters at the end of training. After all trials, we select the configuration that yields the highest final return. Using this best configuration, we then run 10 independent seeds to evaluate performance and report the mean and standard error of the mean across these seeds.

We use policy networks with 3 layers of 256 neurons and a range of environments that demonstrate different capabilities. We evaluate across the Navix (Pignatelli et al., 2024), Craftax (Matthews et al., 2024), Brax (Freeman et al., 2021), Kinetix (Matthews et al., 2025), and Jumanji (Bonnet et al., 2024) suites of environments. We evaluate 16 environments in total. We choose environments that are not trivial or impossible for PPO to solve, according to the original papers. We also choose environments that belong to different categories (e.g., environment size in Kinetix or categories in Jumanji).

We show a subsample of the evaluated environments in Figure 3 with the remaining results and hyperparameter details in Appendix N.1. Our findings show that EGGROLL is competitive with Open ES on 7/16 environments, underperforms on 2/16, and outperforms on 7/16. This does not take into account the speed-ups when compared to using OpenES with full-rank updates (see Figure11). We postulate that the reason for this performance increase is that the large networks are difficult to optimise for OpenES and lend themselves well to low rank updates.

We present here the hyperparameter ranges we used for hyperparameter optimisation, as well as all hyperparameter settings for all the experiments. All RL experiments were run on an NVIDIA L40S GPU. For PPO, we use the same methodology to tune the hyperparameters as we did for OpenES and EGGROLL as described in Section 6.1. We report the ranges and the final hyperparameters here. We train PPO agents using Rejax (Liesen et al., 2024). We use the activation function from Gallici et al. (2025) in our experiments, which we refer to as the "pqn" activation function in our hyperparameter tables.

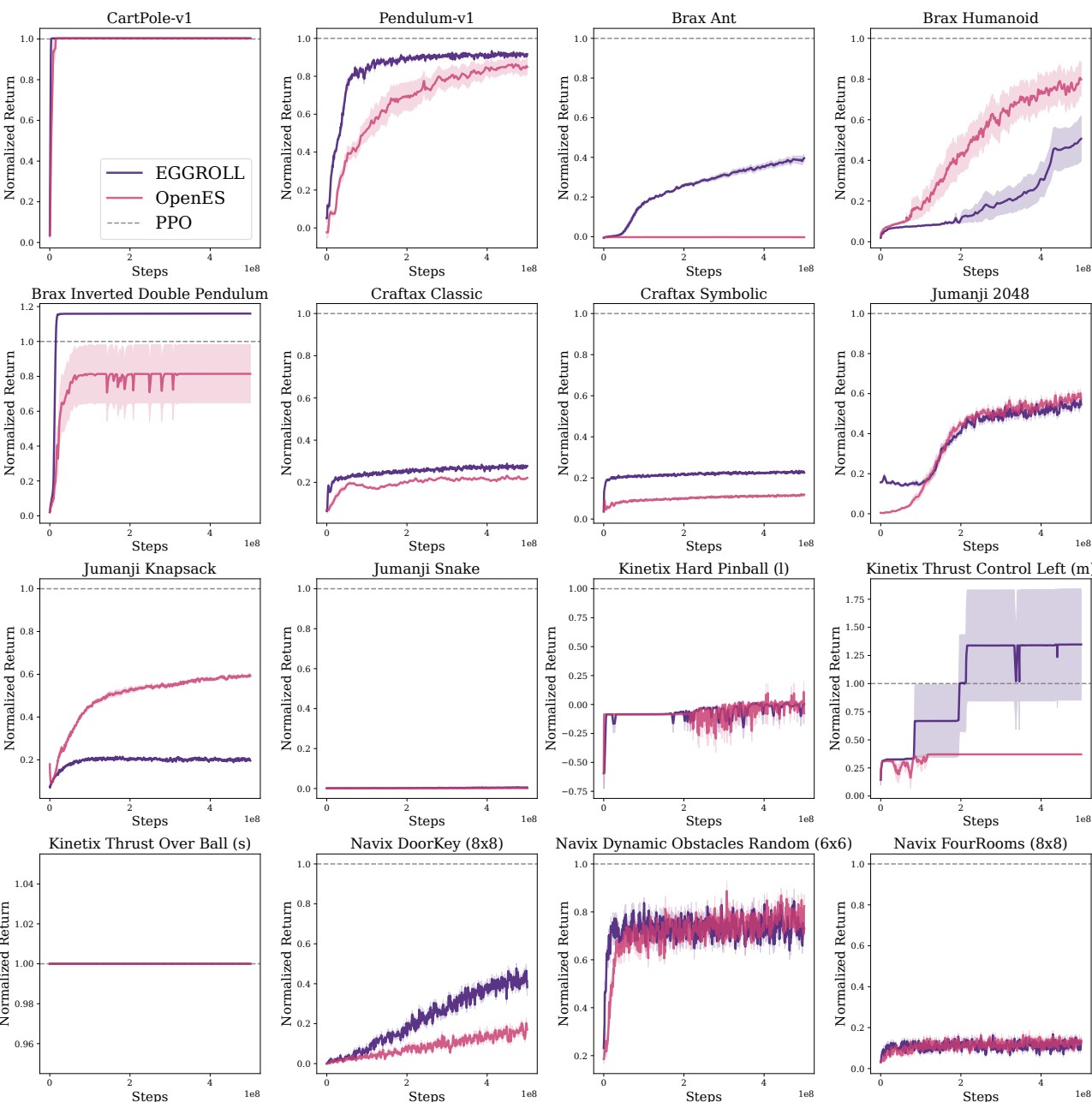

*Figure 10.* Comparison of reinforcement learning results: Mean returns for each environment and algorithm across 10 random seeds. The returns are evaluated using the mean of the parameters. HPO was conducted for each algorithm/environment pair. The shaded region is the standard error of the mean.

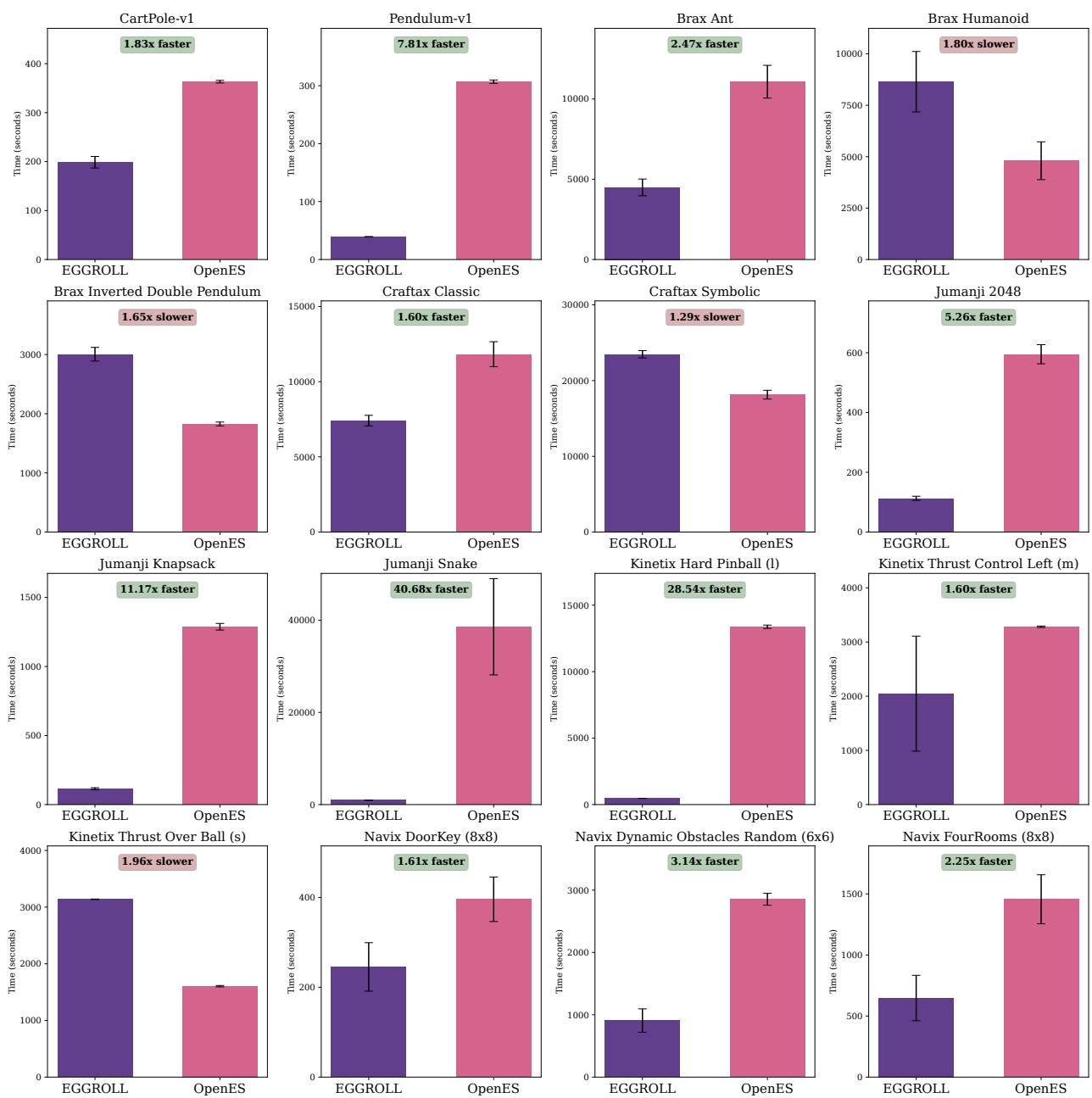

*Figure 11.* Comparison of reinforcement learning results: Mean and standard deviation of training time. Note that some of the timing difference is due to the differences in episode lengths, which is why the total time for EGGROLL sometimes appears longer than OpenES despite EGGROLL being faster on a per-timestep basis.

## N.2. Multi Agent Reinforcement Learning Experiments

*Table 7.* Hyperparameter Ranges for EGGROLL and OpenES

| Hyperparameter | Values |
|---|---|
| pop_size | 512, 1024, 2048, 4096 |
| n_parallel_evaluations | 1, 4, 8 |
| rank | 1, 2, 4 |
| optimiser | adamw, sgd, adam |
| learning_rate | 1e-3, 1e-2, 1e-1 |
| lr_decay | 0.995, 0.999, 0.9995, 1.0 |
| sigma | 0.05, 0.2, 0.5 |
| sigma_decay | 0.995, 0.999, 0.9995, 1.0 |
| rank_transform | true, false |
| deterministic_policy | true, false |

*Table 8.* Hyperparameter Ranges for PPO

| Hyperparameter | Values |
|---|---|
| clip_eps | 0.1, 0.2, 0.3 |
| ent_coef | 0, 0.0001, 0.001 |
| gae_lambda | 0.9, 0.95, 0.98 |
| gamma | 0.95, 0.99, 0.995, 0.999 |
| learning_rate | 0.0001, 0.0003, 0.001 |
| max_grad_norm | 0.5, 1, 2 |
| layer_size | 256 |
| n_layers | 3 |
| normalise_observations | true |
| normalise_rewards | false |
| num_envs | 64, 128, 256 |
| num_epochs | 4, 8, 16 |
| num_minibatches | 16, 32, 64 |
| num_steps | 64, 128, 256 |
| reward_normalisation_discount | 0.99 |
| skip_initial_evaluation | false |
| vf_coef | 0.5, 0.75, 1 |

*Table 9.* CartPole-v1

| Hyperparameter | eggroll | open_es |
|---|---|---|
| activation | pqn | pqn |
| deterministic_policy | false | true |
| learning_rate | 0.1 | 0.1 |
| lr_decay | 0.9995 | 0.9995 |
| layer_size | 256 | 256 |
| n_layers | 3 | 3 |
| n_parallel_evaluations | 1 | 4 |
| pop_size | 2048 | 512 |
| optimiser | sgd | adamw |
| rank | 4 | / |
| rank_transform | false | true |
| sigma | 0.2 | 0.5 |
| sigma_decay | 0.999 | 0.9995 |

*Table 10.* Pendulum-v1

| Hyperparameter | eggroll | open_es |
|---|---|---|
| activation | pqn | pqn |
| deterministic_policy | false | true |
| learning_rate | 0.01 | 0.01 |
| lr_decay | 0.995 | 0.995 |
| layer_size | 256 | 256 |
| n_layers | 3 | 3 |
| n_parallel_evaluations | 1 | 4 |
| pop_size | 4096 | 4096 |
| optimiser | adam | adamw |
| rank | 4 | / |
| rank_transform | false | false |
| sigma | 0.05 | 0.05 |
| sigma_decay | 0.995 | 1 |

We train on three cooperative Multi Particle Environments (MPEs) (Lowe et al., 2017) implemented in JaxMARL (Rutherford et al., 2024) with feed-forward networks of width 64 and depth 3, performing Bayesian hyperparameter optimisation for each environment and algorithm. All runs were executed on NVIDIA A100-SXM4-40GB GPUs. We find that the optimal

*Table 11.* brax/ant

| Hyperparameter | eggroll | open_es |
|---|---|---|
| activation | pqn | pqn |
| deterministic_policy | false | false |
| learning_rate | 0.01 | 0.1 |
| lr_decay | 0.9995 | 0.995 |
| layer_size | 256 | 256 |
| n_layers | 3 | 3 |
| n_parallel_evaluations | 1 | 8 |
| pop_size | 2048 | 512 |
| optimiser | adam | adam |
| rank | 1 | / |
| rank_transform | false | false |
| sigma | 0.05 | 0.05 |
| sigma_decay | 0.9995 | 0.9995 |

*Table 12.* brax/humanoid

| Hyperparameter | eggroll | open_es |
|---|---|---|
| activation | pqn | pqn |
| deterministic_policy | true | false |
| learning_rate | 0.1 | 0.1 |
| lr_decay | 1 | 0.995 |
| layer_size | 256 | 256 |
| n_layers | 3 | 3 |
| n_parallel_evaluations | 8 | 8 |
| pop_size | 4096 | 1024 |
| optimiser | adam | sgd |
| rank | 1 | / |
| rank_transform | true | true |
| sigma | 0.2 | 0.2 |
| sigma_decay | 0.9995 | 0.995 |

*Table 13.* brax/inverted_double_pendulum

| Hyperparameter | eggroll | open_es |
|---|---|---|
| activation | pqn | pqn |
| deterministic_policy | true | true |
| learning_rate | 0.1 | 0.1 |
| lr_decay | 1 | 0.995 |
| layer_size | 256 | 256 |
| n_layers | 3 | 3 |
| n_parallel_evaluations | 1 | 1 |
| pop_size | 2048 | 4096 |
| optimiser | adam | adam |
| rank | 2 | / |
| rank_transform | true | true |
| sigma | 0.5 | 0.05 |
| sigma_decay | 0.995 | 1 |

*Table 14.* craftax/Craftax-Classic-Symbolic-AutoReset-v1

| Hyperparameter | eggroll | open_es |
|---|---|---|
| activation | pqn | pqn |
| deterministic_policy | false | false |
| learning_rate | 0.01 | 0.001 |
| lr_decay | 0.995 | 0.995 |
| layer_size | 256 | 256 |
| n_layers | 3 | 3 |
| n_parallel_evaluations | 4 | 8 |
| pop_size | 2048 | 4096 |
| optimiser | sgd | adamw |
| rank | 1 | / |
| rank_transform | false | false |
| sigma | 0.05 | 0.05 |
| sigma_decay | 1 | 0.995 |

*Table 15.* craftax/Craftax-Symbolic-AutoReset-v1

| Hyperparameter | eggroll | open_es |
|---|---|---|
| activation | pqn | pqn |
| deterministic_policy | false | false |
| learning_rate | 0.01 | 0.1 |
| lr_decay | 0.999 | 0.995 |
| layer_size | 256 | 256 |
| n_layers | 3 | 3 |
| n_parallel_evaluations | 1 | 4 |
| pop_size | 512 | 1024 |
| optimiser | sgd | adam |
| rank | 4 | / |
| rank_transform | true | false |
| sigma | 0.05 | 0.5 |
| sigma_decay | 0.999 | 1 |

*Table 16.* jumanji/Game2048-v1

| Hyperparameter | eggroll | open_es |
|---|---|---|
| activation | pqn | pqn |
| deterministic_policy | false | true |
| learning_rate | 0.1 | 0.01 |
| lr_decay | 1 | 0.999 |
| layer_size | 256 | 256 |
| n_layers | 3 | 3 |
| n_parallel_evaluations | 4 | 4 |
| pop_size | 1024 | 1024 |
| optimiser | adamw | adamw |
| rank | 1 | / |
| rank_transform | false | true |
| sigma | 0.5 | 0.05 |
| sigma_decay | 0.9995 | 0.9995 |

batch size is consistent across algorithms on the same environment. Figure 12 shows that EGGROLL with rank 1 trains up to 2.4 times faster than OpenES for large batch sizes while staying competitive in performance.

*Table 17.* jumanji/Knapsack-v1

| Hyperparameter | eggroll | open_es |
| --- | --- | --- |
| activation | pqn | pqn |
| deterministic_policy | false | false |
| learning_rate | 0.1 | 0.01 |
| lr_decay | 0.999 | 1 |
| layer_size | 256 | 256 |
| n_layers | 3 | 3 |
| n_parallel_evaluations | 4 | 1 |
| pop_size | 1024 | 2048 |
| optimiser | sgd | adamw |
| rank | 4 | / |
| rank_transform | true | true |
| sigma | 0.05 | 0.5 |
| sigma_decay | 1 | 0.995 |

*Table 18.* jumanji/Snake-v1

| Hyperparameter | eggroll | open_es |
| --- | --- | --- |
| activation | pqn | pqn |
| deterministic_policy | false | false |
| learning_rate | 0.001 | 0.001 |
| lr_decay | 0.9995 | 1 |
| layer_size | 256 | 256 |
| n_layers | 3 | 3 |
| n_parallel_evaluations | 8 | 1 |
| pop_size | 4096 | 2048 |
| optimiser | adam | sgd |
| rank | 1 | / |
| rank_transform | true | false |
| sigma | 0.05 | 0.2 |
| sigma_decay | 0.9995 | 1 |

*Table 19.* kinetix/l/hard_pinball

| Hyperparameter | eggroll | open_es |
| --- | --- | --- |
| activation | pqn | pqn |
| deterministic_policy | true | true |
| learning_rate | 0.01 | 0.01 |
| lr_decay | 0.995 | 1 |
| layer_size | 256 | 256 |
| n_layers | 3 | 3 |
| n_parallel_evaluations | 8 | 1 |
| pop_size | 2048 | 512 |
| optimiser | sgd | sgd |
| rank | 4 | / |
| rank_transform | true | true |
| sigma | 0.05 | 0.5 |
| sigma_decay | 0.999 | 0.9995 |

*Table 20.* kinetix/m/h17_thrustcontrol_left

| Hyperparameter | eggroll | open_es |
| --- | --- | --- |
| activation | pqn | pqn |
| deterministic_policy | false | false |
| learning_rate | 0.1 | 0.001 |
| lr_decay | 0.9995 | 1 |
| layer_size | 256 | 256 |
| n_layers | 3 | 3 |
| n_parallel_evaluations | 4 | 1 |
| pop_size | 512 | 1024 |
| optimiser | sgd | adam |
| rank | 4 | / |
| rank_transform | true | true |
| sigma | 0.5 | 0.5 |
| sigma_decay | 1 | 0.999 |

*Table 21.* kinetix/s/h1_thrust_over_ball

| Hyperparameter | eggroll | open_es |
| --- | --- | --- |
| activation | pqn | pqn |
| deterministic_policy | false | false |
| learning_rate | 0.1 | 0.01 |
| lr_decay | 0.995 | 0.995 |
| layer_size | 256 | 256 |
| n_layers | 3 | 3 |
| n_parallel_evaluations | 1 | 1 |
| pop_size | 512 | 2048 |
| optimiser | adamw | sgd |
| rank | 1 | / |
| rank_transform | true | true |
| sigma | 0.5 | 0.05 |
| sigma_decay | 0.9995 | 1 |

*Table 22.* navix/Navix-DoorKey-8x8-v0

| Hyperparameter | eggroll | open_es |
| --- | --- | --- |
| activation | pqn | pqn |
| deterministic_policy | false | false |
| learning_rate | 0.01 | 0.01 |
| lr_decay | 0.9995 | 1 |
| layer_size | 256 | 256 |
| n_layers | 3 | 3 |
| n_parallel_evaluations | 1 | 8 |
| pop_size | 1024 | 2048 |
| optimiser | adamw | adam |
| rank | 1 | / |
| rank_transform | false | true |
| sigma | 0.05 | 0.05 |
| sigma_decay | 1 | 1 |

*Table 23.* navix/Navix-Dynamic-Obstacles-6x6-Random-v0

| Hyperparameter | eggroll | open_es |
|---|---|---|
| activation | pqn | pqn |
| deterministic_policy | false | false |
| learning_rate | 0.01 | 0.01 |
| lr_decay | 0.999 | 1 |
| layer_size | 256 | 256 |
| n_layers | 3 | 3 |
| n_parallel_evaluations | 4 | 1 |
| pop_size | 512 | 4096 |
| optimiser | adam | adam |
| rank | 2 | / |
| rank_transform | false | false |
| sigma | 0.05 | 0.2 |
| sigma_decay | 1 | 0.995 |

*Table 24.* navix/Navix-FourRooms-v0

| Hyperparameter | eggroll | open_es |
|---|---|---|
| activation | pqn | pqn |
| deterministic_policy | false | false |
| learning_rate | 0.01 | 0.001 |
| lr_decay | 0.999 | 0.9995 |
| layer_size | 256 | 256 |
| n_layers | 3 | 3 |
| n_parallel_evaluations | 4 | 4 |
| pop_size | 2048 | 2048 |
| optimiser | sgd | adam |
| rank | 4 | / |
| rank_transform | true | false |
| sigma | 0.05 | 0.05 |
| sigma_decay | 0.9995 | 0.9995 |

*Table 25.* PPO Hyperparameters (Set 1)

| Hyperparameter | CartPole | Pendulum | Ant | Humanoid | IDP | CraftaxClassic | CraftaxSymbolic | Game2048 |
|---|---|---|---|---|---|---|---|---|
| activation | pqn | pqn | pqn | pqn | pqn | pqn | pqn | pqn |
| clip_eps | 0.2 | 0.1 | 0.2 | 0.3 | 0.1 | 0.2 | 0.2 | 0.3 |
| ent_coef | 0.0001 | 0.001 | 0 | 0.0001 | 0.0001 | 0.0001 | 0 | 0.001 |
| gae_lambda | 0.9 | 0.95 | 0.95 | 0.9 | 0.98 | 0.98 | 0.9 | 0.9 |
| gamma | 0.995 | 0.999 | 0.995 | 0.95 | 0.99 | 0.95 | 0.95 | 0.99 |
| learning_rate | 0.0003 | 0.0003 | 0.0003 | 0.0001 | 0.001 | 0.001 | 0.0003 | 0.0003 |
| max_grad_norm | 0.5 | 1 | 0.5 | 2 | 2 | 2 | 2 | 2 |
| layer_size | 256 | 256 | 256 | 256 | 256 | 256 | 256 | 256 |
| n_layers | 3 | 3 | 3 | 3 | 3 | 3 | 3 | 3 |
| normalise_obs | true | true | true | true | true | true | true | true |
| normalise_rew | false | false | false | false | false | false | false | false |
| num_envs | 256 | 256 | 64 | 256 | 64 | 128 | 256 | 64 |
| num_epochs | 4 | 16 | 8 | 4 | 4 | 4 | 4 | 8 |
| num_minibatches | 32 | 16 | 32 | 64 | 64 | 32 | 32 | 16 |
| num_steps | 128 | 256 | 128 | 64 | 128 | 128 | 64 | 64 |
| rew_norm_discount | 0.99 | 0.99 | 0.99 | 0.99 | 0.99 | 0.99 | 0.99 | 0.99 |
| skip_initial_eval | false | false | false | false | false | false | false | false |
| vf_coef | 0.5 | 1 | 1 | 0.75 | 1 | 0.5 | 0.75 | 0.75 |

*Table 26.* PPO Hyperparameters (Set 2)

| Hyperparameter | Knapsack | Snake | HardPinball | ThrustLeft | ThrustBall | DoorKey | DynamicObs | FourRooms |
|---|---|---|---|---|---|---|---|---|
| activation | pqn | pqn | pqn | pqn | pqn | pqn | pqn | pqn |
| clip_eps | 0.1 | 0.3 | 0.1 | 0.2 | 0.2 | 0.1 | 0.1 | 0.1 |
| ent_coef | 0.0001 | 0.001 | 0.0001 | 0.0001 | 0.0001 | 0.0001 | 0.001 | 0.001 |
| gae_lambda | 0.9 | 0.95 | 0.9 | 0.9 | 0.95 | 0.98 | 0.98 | 0.9 |
| gamma | 0.99 | 0.999 | 0.99 | 0.995 | 0.999 | 0.95 | 0.999 | 0.99 |
| learning_rate | 0.0001 | 0.0001 | 0.0001 | 0.0001 | 0.0001 | 0.0003 | 0.001 | 0.001 |
| max_grad_norm | 0.5 | 0.5 | 1 | 2 | 0.5 | 0.5 | 1 | 1 |
| layer_size | 256 | 256 | 256 | 256 | 256 | 256 | 256 | 256 |
| n_layers | 3 | 3 | 3 | 3 | 3 | 3 | 3 | 3 |
| normalise_obs | true | true | true | true | true | true | true | true |
| normalise_rew | false | false | false | false | false | false | false | false |
| num_envs | 256 | 128 | 256 | 256 | 64 | 64 | 128 | 256 |
| num_epochs | 4 | 4 | 16 | 16 | 16 | 16 | 4 | 8 |
| num_minibatches | 64 | 16 | 16 | 32 | 16 | 64 | 16 | 32 |
| num_steps | 128 | 128 | 64 | 128 | 64 | 256 | 128 | 256 |
| rew_norm_discount | 0.99 | 0.99 | 0.99 | 0.99 | 0.99 | 0.99 | 0.99 | 0.99 |
| skip_initial_eval | false | false | false | false | false | false | false | false |
| vf_coef | 0.75 | 0.75 | 0.5 | 0.5 | 0.5 | 0.75 | 0.5 | 0.75 |

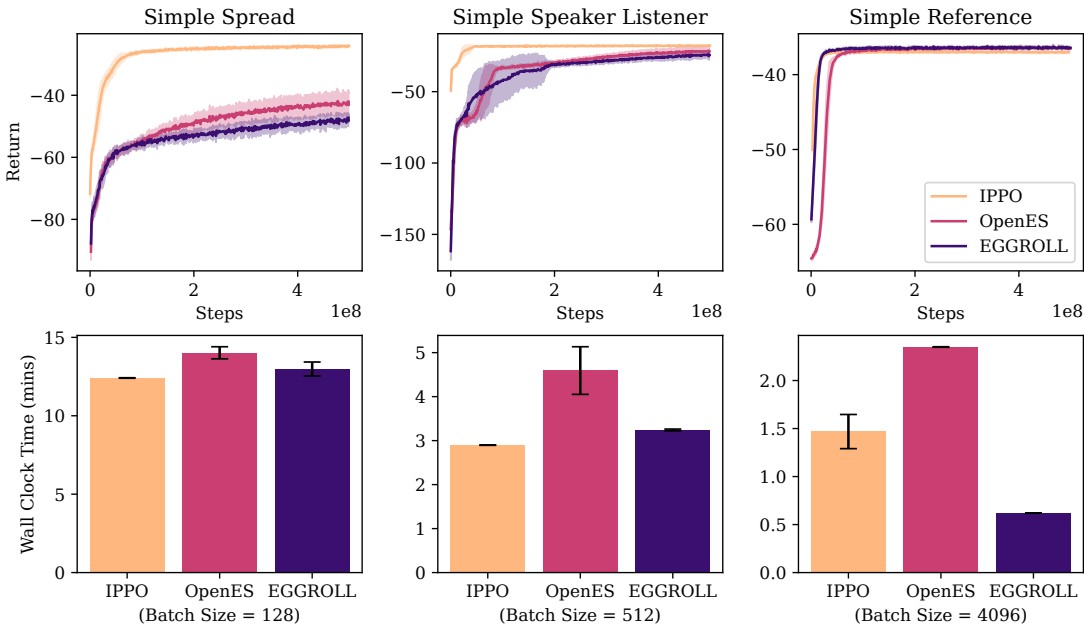

*Figure 12.* Training curves and wall clock times for cooperative Multi Particle Environments. Hyperparameter optimisation yielded equal batch sizes for all algorithms on the same environment. All EGGROLL runs used rank 1 perturbations. Shaded regions are standard errors of mean values.

*Table 27.* Hyperparameter Ranges Used in MPE Sweeps for EGGROLL and OpenES

| Hyperparameter | Values |
| --- | --- |
| activation | pqn, tanh |
| pop_size | 128, 512, 1024, 2048, 4096 |
| learning_rate | 0.01, 0.05, 0.1, 0.5 |
| lr_decay | 0.3, 0.7, 1.0 |
| sigma | 0.1, 0.2, 0.3, 0.4, 0.5 |
| rank_transform | true, false |

*Table 28.* Hyperparameter Ranges Used in MPE Sweeps for IPPO

| Hyperparameter | Values |
| --- | --- |
| activation | relu, tanh |
| pop_size | 128, 512, 1024, 2048, 4096 |
| learning_rate | 5e-5, 1e-4, 2.5e-4, 1e-3 |
| entropy_coef | 0.001, 0.005, 0.01 |

### N.3. Reasoning Fine-tuning Experiments: Countdown

We ran a Bayesian hyper-parameter sweep (Snoek et al., 2012) for both GRPO and EGGROLL and used the best set found to run the experiments in figure 4b. For GRPO we swept over sampling temperature and learning rate, whereas for EGGROLL we swept over the standard deviation of the ES sampling ($\sigma$) and the learning rate scale. The best hyper-parameters found are detailed on tables 32 (EGGROLL) and 33 (GRPO). All of the experiments run in 8 hours on a NVIDIA H200 GPU.

We also run an experiment where we increase the number of GPUs to 8 and use a bigger model, RWKV 7g7B, on the Countdown task, allowing for stronger final performance. Notably, we compare to the results reported by (Qiu et al., 2025) on Countdown. Figure 13a shows that starting from our significantly weaker model (RWKV 7g7B v.s. Qwen 2.5-7B), we are able to train to a higher validation accuracy (72.9%), v.s. the ones reported for training with GRPO (52.8%) and Open ES (66.8%). (Qiu et al., 2025) do not report the wall clock time or the hardware used for their experiments which makes it difficult to establish a fair comparison.

### N.4. Reasoning Fine-tuning Experiments: GSM8K

We used the hyper-parameters found for Countdown as a starting point and reduced the learning rates for both GRPO and EGGROLL using linear search until we found the best performing one on the validation set. Our experiments for GSM8K run on 8 NVIDIA H200 GPUS for 8 hours each. We also increase the standard deviation, $\sigma$, parameter for ES (from $7 \times 10^{-4}$ to $2 \times 10^{-3}$) as the significantly bigger population sizes (8096 v.s. 512) allow for much more stable training and aggressive exploration.

*Table 29.* MPE Simple Spread v3

| Hyperparameter | eggroll | open_es | ippo |
|---|---|---|---|
| activation | tanh | tanh | tanh |
| deterministic_policy | true | true | false |
| learning_rate | 0.01 | 0.01 | 0.001 |
| lr_decay | 0.7 | 0.7 | linear |
| layer_size | 64 | 64 | 64 |
| n_layers | 3 | 3 | 3 |
| pop_size | 128 | 128 | 128 |
| optimiser | adamw | adamw | adam |
| rank | 1 | 1 | - |
| rank_transform | false | false | - |
| sigma | 0.5 | 0.5 | - |
| n_minibatches | - | - | 4 |
| update_epochs | - | - | 4 |
| gamma | - | - | 0.99 |
| gae_lambda | - | - | 0.95 |
| epsilon_clip | - | - | 0.2 |
| entropy_coef | - | - | 0.01 |
| value_coef | - | - | 0.5 |
| max_grad_norm | - | - | 0.5 |

*Table 30.* MPE Simple Speaker Listener v4

| Hyperparameter | eggroll | open_es | ippo |
|---|---|---|---|
| activation | tanh | tanh | relu |
| deterministic_policy | true | true | false |
| learning_rate | 0.01 | 0.01 | 0.001 |
| lr_decay | 0.7 | 0.3 | linear |
| layer_size | 64 | 64 | 64 |
| n_layers | 3 | 3 | 64 |
| pop_size | 512 | 512 | 512 |
| optimiser | adamw | adamw | adam |
| rank | 1 | 1 | - |
| rank_transform | true | true | - |
| sigma | 0.5 | 0.5 | - |
| n_minibatches | - | - | 4 |
| update_epochs | - | - | 4 |
| gamma | - | - | 0.99 |
| gae_lambda | - | - | 0.95 |
| epsilon_clip | - | - | 0.2 |
| entropy_coef | - | - | 0.005 |
| value_coef | - | - | 0.5 |
| max_grad_norm | - | - | 0.5 |

*Table 31.* MPE Simple Reference v3

| Hyperparameter | eggroll | open_es | ippo |
|---|---|---|---|
| activation | pqn | tanh | relu |
| deterministic_policy | true | true | false |
| learning_rate | 0.01 | 0.01 | 0.001 |
| lr_decay | 0.3 | 0.3 | linear |
| layer_size | 64 | 64 | 64 |
| n_layers | 3 | 3 | 3 |
| pop_size | 4096 | 4096 | 4096 |
| optimiser | adamw | adamw | adam |
| rank | 1 | 1 | - |
| rank_transform | false | true | - |
| sigma | 0.1 | 0.3 | - |
| n_minibatches | - | - | 4 |
| update_epochs | - | - | 4 |
| gamma | - | - | 0.99 |
| gae_lambda | - | - | 0.95 |
| epsilon_clip | - | - | 0.2 |
| entropy_coef | - | - | 0.01 |
| value_coef | - | - | 0.5 |
| max_grad_norm | - | - | 0.5 |

## O. Empirical Validation of Rank Analysis

Denote the $N$-sample Monte Carlo estimate of $\hat{g}_{\text{LR}}^r$ as $\hat{g}_{\text{LR}}^{N,r}$ and the $N$-sample Monte Carlo estimate of the full-rank update $\nabla_\mu J(\mu)$ as $\hat{g}_{\text{True}}^N$. We study the convergence rate of $\mathbb{E}\left[\|\hat{g}_{\text{LR}}^{N,r} - \hat{g}_{\text{True}}^N\|_F\right]$. We note that, for finite $N$, the limit $\lim_{r\to\infty}\mathbb{E}\left[\|\hat{g}_{\text{LR}}^{N,r} - \hat{g}_{\text{True}}^N\|_F\right] = \mathbb{E}\left[\|\hat{g}_{\text{True}}^{N'} - \hat{g}_{\text{True}}^N\|_F\right] = b$ converges to some base value $b$ where we expect $b > 0$ for $N < \infty$ because $\hat{g}_{\text{True}}^{N'}$ and $\hat{g}_{\text{True}}^N$ are both independent estimators formed from independent samples and so won't automatically cancel in the limit unless $N \to \infty$. The value of $b$ thus represents the *floor noise* that arises from the variance due to Monte Carlo sampling in the estimators. Our goal is to investigate the rate of convergence to floor noise value $b$, that

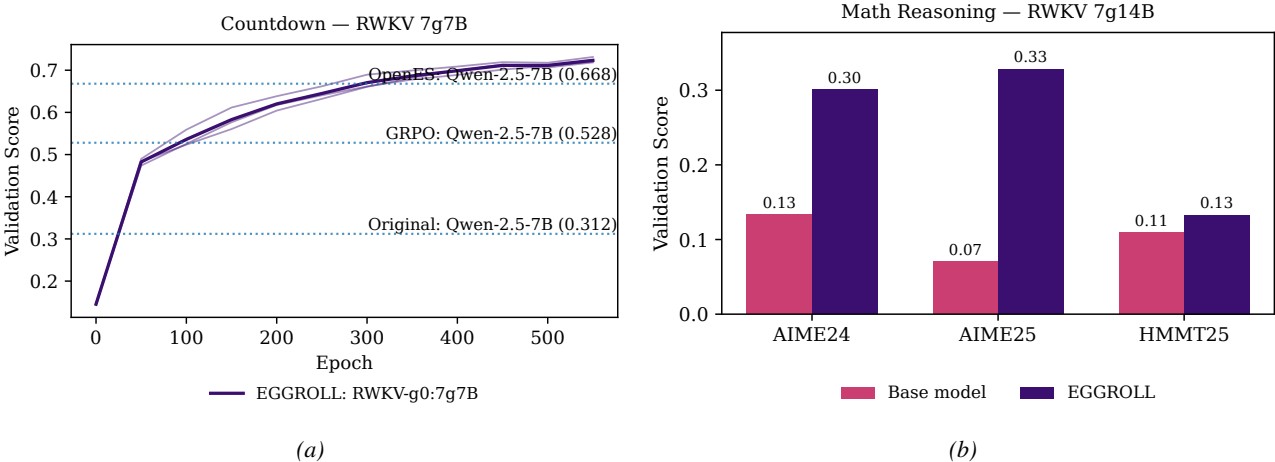

*(a)*            *(b)*

*Figure 13.* (a) Comparison of our finetuned RWKV 7G 7 billion parameter model using 8 GPUS with the results reported by (Qiu et al., 2025) on similarly sized Qwen models. (b) Performance of our finetuned RWKV 7G 14 billion parameter model on hard reasoning tasks using 32 GPUs for 12 hours. The model was trained using the DeepScaleR dataset and the best checkpoint was chosen by evaluating on AIME24. Due to the size of the model we were not able to run similar baseline experiments using GRPO.

| Hyperparameter | Value |
|---|---|
| Model | RWKV 7g1.5B |
| Optimiser | Gradient descent |
| ES standard deviation $\sigma$ | $7 \times 10^{-4}$ |
| Rank $r$ | 1 |
| Learning-rate scale $\eta_{\text{scale}}$ | 0.125 |
| Population size | 256 |
| Parallel generations per GPU | 1536 |
| Prompts per epoch | 6 |
| Generation / thinking length | 1000 tokens |
| Train / val temperature | 0 / 0 |
| Parallel validations | 128 |

*Table 32.* Key hyperparameters for EGGROLL training on Countdown with FastRWKV-7g1.5B.

is a relationship of the form: $\mathbb{E}\left[\|\hat{g}_{\text{LR}}^{N,r} - \hat{g}_{\text{True}}^{N}\|_F\right] = \mathcal{O}(r^{-1}) + b$, which would validate our theory under the Monte Carlo sampling regime encountered in practice.

We conducted additional empirical experiments to confirm this relationship. Specifically, we constructed a ReLU 2-layer MLP with hidden dimension 64 (for a total of 4736 parameters) with He initialisation, to estimate the ES gradient with sigma=1.0 using cross-entropy loss on the first 10 samples of MNIST. We used 10 seeds to estimate the floor noise $b$ by estimating $\mathbb{E}\left[\|\hat{g}_{\text{True}}^{N'} - \hat{g}_{\text{True}}^{N}\|_F\right]$ and 10 seeds to estimate $\mathbb{E}\left[\|\hat{g}_{\text{LR}}^{N,r} - \hat{g}_{\text{True}}^{N}\|_F\right]$ for $r \in \{1 : 32\}$. We fit a curve of the form $ar^{-p} + b$, obtaining $p = 1.191 \pm 0.003$ and $b = 0.0654 \pm 0.0003$. The coefficient of determination was $R^2 = 0.968$. Results are show in Figure 15a, where we also plot the upper bound of the form $\mathcal{O}(r^{-1}) + b$. In Figure 15b, we show the same plot with floor noise removed. These results empirically validate our claim, as $p = 1.191 \pm 0.003 > 1$, confirming our upper bound on convergence rate as rank $r$ increases is correct, even in the Monte Carlo sampling regime.

| Hyperparameter | Value |
| --- | --- |
| Model | RWKV 7g1.5B |
| Optimiser | Radam |
| Learning rate $\eta$ | $3 \times 10^{-6}$ |
| Generations per prompt $G$ | 8 |
| Parallel generations per GPU | 64 |
| Prompts per epoch | 8 |
| Generation length | 1000 tokens |
| Number of minibatches | 4 |
| PPO clip parameter $\epsilon_{\text{clip}}$ | 0.2 |
| Train / val temperature | 1 / 0 |
| Parallel validations | 128 |

*Table 33.* Key hyperparameters for GRPO training on Countdown with AssociativeScanRWKV-7g1.5B.

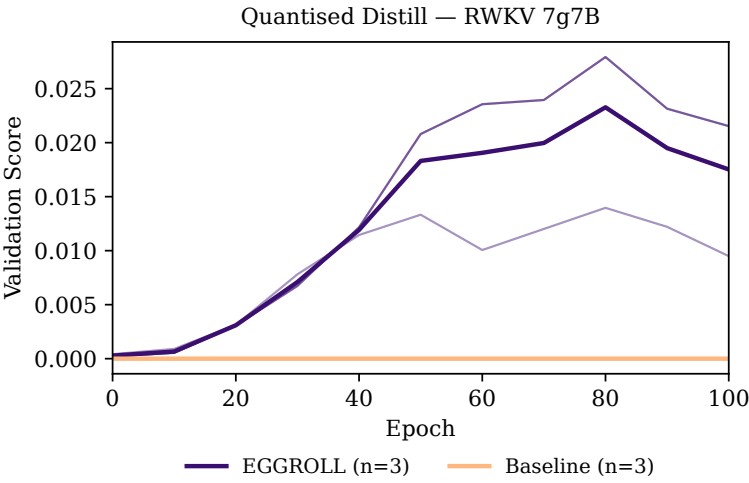

*Figure 14.* Validation score on unseen examples of GSM8K of 3 seeds of a quantised RWKV 7G 7 billion parameter model. Initially the model is unable to solve any problems, but progressively it is capable of solving more problems. The baseline here indicates the validation score of a quantised model without any further training.

| Hyperparameter | Value |
| --- | --- |
| Model | RWKV 7g7B |
| ES standard deviation $\sigma$ | $2 \times 10^{-3}$ |
| Rank $r$ | 1 |
| Learning-rate scale $\eta_{\text{scale}}$ | 0.06 |
| Generations per prompt $G$ | 512 |
| Parallel generations per GPU | 1024 |
| Total parallel generations | 8192 |
| Prompts per epoch | 16 |
| Generation length | 1000 tokens |
| Noise reuse factor | 1 |
| Freeze non-LoRA params | True |
| Train / val temperature | 0 / 0 |
| Parallel validations | 128 |

*Table 34.* Key hyperparameters for multi-GPU EGGROLL training on GSM8K with FastRWKV-7g7B.

| Hyperparameter | Value |
|---|---|
| Model | RWKV 7g7B |
| Learning rate $\eta$ | $1 \times 10^{-6}$ |
| Generations per prompt $G$ | 8 |
| Parallel generations per GPU | 32 |
| Total parallel generations | 256 |
| Prompts per epoch | 32 |
| Generation length | 1000 tokens |
| Number of minibatches | 16 |
| Number of workers (processes) | 8 |
| PPO clip parameter $\epsilon_{\text{clip}}$ | 0.2 |
| Train / val temperature | 1 / 0 |
| Parallel validations | 128 |

*Table 35.* Key hyperparameters for multi-GPU GRPO training on GSM8K with AssociativeScanRWKV-7g7B.

| Hyperparameter | Value |
|---|---|
| Model | RWKV 7g7B |
| Optimiser | EGGROLL (Quantised)) |
| ES standard deviation $\sigma$ | 0.4 |
| Rank $r$ | 1 |
| Learning-rate scale $\eta_{\text{scale}}$ | $3 \times 10^{-7}$ |
| Population size | 8192 |
| Parallel generations per GPU | 256 |
| Prompts per epoch | 1 |
| Generation / thinking length | 256 tokens |
| Train / val temperature | 0 / 0 |
| Parallel validations | 128 |

*Table 36.* Key hyperparameters for quantised EGGROLL training on GSM8K (teacher-forced) with RWKV-7g7B.

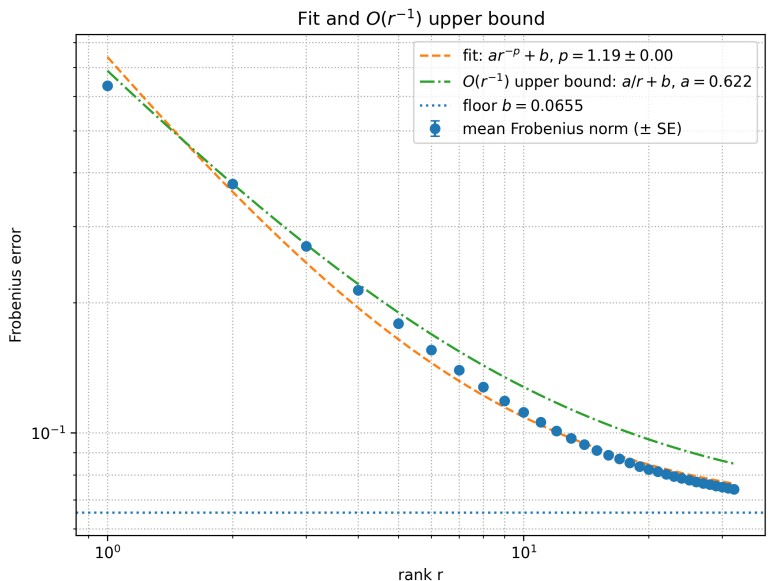

*(a)* Frobenius error versus rank, comparing empirical points to theoretical upper bound.

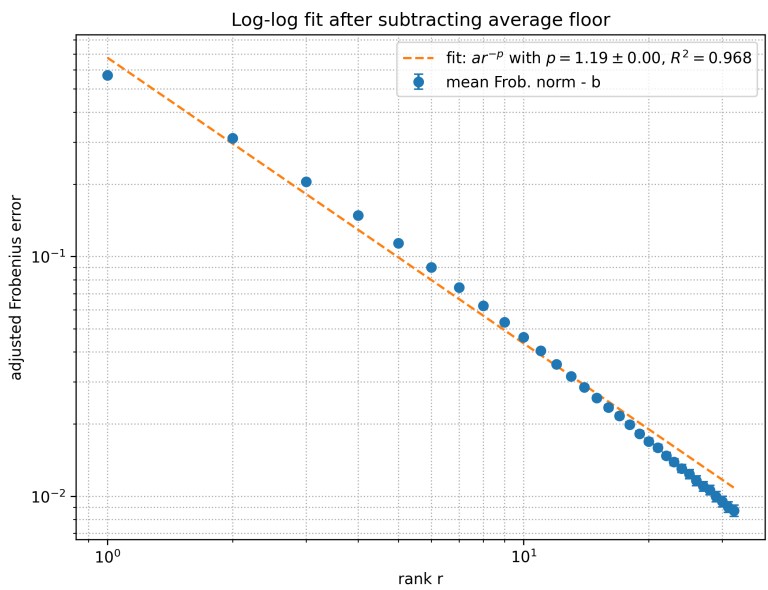

*(b)* Adjusted Frobenius error versus rank, subtracting average floor.

*Figure 15.* Empirical Valiation of Convergence Rate of Monte Carlo Estimate of EGGROLL Update for Rank

