# OpenReview forum: "Evolution Strategies at the Hyperscale"
_ICML.cc/2026/Conference — ICML 2026 regular_

### Official Review · Reviewer_gBKy · 2026-03-06

**Soundness:** 2
**Presentation:** 2
**Significance:** 3
**Originality:** 3
**Overall Recommendation:** 4
**Confidence:** 3

**Summary:**

The paper proposes EGGROLL, a hardware-efficient evolution strategies method for training large neural networks with low-rank perturbations. The main idea is to replace full-rank Gaussian noise by low-rank perturbations, which allows LoRA-like batched computation and gives much higher arithmetic intensity compared to naive ES. The authors also provide convergence analysis, showing that the low-rank estimator becomes closer to the Gaussian ES estimator when the rank increases, and study noise-scaling regimes in high dimensions. Experiments include integer-only pretraining, RL control tasks, reasoning post-training of RWKV models, and quantized distillation.

**Compliance With Llm Reviewing Policy:**

Affirmed.

**Final Justification:**

The new Qwen3-4B GRPO experiments and theory validation addressed the majority of my concerns, and it seems fair to raise my score to 4. However, the revised paper must incorporate the corrections from discussion: confidence intervals for all comparisons (the updated Figure 4b shows substantial overlap between EGGROLL and GRPO on RWKV), the Qwen results in the main body, and an explicit statement that EGGROLL is not compute-efficient for pretraining. Critically, results with matched 16k context length for Qwen-Base are needed, as the reference GRPO at 16k context outperforms all reported GRPO and EGGROLL results at shorter contexts.

**Key Questions For Authors:**

1. Could the authors add wall-clock curves (loss/perplexity vs time) for Section 6.1, and clarify in what regime EGGROLL should be considered as a practically useful method for pretraining?
2. Could the authors provide a direct empirical validation of the theory, for example by showing how the discrepancy between the low-rank update and the Gaussian ES update behaves for different rank $r$, and/or diagnostics for different $\sigma_d$-schedules during training? This is important because these claims are not directly validated in the experiments
3. There is an error in the theory presentation: the line 289 place the rate $O(1/\sqrt r)$ is mentioned, while in Theorem 5.1 place the claim is formulated with $O(1/r)$ rate.
4. Recent work [1] reports that ES can achieve performance close to GRPO, but can also come with substantially larger forgetting and update norms. Could the authors analyze forgetting/OOD-shift/retention of prior abilities for EGGROLL and compare it with GRPO? Current theoretical part analyzes local convergence, but not continual-learning behaviour, so such analysis would be valuable.

[1] Abdi I. et al. Evolutionary Strategies lead to Catastrophic Forgetting in LLMs //arXiv preprint arXiv:2601.20861. – 2026.

**Limitations:**

yes

**Strengths And Weaknesses:**

**Strengths**

- The paper proposes a technically clear idea for making evolution strategies more hardware-efficient, by replacing full-rank perturbations with low-rank ones.
- The work combines systems motivation with theoretical analysis, including the convergence of the low-rank estimator to the Gaussian ES estimator and the analysis of noise scaling in high dimensions.
- The empirical evaluation covers several different settings, including RL control, integer-only pretraining, reasoning post-training, and quantized distillation, which helps to show that the method is not tied only to one narrow application.

**Weaknesses**

- The practical value of the pure-integer pretraining remains not fully convincing. In line 383 it is written that "our largest population size of $2^{20}=1048576$ requires around 180 times more GPU-hours than the backprop baseline", but the comparison is shown in training steps and final quality, not in wall-clock. In this form, the result looks more like a proof-of-concept than a practical alternative to gradient-based pretraining.
- The main LLM results in the main paper are concentrated on RWKV, motivating this by architectural advantage (line 381). Therefore, the generalizability of the headline claims to attention-based LLMs is shown only in a limited way; the results on Qwen are moved to Appendix L and do not look as convincing as the main-text evidence.
- Part of the comparisons on transformer LLMs is not a proper apples-to-apples comparison. In line 2796 the authors write that they take the RL results from [1]. However, in that paper the authors use not the DeepScaler dataset, but math:Orz57K, so the setups for comparison are not correct, since in line 2790 the authors write that they use DeepScaler for EGGROLL. There are DeepScaler results for Qwen3-4B-Base in [2], and there GRPO (Table 1, s=0) shows much better values. This strongly weakens the conclusion about parity of the methods. Also, the conclusions in Figure 4(b) and Figure 14(b) are made without multi-seed uncertainty, which leaves questions about statistical significance.
- Section 4.6 is not fully developed as a comparison of methods. The authors show recovery perplexity/accuracy, but they do not compare this approach with gradient-based distillation on quality, memory, and wall-clock. Because of this, the practical value of black-box distillation itself remains unclear.

[1] Liu Z. et al. Gem: A gym for agentic llms //arXiv preprint arXiv:2510.01051. – 2025.

[2] Zheng H., Zhao J., Chen B. Prosperity before Collapse: How Far Can Off-Policy RL Reach with Stale Data on LLMs? //arXiv preprint arXiv:2510.01161. – 2025.

---

> ### Author Rebuttal · Authors · 2026-03-31
>
> Thank you for the feedback! We appreciate that you found our hardware-efficient algorithm technically clear, combining theoretical analysis and several different settings. In this rebuttal, we would like to present additional experiments on transformer fine-tuning and clarify additional details.
>
>
> # Transformer GRPO Experiments (W2,3)
>
> We placed transformer GRPO results in the appendix due to space constraints; we apologise if this positioning mistakenly suggested weaker performance.
>
> We thank reviewers 4SQb and gBKy for pointing out the issue with taking the baselines from the paper we referenced. We set up GRPO baselines (including hyperparameter tuning) using verl for proper comparison.
>
> | Method| Dataset | Initial model| Max tokens | Math500 | Minerva | OlympiadBench | AIME24 | AIME25 | Average |
> |-|-|-|-|-|-|-|-|-|-|
> | No Training| n/a | Qwen3-4B-Base | 4k | 50.2 | 28.0 | 24.4 | 10.0 | 7.4 | 24.0 |
> | GRPO from reference [2] | DeepScaleR | “ | **16k** | 84.6 | 40.2 | 50.5 | 22.9 | 20.2 | 43.7 |
> | GRPO (ours) | “ | “ | 4k | 72.8 | 30.0 | 30.7 | 7.6 | 10.0 | 30.2 |
> | EGGROLL (16k population) | “ | “ | 4k | 83.6 | 31.3 | 35.2 | 18.0 | 19.5 | 37.5 |
> | No Training | n/a | Qwen3-4B | 8k | 82.0 | 29.8 | 33.3 | 21.1 | 18.9 | 37.0 |
> | GRPO (ours) | DeepScaleR | “ | 8k | 89.4 | 34.7 | 37.8 | 32.1 | 43.6 | 47.5 |
> | EGGROLL (256 population)| “ | “ | 8k | 85.9 | 42.2 | 43.0 | 44.5 | 33.6 | 49.8 |
> | “ (4k “) | “ | “ | 8k | 89.8 | 43.0 | 46.1 | 55.5 | 23.4 | 51.6 |
> | “ (16k “)| “ | “ | 8k | 91.4 | 44.5 | 51.6 | 54.7 | 41.4 | 56.7 |
> | GRPO (ours) | ORZ | “ | 8k | 89.0 | 35.5 | 38.0 | 26.1 | 32.4 | 44.2 |
> | EGGROLL (256 population)| “ | “ | 8k | 93.0 | 41.4 | 45.3 | 35.9 | 26.6 | 48.4 |
> | “ (4k “) | “ | “ | 8k | 91.4 | 44.5 | 49.2 | 56.2 | 38.3 | 55.9 |
> | “ (16k “)| “ | “ | 8k | 90.6 | 43.8 | 52.3 | 64.8 | 41.4 | 58.6 |
>
> We find that EGGROLL significantly outperforms GRPO when starting from Qwen3-4B and -Base, with this holding across both DeepScaleR and ORZ datasets.
>
> We ran EGGROLL with 256 population members per GPU, and then scaled the number of GPUs for larger population sizes. Our results with EGGROLL population size 256 train on one GH200 GPU for 9hrs, hence showing EGGROLL can match GRPO within the same compute budget. Furthermore, we found that GRPO consistently plateaued, and hence these results are effectively the strongest GRPO performances for this setup (context length, initial model, sampling parameters), even with unlimited compute budget. By contrast EGGROLL continued to improve with more compute to significantly outperform GRPO, as we show with population sizes 4k and 16k.
>
> The reference suggested by reviewer gBKy uses a 4x longer context length than our results, and we noticed this too late to run a comparable experiment in time for the rebuttal. We commit to reporting results for EGGROLL with this longer context length for the camera-ready version.
>
> # Catastrophic Forgetting (Q4)
>
> Given the recency of the referenced preprint (released on the day of the ICML deadline), we were unaware of this work prior to our submission. We believe that the “catastrophic forgetting” concern reported in that preprint has multiple confounding factors that invalidate their concerns. The prior work they build on (Evolution Strategies at Scale) demonstrates that unregularised ES is better than unregularised GRPO (on the Pareto frontier of KL-divergence vs reward). The referenced preprint adds a KL-divergence term to their loss function for GRPO, and they find it is more regularised than unregularised ES, but **they never attempt to add the same regularisation term to the fitness function of ES!** Our quantised distillation experiments demonstrate that we can directly optimise for KL-divergence with EGGROLL, so we can easily regularise to prevent forgetting by just adding a weighted regularisation term to the fitness function.
>
> # Other Comments
>
> Q1,W1: Please refer to the “Motivation Behind Experiments” section for 4SQb; we are happy to add wall-clock curves in the camera-ready version.
>
> Q2,3: Please refer to the “Theory vs Practice” section for fPHc. The theory error is a typo, it should read $\mathcal{O}(\frac{1}{r})$, many thanks for spotting it!
>
> W3: Our bar charts (fig 4b, 14b) are already averaged over three seeds, and the results of each run are within 1 question of the mean; we will add confidence intervals in the camera-ready version.
>
> W4: We are unfamiliar with prior works for quantised model self-distillation in a directly comparable setting to ours; QAT requires floating-point activations for backward passes while QLoRA adds additional floating-point parameters. We are happy to compare against methods suggested during the discussion phase for the camera-ready version.

---

> > ### Author Rebuttal · Reviewer_gBKy · 2026-04-04
> >
> > I appreciate the authors' responses. The new transformer GRPO experiments are a welcome addition and make the empirical case much stronger beyond RWKV. That said, several items are deferred to camera-ready, and I think they need to be provided during the discussion period since they directly affect my assessment:
> >
> > 1. Wall-clock curves (W1/Q1): these can be shared now via an anonymized link. Without them, the practical value of EGGROLL for pretraining is hard to evaluate.
> >
> > 2. Confidence intervals (W3): if individual run data already exists, adding error bars is minimal effort. No reason to defer this.
> >
> > 3. Forgetting measurement (Q4): the argument that KL regularization "could" fix forgetting is reasonable but unverified. A simple forward-pass benchmark (e.g., pass@k on GPQA/MMLU) on the fine-tuned vs. base model would suffice and is feasible now.
> >
> > 4. Theory validation (Q2): the theoretical results are still disconnected from experiments. Even a basic diagnostic (e.g., update discrepancy vs. rank r on a controlled problem) would help.
> >
> > Without this evidence I can't raise my score. If the authors provide even a subset of these during discussion, I'm happy to revisit.

---

> > > ### Author Response · Authors · 2026-04-08
> > >
> > > We originally kept rebuttals self-contained as reviewers were not required to follow anonymous links. We have attached new results to this link: https://docs.google.com/document/d/e/2PACX-1vTxGo7afMXq4FXxe-QtKRrxWOof0GRlfV-se1ZMywamO1uoKPIwjL6rkZntOVNLpB4jJo4Dzq33ruvz/pub
> > >
> > > We unfortunately faced technical issues with our larger cluster over the Easter break, so we only had a subset of checkpoints available to test. We plan to test the remaining checkpoints for the camera-ready version but we hope the current results help address your concerns.
> > >
> > > # Wall-clock curves
> > > Unlike the other experiments in the paper, we wanted to demonstrate data-efficiency and the impact of population size over wall-clock time. EGGROLL is currently far from competitive with backprop SGD for pretraining; we shared that 180x more compute is necessary for the largest population sizes for the same data budget to highlight its current impracticality. Put more explicitly, EGGROLL is **not** a compute-efficient alternative to backprop for pretraining with standard neural network architectures and loss functions on standard hardware.
> > >
> > > # Confidence Intervals
> > > We would like to correct our statement in the rebuttal that bar charts were averaged over 3 seeds; we ran 3 seeds and selected the checkpoint with the highest AIME24 performance as explained in the original caption. We have re-plotted Figure 4b with the bars at the mean of the 3 seeds and showing the min and max as confidence intervals.
> > >
> > > # Forgetting Measurement
> > > In Table R.3 we tested Qwen3-4B models trained with EGGROLL and GRPO on a basic “forgetting” benchmark. We find it is not a major concern, with relatively random fluctuations from the baseline. In the camera-ready version we will run more seeds to get confidence intervals.
> > >
> > > # Theory Validation
> > > We conducted additional experiments to check update discrepancy versus rank on a controlled setting. Denote the $N$-sample Monte Carlo estimate of $\hat{g}^{r}\_\textrm{LR}$ as $\hat{g}^{N,r}\_\textrm{LR}$ and the $N$-sample Monte Carlo estimate of the full-rank update $\nabla\_\mu J(\mu)$ as $\hat{g}^{N}\_\textrm{True}$. We study the convergence rate of $\mathbb{E}\left[\lVert\hat{g}^{N,r}\_\textrm{LR}-\hat{g}^{N}\_\textrm{True}\rVert\_F \right]$. We note that, for finite $N$, the limit  $\lim\_{r\rightarrow\infty}\mathbb{E}\left[\lVert\hat{g}^{N,r}\_\textrm{LR}-\hat{g}^{N}\_\textrm{True}\rVert\_F \right]=\mathbb{E}\left[\lVert \hat{g}^{{N'}}\_\textrm{True}-\hat{g}^{N}\_\textrm{True}\rVert\_F \right]=b$ converges to some base value $b$ where we expect $b>0$ for $N<\infty$ because $\hat{g}^{{N'}}\_\textrm{True}$ and $\hat{g}^{N}\_\textrm{True}$ are both independent estimators formed from independent samples and so won't automatically cancel in the limit unless $N\rightarrow\infty$. The value of $b$ thus represents the \emph{floor noise} that arises from the variance due to Monte Carlo sampling in the estimators. Our goal is to investigate the rate of convergence to floor noise value $b$, that is a relationship of the form: $\mathbb{E}\left[\lVert\hat{g}^{N,r}\_\textrm{LR}-\hat{g}^{N}\_\textrm{True}\rVert\_F \right]=\mathcal{O}(r^{-1})+b$, which would validate our theory under the Monte Carlo sampling regime encountered in practice.
> > >
> > > We conducted additional empirical experiments to confirm this relationship. Specifically, we constructed a 2-layer ReLU MLP with hidden dimension 64 (for a total of 4736 parameters) with He initialisation, to estimate the ES gradient with sigma=1.0 using cross-entropy loss on the first 10 samples of MNIST, using a population size of $N=100,000,000$. We used 10 seeds to estimate the floor noise $b$ by estimating $\mathbb{E}\left[\lVert \hat{g}^{{N'}}\_\textrm{True}-\hat{g}^{N}\_\textrm{True}\rVert\_F \right]$ and 10 seeds to estimate $\mathbb{E}\left[\lVert\hat{g}^{N,r}\_\textrm{LR}-\hat{g}^{N}\_\textrm{True}\rVert\_F \right]$ for $r\in\\{1:32\\}$. We fit a curve of the form $ar^{-p}+b$ to the data, obtaining $ p = 1.191 \pm 0.003$ and $b=0.0654 \pm 0.0003$. The coefficient of determination was $R^2=0.968$.
> > > Results are shown in Figures R.4a and R.4b, where we also plot the upper bound of the form $\mathcal{O}(r^{-1})+b$. These results empirically validate our claim, as $ p = 1.191 \pm 0.003>1$, confirming our upper bound on convergence rate as rank $r$ increases is correct, even in the Monte Carlo sampling regime.

---

### Official Review · Reviewer_4SQb · 2026-03-10

**Soundness:** 3
**Presentation:** 2
**Significance:** 3
**Originality:** 3
**Overall Recommendation:** 5
**Confidence:** 3

**Summary:**

The paper proposes EGGROLL, a scalable evolution strategies framework that uses low rank perturbations to enable efficient population based optimization of very large neural networks. The method restructures parameter perturbations as low rank matrices so that large populations can be evaluated in a single batched forward pass, improving arithmetic intensity and hardware utilization. Empirical experiments include integer RNN pretraining, reinforcement learning control tasks, and LLM finetuning for reasoning tasks. The results suggest that EGGROLL can scale ES style optimization to large models and achieve competitive performance with gradient-based approaches in some settings.

**Compliance With Llm Reviewing Policy:**

Affirmed.

**Final Justification:**

The authors have fully addressed my concerns regarding the motivation, the scope of contribution, and the benchmarking comparisons. I decide to raise my review score from 4 to 5.

**Key Questions For Authors:**

1) The update rule used in EGGROLL resembles natural evolution strategies (NES) or OpenAI-ES style gradient estimation rather than classical evolution strategies (ES). The authors should clarify whether the theoretical analysis and empirical claims in the paper apply specifically to NES style methods such as OpenAI-ES, or whether they are intended to generalize to classical ES algorithms more broadly? If the claims are limited to NES style formulations, this distinction should be clarified in the paper.

2) The paper emphasizes the advantage of massive parallelization through large populations of perturbations evaluated in a single batched forward pass. However, gradient-based methods such as LoRA fine tuning and RL methods such as GRPO can also achieve substantial parallelism through batched rollouts and distributed training. In which practical regimes the parallelization of EGGROLL provides clear advantages over these existing approaches?

3) GRPO and other policy gradient methods can also optimize discrete or non-differentiable reward functions through expectation-based gradient estimators. In which practical scenarios does EGGROLL provide advantages over GRPO and other policy gradient approaches?

4) The paper states that EGGROLL achieves similar performance to GRPO when fine tuning transformer models in Appendix L. However, the RL baseline results appear to be taken from prior work rather than reproduced in the same experimental setup. The authors need to clarify which RL algorithm was used in those experiments and whether the comparison was conducted under the same hardware and training configuration.

5) In Section 6.1 the pretraining experiment uses a nonlinear RNN architecture operating entirely in int8 datatypes where the only nonlinearity are from clipping operations. This architecture appears somewhat tailored to this gradient-free optimization experiment and does not resemble transformer-based architectures in modern language models.

**Limitations:**

yes

**Strengths And Weaknesses:**

Soundness: The technical formulation of the algorithm appears generally sound and the hardware-motivated design is well justified. However, the algorithm described in the paper is closer to natural evolution strategies (NES) or OpenAI-ES style gradient estimation rather than classical evolution strategies (ES) in evolutionary computation. The update rule essentially estimates a gradient of the expected fitness using score function estimators, which aligns more with NES than with the typical selection-mutation iterative mechanism in ES frameworks.

Presentation: The paper is generally well written. However, some experimental comparisons and baseline descriptions are not presented with sufficient clarity, particularly the RL baselines used in Appendix L and the relationship between GRPO and the reported RL results. In addition, some experimental choices appear tailored to the proposed method, which makes it harder to evaluate the broader applicability of the approach.

Significance: The paper demonstrates that population-based zeroth-order methods can be efficiently applied in deep learning optimization. However, the empirical evidence for advantages in foundation model training remains somewhat limited. In particular, the experiments do not convincingly show that EGGROLL truly outperforms GRPO on standard transformer-based LLM finetuning, and the only pretraining experiment uses a small integer RNN architecture that is not representative of modern LLMs.

Originality: The contribution primarily lies in the system design and integration of several existing ideas rather than in proposing fundamentally new optimization algorithms. In particular, the combination of ES with LoRA style low-rank perturbations and batched inference systems is interesting and well-motivated from the perspective of improving hardware efficiency and scalability. While many of the individual components (e.g., ES gradient estimators and low-rank parameterizations) are already well-established, their integration into a single practical framework represents a non-trivial contribution.

---

> ### Author Rebuttal · Authors · 2026-03-31
>
> Thank you for the feedback! We appreciate that you found our hardware-motivated method sound and novel and that you found our paper well-written. We would like to answer your questions in this rebuttal and clarify the motivation behind our experiments.
>
> # Definition of Evolution Strategies (Q1)
>
> The term “evolution” is quite overloaded in AI, so we tried to follow the most standard definitions in current literature. In particular, the evolution “strategies” we refer to are on the lineage of CMA-ES, NES, and OpenAI’s ES, which we distinguish from population-based “genetic algorithms” like deep neuroevolution. We are more than happy to clarify our usage of ES in the paper, like “search-gradient ES” in contrast to population-based genetic algorithms.
>
> As stated, dimension analysis is for EGGROLL and Gaussian ES. Extension to NES OpenES is trivial - the inverse Fisher information matrix term results in an update that just multiplies the Gaussian ES update by $\sigma^2$ and so that analysis holds automatically. This provides an interesting result as it implies convergence to linearity is not possible, but deserves a more thorough investigation. Generalising our results to non-Gaussian population distributions is possible given the strong foundations, and is something we are working on for a separate theoretical paper; both of these directions lie beyond the scope of a single conference paper.
>
> # Motivation Behind Experiments (Q5)
>
> Our goal, when writing this paper, has been to give readers a balanced understanding of the potential of black-box optimisation (which is efficiently scalable thanks to EGGROLL) relative to backprop along with its drawbacks. To this end, we conducted experiments that have natural backprop baselines while also highlighting settings that were previously impractical.
>
> Our EGG pretraining experiment was explicitly designed to highlight some key features of EGGROLL, which we detail in Appendix G.1. We emphasise that EGG is very hardware-friendly at inference time while being impractical to train with standard backprop. Our comparison against standard dense fp32 Transformers in our plots (figure 1b, 7) only serves as a point of reference; we expected EGG to **approach** backprop in the limit of scaling population size so the fact that it **surpasses** backprop with sufficient compute is a surprising discovery.
>
> The critical role of these experiments is to demonstrate that pretraining is possible with EGGROLL, which has not been demonstrated with any backprop-free method to our knowledge until now. Given that EGGROLL has reduced the barrier for training down to batched LoRA inference, we can now experiment with other novel architectures, like neurosymbolic systems with external memory or logic modules. Furthermore, future developments in efficient inference-only hardware can be directly leveraged by EGGROLL.
>
> # Scaling EGGROLL vs GRPO (Q2)
>
> Our RWKV experiments highlight that **EGGROLL is already competitive with GRPO** for standard LLM reasoning with the same resources (same number of GPUs for the same wall-clock time).
>
> EGGROLL enables significantly higher experience throughput than backprop-based methods (figure 1(a)), being only marginally slower than batched inference. EGGROLL is very powerful in settings where inference is already happening (i.e. continual learning with human feedback) as we no longer need to waste additional compute for backprop and can even train on inference-only infrastructure.
>
> Scaling up population size benefits EGGROLL (and search-gradient ES more generally) over gradient-based methods, since it optimises a smoothened objective of the fitness function instead of the point estimate. OpenAI’s ES paper found a linear speed-up with population size, and we also find that, for LLM reasoning, EGGROLL benefits from larger population sizes while GRPO requires smaller batch sizes for more steps (caption of figure 1c).
>
> Finally, EGGROLL can handle global-scale distributed training of large models, because it simply needs to synchronise fitnesses instead of gradients, reducing terabits of data bandwidth requirements down to megabits.
>
> # Settings for EGGROLL vs Policy Gradients (Q3)
>
> In settings where non-differentiable components can be expressed as part of an MDP, policy gradients are valid, but this may become impractical for neurosymbolic systems with deeper integrations between parts. E.g.  Credit assignment for a neural network can be difficult in a system that iterates between neural and symbolic logic.
>
> Most LLMs are deployed with inference-time scaffolds like tool routers and CoT harnesses where the next-token probability calculation is non-differentiable. Policy gradients cannot optimise through scaffolds whereas EGGROLL can, effectively enabling the optimisation of the LLM harness end-to-end at inference cost.
>
> # Comparison with GRPO on Transformer Models (Q4)
>
> We refer the reviewer to the “Transformer GRPO experiments” section in our response to reviewer gBKy.

---

> > ### Author Rebuttal · Reviewer_4SQb · 2026-04-02
> >
> > The authors have addressed my concerns. I am happy to raise my review score from 4 to 5.
> >
> > Regarding the type of evolution strategies (ES), the authors should be more explicit in the revised version that they are referring to the lineage of NES and OpenAI ES, rather than classical ES. Since the authors do not deal with population-based evolutionary algorithms, the term "population size," which is used frequently in the paper, should instead be clarified as "sample size." A brief definition (i.e., aka) should be sufficient.

---

> > > ### Author Response · Authors · 2026-04-02
> > >
> > > Thank you! Yes, we will more explicitly clarify and define the lineage of ES we build on and "population size" in our revisions of the paper.

---

### Official Review · Reviewer_fPHc · 2026-03-12

**Soundness:** 3
**Presentation:** 2
**Significance:** 4
**Originality:** 3
**Overall Recommendation:** 4
**Confidence:** 3

**Summary:**

The paper proposes the EGGROLL method, an efficient training method for large models based on ES. This method approximates a Gaussian matrix by multiplying two low rank matrices and optimizes matrix operations, enabling efficient sampling and parameter updating on multiple GPU nodes to complete model training. For low rank matrix approximation of Gaussian matrix, the paper provides detailed proof that the update of EGGROLL converges to full rank Gaussian ES update at a fast rate of O (1/r). In addition, the paper also claims that this method can achieve stable pre training of non-linear recurrent language models with pure integer data types, and is competitive with GRPO in inference tasks of large language models in post training. Compared with evolutionary strategies, it still maintains performance without degradation and is faster in reinforcement learning with blank settings.

**Compliance With Llm Reviewing Policy:**

Affirmed.

**Final Justification:**

I appreciate the authors for the responses. Now it is clear and I hope the limitation and reference addition could be included into the paper. Under such condition I raise my score.

**Key Questions For Authors:**

1. What is the hardware setting you used for Qwen experiments (Appendix)?

2. To the best of my knowledge, vLLM does not support parallel batched inference with multiple LLM, since it is tailored for gradient method rather ES. Can the authors further explain how they address this?

3. The core argument of the paper is based on the assumption that both r and d tend towards infinity, resulting in an approximate score function ^ S (E). In practical settings, both r and d are finite. So, what is the specific error caused by approximation? Is LLM fine-tuning sensitive to this approximation?

4. The argument in the paper that low rank matrices approximate Gaussian matrices mentions that variance σ=o (1/√ d) can ensure convergence and linearization. Are there any results indicating that if σ=o (1/√ d) is not satisfied, there will be significant divergence and non convergence?

**Limitations:**

While I expect some in-depth analysis on the limitations of this paper, the authors did not provide such content. I hope the authors could provide some, in particular, at the low-rank approximation aspect (e.g., information loss and corresponding unstable training or premature), and at the efficiency aspect.

**Strengths And Weaknesses:**

Strengths:

1. Novelty: I think this is the first paper that introduces LoRA-like paradigm into evolutionary strategy, and especially, benefiting from this point, the work have extended the ability of evolutionary strategy toward LLM training.

2. Solid theoretical proof: I have read the step-by-step proof and think it is solid.

3. Comprehensive experiments: The authors have

Weaknesses:

1. The writing and content organization of this paper should be further polished. I think especially the methodology part is relatively chaos. For a reader such as a new commers, it will be very easy for him to get lost in the elaboration. For example, the pesudocode is actually not accurate enough, if I understood the paper correctly, the authors did not materialize the noise E until the evaluation (LLM inference) ends. However, this is not the case in the provided pesudocode. Besides, another bad-writing example is.

2. While the authors provide a analysis on the selection of rank r. They actually use rank=1 setting. However, for rank=1, I wonder how much information loss and training oscillation could happen. And why not present the results of rank>1?

3. What is on earth the training time of EGGROLL? Compared to GRPO, is it acceptable? From my angle, the time may still be enormous, even with lora.

4. I also expect more elaboration on how to make 2048 (even more) LLM individuals parallel on GPU node, which is not clearly provided in the paper. Maybe it is possible if we only consider the GPU memory (since you use lora), however, how to inference on such large population in a parallel fashion?

5. The authors have mainly conducted experiments on RNN-like structures, while the only one Qwen experiment is put in Appendix. Does this indicate that the proposed EGGROLL has certain architecutre preference. Maybe the low-rank computation will be more effective on RNN-like models? Why?

---

> ### Author Rebuttal · Authors · 2026-03-31
>
> Thank you for the feedback! We appreciate that you found our method novel and that our paper has solid theoretical proofs and comprehensive experiments. In this rebuttal, we would like to clarify some potential misconceptions and answer your questions.
>
> # Training Time and Large Population Sizes (W3,4, Q2)
>
> For our RWKV fine-tuning results, the wall-clock time for EGGROLL and GRPO are on the x-axis, with both using the same amount of time with the same resources. This demonstrates that EGGROLL is **already practical** in terms of resource use for LLM fine-tuning.
>
> The key idea of our work is that each perturbation is simply a single LoRA instance, so training is as easy as batched LoRA inference, which is supported in vLLM (for our transformer results) and reimplemented from scratch in our main jax codebase. In particular, this means that evaluating thousands of unique perturbations is as easy as batched inference with thousands of separate generations. We explain why this is efficient in section 4.2 and Appendix F.
>
> This is the **critical advantage** of EGGROLL; past ES work would not be able to have such a large population since inference across this large population size in parallel on a single GPU is impossible. As you correctly mention, prior ES works would have needed parallel batched inference with multiple different LLMs, which is unsupported with vLLM and extremely inefficient on hardware, but EGGROLL avoids this by instead reusing efficient batched LoRA inference infrastructure.
>
> # Architecture Preference (W5, Q1)
>
> EGGROLL’s throughput is mainly bottlenecked by the ability to do large batch inference. RNNs like RWKV support much larger batch sizes than Transformers since their fixed state sizes are smaller than the KV-cache of Transformers, making them better for batch inference and therefore better for EGGROLL. Notably, the low-rank computation is not relevant for why RWKV is preferable to Transformers.
>
> We note that our placement of the transformer results in the appendix was due to space constraints before the deadline, rather than any weakness in performance. Furthermore, since the paper deadline we have run additional experiments where we find that EGGROLL significantly outperforms GRPO with Qwen3-4B (see “Transformer GRPO experiments” in reviewer gBKy response).
>
> # Theory vs Practice (W2, Q3,4)
>
> As you correctly point out, some of our theoretical analysis focuses on higher ranked perturbations, proving that it approaches standard ES in the limit. In our experiments, we found that having a rank of r=1 was sufficient for large (>1B) models and that the reduction in speed due to increasing r was almost never worth the tradeoff for slightly more accurate updates.
>
> Our analysis of the limits as r and d increase to infinity exists to **theoretically validate** our modification to standard ES. In practice, we find that sufficiently large models (>1B) **already behave similarly** to the limiting behavior of the theory as dimensionality increases, so we do not experiment with larger values of r. In practice, the error due to the approximation (Q3) is negligible relative to the variance in our gradient estimate, even at large population sizes.
>
> Regarding the scale of the noise (Q4), we provide a counterexample formally showing **provable** and **significant divergence** for general cubic functions (i.e. updates tending toward infinity as $d$ increases) with full details in Appendix D.2. Critical Convergence Rate. This naturally includes a large subset of convex optimisation, which contain some of the most well-behaved optimisation problems. Moreover, as cubics represent smooth, local third order Taylor expansions to functions (for example around local minima), this demonstrates the issue for all locally smooth fitness functions, which is a strictly more general result than running simulations over the function space. We can move this result to the main body given the extra page space in a camera ready version.
>
> # Pseudocode Clarity (W1)
>
> We aim to continue polishing our presentation in the camera-ready version of the paper. Our pseudocode in Algorithm 1 was designed to mirror the pseudocode in Algorithm 2 of “Evolution Strategies as a Scalable Alternative to Reinforcement Learning” and highlight the **mathematical** difference when using a low-rank perturbation. The efficient fitness evaluation on GPUs is an important implementation detail, described in section 4.2, but is abstracted away in the pseudocode.

---

> > ### Author Rebuttal · Reviewer_fPHc · 2026-04-01
> >
> > I think the responses are still less-detailed. I will post follow-up questions ASAP.
> >
> > ======================
> > I have two more questions for the authors:
> >
> > I still have questions on details how the authors train Qwen models with EGGROLL. Can you further explain how youe match EGGROLL with batch-lora? A more detailed step-by-step elaboration?
> >
> > The authors have to explicitly indicate EGGROLL's potential limitations to ensure an objective perspective.
> >
> > I have also an additional reference suggestion:
> >
> > https://arxiv.org/abs/2512.05760, where the authors also explore how ES could be used for LLM's thinking enhancement.

---

> > > ### Author Response · Authors · 2026-04-08
> > >
> > > # EGGROLL Implementation
> > >
> > > In our supplemental materials, we share the main jax codebase and an ipynb notebook (HyperscaleES/eggroll.ipynb) that demonstrates how EGGROLL is applied to a simple training example.
> > >
> > > The central logic replaces batched inference with batched LoRA inference as follows:
> > > ```
> > > def forward(base_perturbation_key, sigma, parameter, x, thread_id, rank=1):
> > >     key = jax.random.fold_in(base_perturbation_key, thread_id)
> > >     a, b = parameter.shape
> > >     perturbation = jax.random.normal(key, (a+b, rank))
> > >     B = lora_params[:b]  # b x r
> > >     A = lora_params[b:]  # a x r
> > >     return x @ parameter.T + x @ B @ A.T * sigma
> > >
> > > batch_forward = jax.vmap(forward, in_axes=(None, None, None, 0, 0))
> > > ```
> > >
> > > For vLLM, here are the steps performed at each update:
> > > 1. Save multiple LoRAs to disk using random normal initialisation, scaled by the sigma hyperparameter. Note that these are “untrained” and are different from standard initialisation (which typically saves either A or B as a vector of zeros); both A and B are sampled from random normals as is the case with the jax implementation.
> > > 2. Conduct MultiLoRA inference (documented [here](https://docs.vllm.ai/en/v0.4.1/getting_started/examples/multilora_inference.html)) to use the different LoRAs to generate COT and answers to different questions.
> > > 3. Score the answers to get the “fitness” of each LoRA perturbation.
> > > 4. Calculate the fitness-weighted average of the perturbations as the pseudogradient, and update the original weights using gradient ascent (weighted by a learning rate hyperparameter).
> > >
> > > # Limitations
> > >
> > > Our experiments explore the utility of EGGROLL across a variety of settings to highlight both the strengths and weaknesses relative to prior work. For instance, EGGROLL outperforms GRPO for LLM reasoning, but in tabula-rasa RL and MARL, EGGROLL (and OpenES) are worse than PPO. Similarly, EGGROLL is not a compute-efficient method for LLM pretraining, but it is a clear sign of life for black-box pretraining from scratch; future algorithms and hardware can build upon EGGROLL to decrease the efficiency gap.
> > >
> > > In general, EGGROLL should be treated as a hardware efficient variant of evolution strategies that enables large population sizes, typically inheriting the general strengths and weaknesses of ES. However, there are some exceptions that arise from the fact that we are *simulating* the effects of a perturbation instead of directly applying the perturbation. In particular, hardware peculiarities like those presented in “Nonlinear computation in deep linear networks” may not be correctly handled with EGGROLL since the simulated perturbation follows a different computation pipeline. Furthermore, ES methods based on search gradients (EGGROLL, OpenES, CMA-ES) have different properties from population-based neuroevolution methods, as the latter can explicitly maintain a diverse set of solution candidates instead of simply treating the population as a probability distribution centered around a single set of mean parameters. There is significant potential in merging EGGROLL with population-based methods, but we delegate this to future work.
> > >
> > > Likewise, high-dimensional analysis, whilst rigorous and having mild assumptions that are not violated in practice, currently applies for all settings we consider within the scope of the paper (Gaussian ES and EGGROLL with any sub-Gaussian generating distribution) rather than settings beyond this. The theoretical framework lays the foundations for analysis of general ES approaches, however this should be treated as a separate theory specific project.

---

### Official Review · Reviewer_sMe7 · 2026-03-13

**Soundness:** 3
**Presentation:** 3
**Significance:** 4
**Originality:** 3
**Overall Recommendation:** 5
**Confidence:** 4

**Summary:**

This work introduces Evolution Guided GeneRal Optimisation via Low-rank Learning (EGGROLL), a method for enabling billion-parameter scale Evolution Strategies (ES). EGGROLL replaces full-rank perturbation matrices with low-rank decompositions—similar to LoRA, but with ES perturbations. The key finding is that low-rank perturbations dramatically improve arithmetic intensity on GPUs—enabling population sizes of  up to 1M on a single GH200 GPU and up to 91% of pure batch inference throughput. The paper provides a rigorous theoretical analysis on how the low-rank approximation converges to the true ES gradient, as well as a high-dimensional analysis that shows critical noise scaling regimes for convergence/divergence. Further, the paper introduces a hardware-efficient implementation, and a broad range of experiments on RNN LM (EGG), LLM fine-tuning (outperforming GRPO), RL across 16 environments, multi-agent RL, quantised model distillation, and high-frequency trading.

**Compliance With Llm Reviewing Policy:**

Affirmed.

**Final Justification:**

The authors have addressed most of my concerns in the rebuttal process and I have adjusted my recommendation to "accept".

**Key Questions For Authors:**

a) How does EGGROLL respond to changes in rank? (you used r=4 for HFT and r=1 in other experiments)
b) How does EGGROLL's performance compare to similar recent methods:
https://arxiv.org/pdf/2509.24372 - Qiu et al. (2025)
https://arxiv.org/pdf/2511.16652 - Korotyshova et al. (2025)
c) How does the total compute (GPU-hours) compare between EGGROLL and backprop/GRPO for the LLM-finetuning experiments?

**Limitations:**

Yes

**Strengths And Weaknesses:**

The theoretical analysis appears thorough and the convergence result is nicely presented in Theorem 5.1.  The high-dimensional analysis identifying the three regimes (convergence, critical, divergence) as a function of sigma_d was a nice touch. The RL comparison demonstrating large wall-clock time speedups on most enviornments. I'm a bit skeptical of the GRPO comparison using RWKV, since it raises the question whether the architectural fit is the reason for the performance gap. Another concern was the integer pre-training experiment in comparison with the fp32 transformer trained using backpropagation, since this seems like an apples to oranges comparison.

The paper has a clear structure and is well written. Explanations are clear and intuitive.

he paper demonstrates the benefit of being able to optimize a non-differentiable objective (advantage over gradient based methods). The paper's contributions are significant and can translate beyond this work. As is common with evolutionary methods, EGGROLL is comptuationally inefficient when compared to backprop, and thus seems to make EGGROLL more of contender for situations where backprop is not possible or not practical.

The low-rank perturbation idea is novel (thought the authors acknowledge Choromanski et al. (2019) and Garbus & Pollack (2025) for having similar ideas). Further, EGGROLL's combination of low-rank perturbations, counter based RNG reconstruction and batched LoRA-style inference kernels for orders-of-magnitude improvements in throughput is novel.

---

> ### Author Rebuttal · Authors · 2026-03-31
>
> Thank you for the feedback! We appreciate that you find our technique novel and significant and that the paper itself is well written, with thorough theoretical analysis. We would like to clarify some potential misconceptions in this rebuttal.
>
> # EGGROLL is Computationally Efficient (Qc)
>
> We would like to highlight that EGGROLL has a very high experience throughput (figure 1a), imposing very little overhead relative to standard batched inference.
>
> In our RWKV fine-tuning experiments, we use **identical compute** for EGGROLL and GRPO; note that x-axis in Figures 1c and 4a is wall-clock time using the same number of GPUs. Since GRPO has a much lower experience throughput than EGGROLL, we swept for the best hyperparameter configuration for GRPO in terms of final performance within our fixed wall-clock time, which was ultimately fewer parallel generations at more updates (caption of figure 1c).
>
> Based on our LLM reasoning results, we argue that EGGROLL is **already a practical method** in settings where backprop is being used, in addition to unlocking **new use-cases** where backprop is impractical as you highlighted.
>
> # Architectural Fit
>
> We chose RWKV due to its strong batch inference qualities (namely a fixed-size hidden state), which enabled us to implement EGGROLL and (DR)GRPO within a unified jax codebase for fair comparisons. We find that EGGROLL for Qwen3-4B (reimplemented in vLLM) also outperforms GRPO; see the “Transformer GRPO experiments” in our response to gBKy for new results.
>
> For pretraining, we wanted to emphasise that EGGROLL works best with architectures that have high throughput during batched inference, whereas the key benefit of Transformers is the ability to do sequence-parallel training to avoid backpropagation through time. Our integer pretraining experiments serve as a proof-of-concept that (1) alternative AI architectures that are hard or impossible to train under backprop are now trainable and that (2) backprop-free training from scratch is finally feasible thanks to EGGROLL, even if it is not necessarily optimal relative to backprop. The fp32 transformer baseline (using the standard dense transformer recipe) was provided for context to highlight the necessity of large population sizes (enabled by EGGROLL), not to prove that EGGROLL is superior to backprop for pretraining.
>
> # Changes in Rank (Qa)
>
> For >1B parameter models, we did not see any benefit from increasing rank beyond 1. We included higher ranks for our tabula-rasa RL sweeps, where we randomised hyperparameters and chose the configuration with the highest return within a fixed number of steps, regardless of wall-clock time (see N.1 for a more detailed description). Due to the random nature of this sweep and the unimportance of higher ranks for final return, the selected hyperparameters often had a rank greater than 1 by chance.
>
> For HFT experiments, we similarly swept for highest return in a fixed number of steps (not wall-clock time), and we found that r=4 has slightly higher returns than r=1. If our objective was instead the best performance given a compute budget, r=1 would have been chosen instead. We will include these results in the camera-ready version of our paper.
>
> We note that r=1 has significantly lower wall-clock time, so we recommend typically using r=1 in practice.
>
> # Other ES works (Qb)
>
> We provide a comparison against both works mentioned in your question in the second paragraph of section 3.2. The key point is that **EGGROLL efficiently allows different rollouts to have different perturbations**, enabling a much larger population size without compromising token generation throughput. Our experiments have on the order of thousands of unique perturbations per GPU while the other works have tens to hundreds, and must load each perturbation separately instead of in parallel (within each GPU).

---

> > ### Author Rebuttal · Reviewer_sMe7 · 2026-04-04
> >
> > Considering the entirety of rebuttals provided, the authors have addressed all my concerns. The question of how many GPU hours were used is still of interest and should be included if possible, but not a pre-condition for publishing this work in my opinion.

---

### Decision · Program_Chairs · 2026-04-30

**Decision:**

Accept (regular)

**Comment:**

This work introduces evolution guided general optimisation via low-rank learning, a method for enabling billion-parameter scale evolution strategies. After the rebuttal, all reviewers are positive. They generally acknowledged the novelty of the proposed method, the theoretical analysis (on how the low-rank approximation converges to the true ES gradient), and comprehensive experiments (on RNN LM, LLM fine-tuning, RL across 16 environments, multi-agent RL, quantised model distillation, and high-frequency trading).